# Implicit Regularisation in Diffusion Models: An Algorithm-Dependent Generalisation Analysis

**Tyler Farghly, Patrick Rebeschini, George Deligiannidis & Arnaud Doucet**
Department of Statistics
University of Oxford
`{farghly,rebeschini,deligian,doucet}@stats.ox.ac.uk`

## Abstract

The success of denoising diffusion models raises important questions regarding their generalisation behaviour, particularly in high-dimensional settings. Notably, it has been shown that when training and sampling are performed perfectly, these models memorise training data—implying that some form of regularisation is essential for generalisation. Existing theoretical analyses primarily rely on algorithm-independent techniques such as uniform convergence, heavily utilising model structure to obtain generalisation bounds. In this work, we instead leverage the algorithmic aspects that promote generalisation in diffusion models, developing a general theory of algorithm-dependent generalisation for this setting. Borrowing from the framework of algorithmic stability, we introduce the notion of score stability, which quantifies the sensitivity of score-matching algorithms to dataset perturbations. We derive generalisation bounds in terms of score stability, and apply our framework to several fundamental learning settings, identifying sources of regularisation. In particular, we consider denoising score matching with early stopping (denoising regularisation), sampler-wide coarse discretisation (sampler regularisation) and optimising with SGD (optimisation regularisation). By grounding our analysis in algorithmic properties rather than model structure, we identify multiple sources of implicit regularisation unique to diffusion models that have so far been overlooked in the literature.

## 1 Introduction

Diffusion models (Sohl-Dickstein et al., 2015; Ho et al., 2020; Song et al., 2021) are a class of generative models that have achieved state-of-the-art performance across image, audio, video, and protein synthesis tasks (Rombach et al., 2022; Saharia et al., 2022; Ramesh et al., 2022; Watson et al., 2023; Esser et al., 2024). Their ability to generate high-quality samples from complex, high-dimensional distributions with limited data motivates the need for a theoretical understanding of the mechanisms underpinning their strong generalisation capabilities.

The goal of diffusion models is to generate new synthetic samples from a data distribution $\nu_{\text{data}}$ using a finite set of $N$ data points $\{x_i\}_{i=1}^N$. Central to the methodology is a unique approach to generating data, formulating it as the iterative transformation of noise into data, or equivalently, the reversal of a diffusion process (Song et al., 2021). This diffusion process, called the *forward process*, is defined by the stochastic differential equation (SDE),

$$dX_t = -\alpha X_t \, dt + \sqrt{2} \, dW_t, \qquad X_0 \sim \nu_{\text{data}}, \qquad t \in [0, T], \tag{1}$$

for some $\alpha \geq 0$, where $W_t$ denotes the Brownian motion in $\mathbb{R}^d$ and $T > 0$ is the terminal time. It can then be shown that the time-reversal of this process, $Y_t := X_{T-t}$ admits a weak formulation as a solution to the SDE,

$$dY_t = \alpha Y_t dt + 2\nabla \log p_{T-t}(Y_t) dt + \sqrt{2} dW_t, \qquad Y_0 \sim p_T, \qquad t \in [0, T), \tag{2}$$

where $p_t$ denotes the marginal density of $X_t$ (Haussmann & Pardoux, 1986). Therefore, simulating samples from $\nu_{\text{data}} = p_0$ can be achieved by solving the diffusion process in (2), which requires

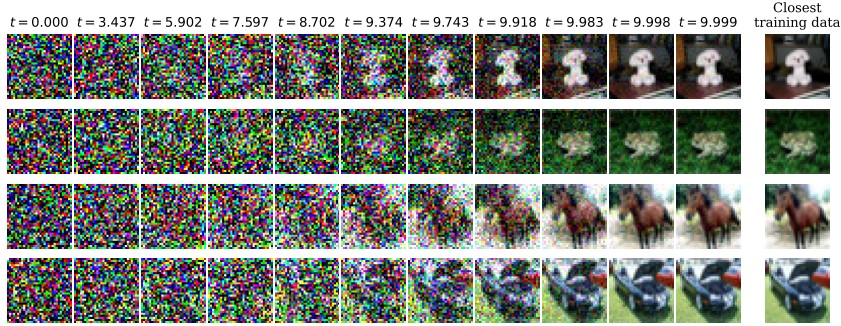

Figure 1: Samples generated using the empirical score function on CIFAR-10 compared to the closest image in the dataset, illustrating memorisation of the training data.

an approximation of the score function, $\nabla \log p_t$. This is achieved by fitting a time-dependent deep neural network to minimise a weighted $L^2$-distance called the *(population) score matching loss*:

$$\ell_{\mathrm{sm}}(s;\tau) := \int \mathbb{E}_{X_t}[\|s(X_t,t) - \nabla \log p_t(X_t)\|^2]\,\tau(dt), \tag{3}$$

where $\tau$ is a probability measure over $(0,T]$ that determines the weighting of the timepoints. Since $\nabla \log p_t$ is unknown, typically the (population) *denoising score matching loss* $\ell_{\mathrm{dsm}}$, which differs from $\ell_{\mathrm{sm}}(s)$ only by a constant, is used instead and is then approximated using the dataset, forming the *empirical denoising score matching loss* $\hat{\ell}_{\mathrm{dsm}}$ (see equations (6) and (8)). The score network $s(x,t)$ is trained on this objective using standard stochastic optimisation methods relying on mini-batching. Once an approximation is obtained, samples are generated by numerically solving the reverse-time SDE, (2). Both *score matching* and *sampling* introduce distinct challenges and design choices that impact the quality of model output (Karras et al., 2022).

Score matching presents a key difference from standard supervised learning. In the space of all $L^2$ score functions, the empirical objective $\hat{\ell}_{\mathrm{dsm}}$ possesses a unique minimiser—the empirical score function—as a result of the integration over $X_t|X_0$ (see Lemma 1). This contrasts with traditional supervised learning, where the empirical risk minimisation problem can have infinitely many solutions (e.g., in overparameterised regression) and often requires regularisation to be well-posed. As shown in Figure 1, sampling with this empirical score leads to exact recovery of the training data (Pidstrigach, 2022). This behaviour is distinct from 'benign overfitting', a phenomenon from the deep learning literature where interpolating the data does not necessarily prevent generalisation (Bartlett et al., 2021; Zhang et al., 2021). This divergence suggests that existing theory may be insufficient to explain the success of diffusion models, highlighting the need for new frameworks tailored to this setting.

Recently, there has been a drive towards developing theory for better understanding the unique structure of diffusion models. The most developed subset of this work focuses on connecting sample quality to score matching by deriving upper bounds on distribution error (e.g. KL divergence, total variation, or Wasserstein distance) between model samples and the data distribution, controlling it by the population score matching loss (De Bortoli et al., 2021; De Bortoli, 2022; Lee et al., 2022; Chen et al., 2023; Benton et al., 2024; Potaptchik et al., 2024). These results, often referred to as *convergence bounds*, typically take the form,

$$\text{Distribution error} \lesssim \ell_{\mathrm{sm}}(s) + \Delta,$$

where $\Delta$ is the discretisation error of the sampling scheme, which can be made small with sufficiently fine discretisation. However, since $\ell_{\mathrm{sm}}$ is not computable, these bounds say little about performance under empirical guarantees—that is, their *generalisation* properties. One line of work, initiated by Oko et al. (2023) and extended in (Azangulov et al., 2024; Tang & Yang, 2024), applies classical uniform convergence theory to bound the generalisation gap from the decomposition,

$$\ell_{\mathrm{sm}}(s) = \hat{\ell}_{\mathrm{sm}}(s) + \underbrace{\ell_{\mathrm{sm}}(s) - \hat{\ell}_{\mathrm{sm}}(s)}_{\text{generalisation gap}}, \tag{4}$$

where $\hat{\ell}_{\mathrm{sm}}$ denotes the empirical counterpart to $\ell_{\mathrm{sm}}$. These results rely on covering number bounds for specific classes of neural networks and, while informative, are limited to carefully chosen

model classes and do not account for algorithmic properties. An alternative approach by De Bortoli (2022) uses a decomposition of the Wasserstein distance that leverages convergence properties of the empirical measure. Though more model-agnostic, this method overlooks how diffusion models generate *novel* data. Both lines of work are fundamentally algorithm-independent, in that they lack any utilisation of the algorithmic aspects that uniquely define diffusion models. Recent efforts aim to incorporate algorithmic effects by restricting the problem. For instance, Shah et al. (2023); Chen et al. (2024) consider Gaussian mixture targets, while Li et al. (2023); Yang (2022) study random feature models. These settings allow for finer analysis of the role of the score matching algorithm, but remain limited in scope, leaving open the challenge of developing a more general algorithm-dependent theory of generalisation in diffusion models.

As noted earlier, if the empirical score matching loss was completely minimised and sampling was performed perfectly, the diffusion model would simply return training data, failing to generalise. Therefore, the observed success of diffusion models in producing novel data implies that, in practice, they either avoid completely minimising $\hat{\ell}_{\mathrm{sm}}$ or must avoid perfectly sampling. This suggests that (implicit) regularisation in the score matching or sampling algorithm is crucial for generalisation, making algorithmic considerations essential for understanding diffusion models.

## 1.1 OUR CONTRIBUTIONS

We introduce score stability, a general, algorithm-dependent framework for analysing diffusion model generalisation based on the classical approach of algorithmic stability. This framework quantifies an algorithm's dependence on individual training examples, from which we derive expected generalisation gap bounds for score matching losses. Using the score stability framework, we then analyse several examples of score matching algorithms, identifying three distinct sources of implicit regularisation in diffusion model training and sampling: noising, sampler, and optimisation-induced regularisation.

**Denoising regularisation.** To begin with, we consider the empirical risk minimisation algorithm (ERM) that minimises $\hat{\ell}_{\mathrm{dsm}}$ over a hypothesis class $\mathcal{H}$. Through a score stability analysis, we reveal a regularisation source within this objective when early stopping of the forward process is used—a standard practice in the diffusion model literature. Utilising properties of the noising forward process, we obtain generalisation gap bounds with near-linear rate, $\epsilon^{-d^*/4}(\epsilon^{-d^*/2}N^{-2} + \min_{\mathcal{H}} \hat{\ell}_{\mathrm{sm}})^{c/2}$ for any $c < 1$, where $\epsilon > 0$ is the early stopping time and $d^*$ is the dimension of the data support.

**Sampler regularisation.** We then apply this analysis to discrete-time sampling algorithms, deriving statistical guarantees for the expected KL divergence between the true data distribution and samples generated by the diffusion model. The bound we derive is formed of two stages: we obtain generic rates $\epsilon^{-1/2}(\epsilon^{-d^*/2}N^{-2} + \min_{\mathcal{H}} \hat{\ell}_{\mathrm{sm}})^{c/d^*}$ but when $N^{-2}$ and $\min_{\mathcal{H}} \hat{\ell}_{\mathrm{sm}}$ are sufficiently small relative to $\epsilon$, we obtain bounds with rates $\epsilon^{-d^*/4}(\epsilon^{-d^*/2}N^{-2} + \min_{\mathcal{H}} \hat{\ell}_{\mathrm{sm}})^{c/2}$ that are faster in $N$ and $\min_{\mathcal{H}} \hat{\ell}_{\mathrm{sm}}$. To derive this bound, we utilise regularisation brought about by the coarseness of the discretisation. We find that by increasing discretisation coarseness, we can improve the generalisation gap bound at the expense of worsening the discretisation error term.

**Optimisation regularisation.** Finally, we consider the role of the optimisation scheme, analysing stochastic gradient descent (SGD) with gradient clipping and weight decay. On the model class, we assume only structural assumptions typical in the optimisation literature, including non-global Lipschitz and smoothness assumptions. While this initially yields bounds that grow with the number of iterations, we more closely inspect the impact of the high-variance gradient estimator used in diffusion training. We show this gradient noise induces a contractive behaviour in the training dynamics, which we harness to obtain stability bounds that do not grow with the number of iterations (Proposition 14), showing that the noisy dynamics enable tighter generalisation guarantees.

## 2 BACKGROUND

Suppose that the data distribution $\nu_{\mathrm{data}}$ is on $\mathbb{R}^d$ and we are provided a finite dataset of samples $S = \{x_1, ..., x_N\}$ which we assume are sampled independently and identically (i.i.d.) from $\nu_{\mathrm{data}}$. As discussed in the introduction, diffusion models are formed of two distinct stages. The first stage, *score matching*, consists of learning an approximation to the score function $\nabla \log p_t$ using the dataset $S$. In

this work, we take a *score function* to be any function belonging to the set $L^0(\mathbb{R}^d \times [0, T]; \mathbb{R}^d)$, the set of Borel measurable functions of the form $\mathbb{R}^d \times [0, T] \to \mathbb{R}^d$. Then, a *score matching algorithm* is taken to be any mapping of the form $A_{\mathrm{sm}} : (\cup_{N=1}^\infty (\mathbb{R}^d)^{\otimes N}) \times \Omega \to \mathcal{H}$ where $\mathcal{H}$ is a measurable subset of $L^0(\mathbb{R}^d \times [0, T]; \mathbb{R}^d)$. Here, $\Omega$ is the event space belonging to a probability space $(\Omega, \mathcal{F}, \mathbb{P})$.

The second stage of diffusion models, *sampling*, consists of generating samples with the learned score function. We take a *sampling algorithm* to be a mapping of the form, $A_{\mathrm{samp}} : \mathcal{H} \to \mathcal{P}(\mathbb{R}^d)$ where $\mathcal{P}(\mathbb{R}^d)$ denotes the set of Borel measures on $\mathbb{R}^d$. Typically, sampling is performed using an approximation to the reverse process given in (2), replacing $\nabla \log p_t$ with the learned score function $s(\cdot, t)$ and replacing its initial distribution, $p_T$ with a data-independent prior, $p_{\mathrm{prior}} = \mathcal{N}(\mathbf{0}, \sigma_{\mathrm{prior}}^2 I)$. In the case of $\alpha > 0$, we choose $\sigma_{\mathrm{prior}}^2 = \alpha^{-1}$ so that the prior coincides with the stationary distribution of the forward process, and when $\alpha = 0$ we simply set $\sigma_{\mathrm{prior}}^2 = 2T$. With this, we arrive at the SDE,

$$d\hat{Y}_t = \alpha\hat{Y}_t dt + 2s(\hat{Y}_t, T - t)dt + \sqrt{2}dW_t, \quad \hat{Y}_0 \sim p_{\mathrm{prior}}. \tag{5}$$

Thus, a sample is generated by sampling from $\hat{Y}_T$, or more commonly, the process is terminated early, sampling from $\hat{Y}_{T-\epsilon}$ for some small $\epsilon > 0$. Therefore, diffusion models are density estimation algorithms formed from the composition $A_{\mathrm{samp}} \circ A_{\mathrm{sm}}$.

**Denoising score matching and overfitting.** As stated in the introduction, computing $\ell_{\mathrm{sm}}$ requires access to the population score function, $\nabla \log p_t$. So instead, the *(population) denoising score matching loss* is used in its place:

$$\ell_{\mathrm{dsm}}(s; \tau) := \mathbb{E}_{X_0 \sim \nu}\left[ \int \mathbb{E}_{X_t|X_0}[\|s(X_t, t) - \nabla \log p_{t|0}(X_t|X_0)\|^2|X_0] \, \tau(dt)\right], \tag{6}$$

which differs from $\ell_{\mathrm{sm}}(s)$ only by a constant $C_{\mathrm{sm}}$, (see Lemma 16) whilst being easier to approximate without access to $\nabla \log p_t$ (Hyvärinen, 2005). Since $p_{t|0}$ is a Gaussian kernel, its score is given by,

$$\nabla_y \log p_{t|0}(y|x) = \frac{\mu_t x - y}{\sigma_t^2}, \qquad \mu_t = e^{-\alpha t}, \qquad \sigma_t^2 = \begin{cases} \alpha^{-1}(1 - \mu_t^2), & \text{if } \alpha > 0, \\ 2t, & \text{otherwise.} \end{cases} \tag{7}$$

In practice, the objective in (6) is further approximated via Monte Carlo estimation using the dataset which leads to the *empirical denoising score matching loss*,

$$\hat{\ell}_{\mathrm{dsm}}(s; S, \tau) := \frac{1}{N}\sum_{i=1}^N \int \mathbb{E}_{X_t|X_0}[\|s(X_t, t) - \nabla \log p_{t|0}(X_t|x_i)\|^2|X_0 = x_i] \, \tau(dt). \tag{8}$$

In the following lemma, we highlight the important property that this can equivalently be defined as the denoising score matching objective for the process $\hat{X}_t$ which evolves as in (1) but with the initial distribution given by the empirical distribution, $\hat{X}_0 \sim \frac{1}{N}\sum_{i=1}^N \delta_{x_i}(dx)$.

**Lemma 1.** *The objective $\hat{\ell}_{\mathrm{dsm}}(s; S, \tau)$ is identical, up to a constant, to the objective*

$$\hat{\ell}_{\mathrm{sm}}(s; S, \tau) := \int \mathbb{E}[\|s(\hat{X}_t, t) - \nabla \log \hat{p}_t(\hat{X}_t)\|^2|S]\tau(dt), \tag{9}$$

*where $\hat{p}_t$ is the marginal density of $\hat{X}_t$. Therefore, any minimiser of $\hat{\ell}_{\mathrm{dsm}}(\cdot; S, \tau)$ on $L^0(\mathbb{R}^d \times [0, T]; \mathbb{R}^d)$ is identical to $\nabla \log \hat{p}_t$ a.e. for any $t \in \mathrm{supp}(\tau)$.*

See Appendix A.2 for the proof. This lemma shows that, unlike in traditional supervised learning problems, the empirical objective here admits a single unique minimiser, the *empirical score function*, $\nabla \log \hat{p}_t$. The nature of this score function and the samples it generates has been the focus of several recent studies, notably (Pidstrigach, 2022) which shows that with perfect sampling, any score function sufficiently close to $\nabla \log \hat{p}_t$ recovers the training data.

**Other notation.** When the score matching algorithm $A_{\mathrm{sm}}$ is random, we use $A_{\mathrm{sm}}(S)$ as shorthand for the random score function $(x, t, \omega) \mapsto A_{\mathrm{sm}}(S, \omega)(x, t)$. Given two random score functions $s, s'$, we let $\Gamma(s, s')$ denote the set of all couplings of these random functions (Appendix A.1 for details).

## 3 SCORE STABILITY AND GENERALISATION

Algorithmic stability is a classical technique in learning theory used to understand the generalisation properties of a variety of important learning algorithms (Kearns & Ron, 1999; Devroye & Wagner, 1979; Bousquet & Elisseeff, 2002; Hardt et al., 2016). While there are various formulations, they all share the common aim of connecting properties of a learning algorithm to its robustness under changes in the dataset. Its use has primarily been focused around regression and classification problems—in this section, we propose a notion of stability that applies specifically to diffusion models.

We introduce the notion of *score stability* which quantifies how sensitive a score matching algorithm $A_{\mathrm{sm}}$ is to individual changes in the dataset. We do this by defining the adjacent dataset $S^i := \{x_1, ..., x_{i-1}, \tilde{x}, x_{i+1}, ..., x_N\}$ where $\tilde{x} \sim \nu_{\mathrm{data}}$, independent from $S$, and then measuring the similarity between the score functions $\hat{s} = A_{\mathrm{sm}}(S)$ and $\hat{s}^i = A_{\mathrm{sm}}(S^i)$.

**Definition 2.** *A score matching algorithm $A_{\mathrm{sm}}$ is score stable with constant $\varepsilon_{stab} > 0$ if for any $i \in [N]$ it holds that,*

$$\mathbb{E}_{S, \tilde{x}}\left[\inf_{(\hat{s}, \hat{s}^i) \in \Gamma_i} \int \mathbb{E}[\|\hat{s}(X_t, t) - \hat{s}^i(X_t, t)\|^2 | X_0 = \tilde{x}, S, \tilde{x}] \, \tau(dt)\right] \leq \varepsilon_{stab}^2,$$

*where $\Gamma_i = \Gamma(A_{\mathrm{sm}}(S), A_{\mathrm{sm}}(S^i))$.*

Since $A_{\mathrm{sm}}$ may be random, we define score stability in terms of the best-case coupling of the random score functions $\hat{s}$, $\hat{s}^i$. We recall that $\Gamma(\cdot, \cdot)$ denotes the set of couplings between two random score functions, and when it is not random, it is given by the singleton $\Gamma_i = \{(A_{\mathrm{sm}}(S), A_{\mathrm{sm}}(S^i))\}$. In the following theorem, we connect score stability to generalisation by controlling the expected generalisation gap by the score stability constant.

**Theorem 3.** *Suppose that the score matching algorithm $A_{\mathrm{sm}}$ is score stable with constant $\varepsilon_{stab}$. Then, with $\hat{s} = A_{\mathrm{sm}}(S)$, it holds that*

$$\left|\mathbb{E}\left[\ell_{\mathrm{dsm}}(\hat{s}; \tau)\right]^{1/2} - \mathbb{E}\left[\hat{\ell}_{\mathrm{dsm}}(\hat{s}; S, \tau)\right]^{1/2}\right| \leq \varepsilon_{stab}. \tag{10}$$

*Furthermore, it holds that*

$$\mathbb{E}\left[\ell_{\mathrm{sm}}(\hat{s}; \tau)\right] - \mathbb{E}\left[\hat{\ell}_{\mathrm{sm}}(\hat{s}; S, \tau)\right] \leq 2\,\varepsilon_{stab}\,\mathbb{E}\left[\hat{\ell}_{\mathrm{dsm}}(\hat{s}; S, \tau)\right]^{1/2} + \varepsilon_{stab}^2. \tag{11}$$

With Theorem 3, we obtain that the generalisation gap for both the denoising score matching loss and the score matching loss decays at the same rate as score stability. We can further simplify the bound for the score matching loss using the fact that $\hat{\ell}_{\mathrm{dsm}}$ and $\hat{\ell}_{\mathrm{sm}}$ are identical up to a constant, to obtain,

$$\mathbb{E}\left[\ell_{\mathrm{sm}}(\hat{s}; \tau)\right] \lesssim \mathbb{E}\left[\hat{\ell}_{\mathrm{sm}}(\hat{s}; \tau)\right] + \varepsilon_{\mathrm{stab}}\,C_{\mathrm{sm}}^{1/2} + \varepsilon_{\mathrm{stab}}^2.$$

One should expect that if the score matching algorithm is effective, both $\hat{s}$ and $\hat{s}^i$ converge to the ground truth as $N$ grows, and thus $\varepsilon_{\mathrm{stab}}$ should decrease to 0. Ascertaining the rate at which $N$ decreases requires an analysis of the algorithm at hand, hence the categorisation of algorithmic stability as an algorithm-dependent approach. This contrasts with uniform learning, which utilises control over the hypothesis class, providing a worst-case bound that is independent from the algorithm.

In the following sections, we apply the framework of score stability to some common learning settings for diffusion models. We derive estimates of the score stability constant for these algorithms and identify features that promote generalisation.

## 4 EMPIRICAL SCORE MATCHING AND IMPLICIT REGULARISATION

We begin our examples by considering the score matching algorithm that minimises the empirical denoising score matching loss. Given a hypothesis class $\mathcal{H} \subseteq L^0(\mathbb{R}^d \times [0, T]; \mathbb{R}^d)$, we define this algorithm by,

$$A_{\mathrm{erm}}(S) = \mathrm{argmin}_{s \in \mathcal{H}} \hat{\ell}_{\mathrm{dsm}}(s; S, \tau).$$

While this algorithm is not often used in practice, it is the natural analogue to empirical risk minimisation from traditional supervised learning and thus serves as a canonical example. We consider the setting of the manifold hypothesis where the data distribution is supported on a submanifold of $\mathbb{R}^d$.

**Assumption 4.** *Suppose that $\nu_{\text{data}}$ is supported on a smooth submanifold of $\mathbb{R}^d$ that has dimension $d^*$ and reach $\tau_{reach} > 0$. Furthermore, its density on the submanifold, $p_\nu$, satisfies $c_\nu := \inf p_\nu > 0$.*

The reach describes the maximum distance where the projection to the manifold is uniquely defined and therefore, it quantifies the maximum curvature of the manifold. We refer to Appendix A.3 for the full definition. Several recent works have considered the assumption that $\nu_{\text{data}}$ lies on a submanifold of $\mathbb{R}^d$. These works argue that $d^*$ can often be far smaller than $d$ and so dependence with respect to $d^*$ over $d$ is favourable (De Bortoli, 2022; Pidstrigach, 2022; Loaiza-Ganem et al., 2024; Potaptchik et al., 2024; Huang et al., 2024). The assumption that the density is bounded from below has also appeared in several of these works (Potaptchik et al., 2024; Huang et al., 2024). We also make the following assumption about the class of score networks.

**Assumption 5.** *Suppose there exists $D_{\mathcal{H}} \geq 0$ such that for any $s, s' \in \mathcal{H}$, it holds that*

$$\|s(\cdot, t) - s'(\cdot, t)\|_{L^\infty} \leq D_{\mathcal{H}}/\sigma_t^2, \qquad \text{for all } t \in \text{supp}(\tau).$$

Under these assumptions, we obtain the following estimate for the stability constant.

**Proposition 6.** *Suppose that assumptions 4 and 5 hold and that $\epsilon := \inf \text{supp}(\tau) \in (0, \tau_{reach}^2)$, then for any $c \in (0,1)$ and sufficiently large $N$, the score matching algorithm $A_{erm}$ is score stable with,*

$$\varepsilon_{stab}^2 \lesssim C\big(CC_{\text{sm}}N^{-2} + \mathbb{E}[\hat{\ell}_{\text{sm}}(\hat{s})]\big)^c, \qquad C = \frac{D_{\mathcal{H}}^2}{\sigma_\epsilon^4} \vee \frac{1}{c_\nu \sigma_\epsilon^{d^*}}.$$

An interesting feature of Proposition 6 is that we obtain generalisation bounds for the empirical risk minimisation algorithm under relatively light assumptions on the structure of the hypothesis class *without any additional regularisation*. This contrasts with algorithmic stability in the setting of traditional supervised learning, where ERM is stable only when restricting the hypothesis class or with the use of explicit regularisation (Zhang et al., 2021; Bousquet & Elisseeff, 2002). Here, we show that the denoising score matching loss possesses the unique property that it is stable without the need for additional regularisation, suggesting that the denoising score matching loss possesses a form of *implicit regularisation.*

When $d^* > 4$, for $\epsilon$ sufficiently small, we have that $C = \mathcal{O}(c_\nu^{-1}\epsilon^{-d^*/2})$, $C_{\text{sm}} = \mathcal{O}(d^*\epsilon^{-1})$. Since the bound only depends on $d^*$ and not $d$, this suggests that diffusion models are automatically manifold-adaptive. The bound also heavily depends on $\epsilon$, with it being smaller for larger $\epsilon$ and growing exponentially fast as $\epsilon$ approaches zero, indicating that the natural regularisation present in the score matching objective is more prevalent at larger noise scales. The requirement to have $\epsilon > 0$ is closely related to the technique of early stopping which is frequently used in the diffusion model literature (Song & Kingma, 2021; Karras et al., 2022). This is where the backwards process $\hat{Y}_t$ is terminated early by some small amount of time to avoid irregularity issues of the score function when close to convergence. Other theoretical works have also identified the importance of early stopping in the generalisation properties of diffusion models (Oko et al., 2023; Azangulov et al., 2024).

**Proof summary** We now provide a brief summary of the proof of Proposition 6. The first step of the proof technique utilises a fundamental property of the empirical denoising score matching objective, $\hat{\ell}_{\text{dsm}}(s; S, \tau)$: that it is strongly convex in $s$ in a data-dependent weighted $L^2$-space. Strong convexity is often used in algorithmic stability analyses, especially in deriving stability bounds for linear models—here we borrow a similar approach, but we analyse the stability of the algorithm in function space. With this, we arrive at the following inequality (see Lemma 19):

$$\int \mathbb{E}[\|\hat{s}(\hat{X}_t, t) - \hat{s}^i(\hat{X}_t, t)\|^2]\tau(dt) \lesssim \mathbb{E}[\hat{\ell}_{\text{sm}}(\hat{s})] + \frac{\varepsilon_{\text{stab}}}{N}(C_{\text{sm}}^{1/2} + \varepsilon_{\text{stab}}), \tag{12}$$

where $\varepsilon_{\text{stab}}$ is the (yet-to-be bounded) score stability constant of $A_{\text{erm}}$.

The second step of the proof technique utilises a characteristic property of the heat kernel—that it smooths out functions. In particular, we utilise the celebrated Harnack inequality of Wang (1997) that captures this property by showing that for any positive measurable $\phi : \mathbb{R}^d \to \mathbb{R}_+$, $x, y \in \mathbb{R}^d$, it holds that

$$\mathbb{E}[\phi(X_t)|X_0 = x] \leq \mathbb{E}[\phi(X_t)^p|X_0 = y]^{1/p} \exp\big(\tfrac{\mu_t^2\|x-y\|^2}{2(p-1)\sigma_t^2}\big),$$

for any $t > 0, p > 1$. Utilising this bound, we convert the upper bound in (12) to a bound on the stability constant. The full proof can be found in Appendix C.

## 5 STOCHASTIC SAMPLING AND SCORE STABILITY

In practice, the backwards process in (5) cannot be sampled exactly, so we instead rely on approximations based on numerical integration schemes. In this section, we investigate how algorithmic stability interacts with these sampling schemes. We consider the Euler-Maruyama-type sampling scheme proposed in (Benton et al., 2024; Potaptchik et al., 2024) which discretises at the timesteps $(t_k)_{k=0}^K$, where $t_k = T - (1 + \kappa)^{\frac{T-1}{\kappa} - k}$ for large $k$. The quantity $\kappa > 0$ is chosen freely and we choose $K \sim \log(\epsilon^{-1})/\log(1 + \kappa)$ so that $t_K \approx \epsilon$ (see Appendix D for details). By sampling its terminating iterate $\hat{y}_K$, we obtain a *sampling algorithm*, $A_{\mathrm{em}}$, that maps a score function $s$ to the distribution $\mathrm{law}(\hat{y}_K)$, which approximates the distribution $\mathrm{law}(\hat{Y}_\epsilon)$.

In the previous section, we identified that early stopping of the backwards process benefits generalisation. In the present section, we will consider how coarseness of the discretisation scheme produces similar benefits. It is often the case that the score function is trained only at those time steps considered by the sampler, i.e. using the time-weighting, $\hat{\tau}_\kappa(dt) = \frac{1}{K} \sum_{k=0}^{K-1} \delta_{T-t_k}(dt)$ (Ho et al., 2020). As a result, the effective stopping time of the algorithm can be much larger than the early stopping time, $\epsilon$. In the following proposition, we demonstrate how this benefits generalisation.

**Proposition 7.** *Consider the setting of Proposition 6 with $\alpha = 1$ and set $\tau = \hat{\tau}_\kappa$, then for sufficiently large $N$, $\kappa \leq \epsilon^{-1}/4$ and any $c \in (0, 1)$, we have that for $q_K = A_{\mathrm{em}} \circ A_{\mathrm{erm}}(S)$,*

$$\mathbb{E}[D(p_\epsilon \| q_K)] \lesssim \mathbb{E}[\hat{\ell}_{\mathrm{sm},\kappa}^\star] + B_\kappa^{\frac{1}{2}} (1 + \kappa)^{-d^*} + \frac{B_\kappa}{C_{\mathrm{sm}}} (1 + \kappa)^{-2d^*} + \kappa(1 + \kappa) d^* \log(\epsilon^{-1})^2 + d e^{-2T},$$

*where $B_\kappa = \frac{C_{\mathrm{sm}}}{c_\nu} (\frac{C_{\mathrm{sm}}}{c_\nu} N^{-2} + \mathbb{E}[\hat{\ell}_{\mathrm{sm},\kappa}^\star])^c \epsilon^{-d^*}$, $\hat{\ell}_{\mathrm{sm},\kappa}^\star := \inf_\mathcal{H} \hat{\ell}_{\mathrm{sm}}(h; S, \hat{\tau}_\kappa)$.*

The second and third terms of the bound in Proposition 7 are due to the score stability of the ERM algorithm and decay as $\kappa$ increases. The fourth term of the bound captures the discretisation error and therefore increases with $\kappa$. What this result captures is that there is a trade-off between sampler accuracy and generalisation that is managed by the discretisation of the diffusion model. In the following corollary, this trade-off is optimised.

**Corollary 8.** *Consider the setting of Proposition 7, then for any $c \in (0, 1)$ and sufficiently small $\epsilon$, there exists $\kappa > 0$ such that with $q_K = A_{\mathrm{em}} \circ A_{\mathrm{erm}}(S)$*

$$\mathbb{E}[D(p_\epsilon \| q_K)] \lesssim \begin{cases} B_\kappa^{\frac{1}{2}} + C_{\mathrm{sm}}^{-1} B_\kappa, & \text{if } B_\kappa \leq \log(\epsilon^{-1})^2, \\ \log(\epsilon^{-1}) B_\kappa^{\frac{1}{2(d^*+1)}} + (C_{\mathrm{sm}}^{-1} + d^*) \log(\epsilon^{-1})^2 B_\kappa^{\frac{1}{d^*+1}} + d e^{-2T}, & \text{otherwise.} \end{cases}$$

The primary strength of this result over (Oko et al., 2023; Azangulov et al., 2024) is that we assume little about the hypothesis class. Their results require carefully constrained network architectures and a specific early stopping time to control complexity. In contrast, our result holds for any sufficiently small early stopping time, relying instead on a carefully chosen discretisation scheme, which is usually tuned in practice (Karras et al., 2022; Williams et al., 2024). The main drawback is that our general approach does not exploit the model class to adapt to smoothness properties of the underlying measure, which we leave for future work.

In Figure 2, we provide a toy experiment that reproduces the implicit regularisation brought about by coarse discretisation and early stopping. Specifically, we track the population KL divergence of a 1-dimensional Gaussian diffusion model guided by an empirical score function evaluated on a small dataset (N=40). We observe distinct U-shaped trajectories across the scaling parameter $\kappa$, the number of discretisation steps, and the early stopping time ($\epsilon$). The pronounced minima in these curves demonstrate that restricting the sampling process promotes generalisation by acting as a form of effective regularisation.

## 6 STOCHASTIC OPTIMISATION AND IMPLICIT REGULARISATION

To learn the score function, it is common to choose it from a parametric hypothesis class $\{s_\theta : \theta \in \mathbb{R}^n\}$ (e.g. a deep neural network) by minimising $\hat{\ell}_{\mathrm{dsm}}$ via stochastic optimisation (Karras et al., 2024). In this section, we consider the score stability of this setting, focusing on stochastic gradient descent

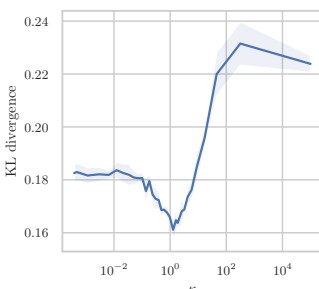 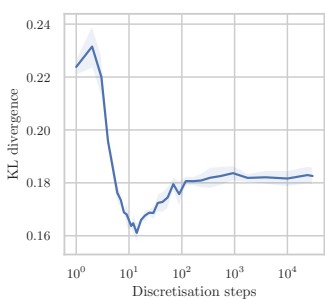 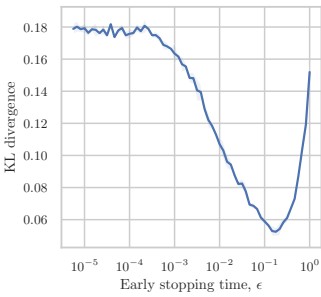

Figure 2: Impact of discretisation and early stopping on population KL divergence for a diffusion model using the empirical score function. Early stopping and coarse discretisation provide a form of implicit regularisation. Evaluated on a 1-dimensional Gaussian target with dataset of size 40.

(SGD) with gradient clipping and weight decay. We consider the standard gradient estimator: given the mini-batch $(x_i')_{i=1}^{N_B}$ of size $N_B \ll N$ we define the random estimator,

$$G(\theta, (x_i')_{i=1}^{N_B}) = \frac{1}{N_B P} \sum_{i=1}^{N_B} \sum_{j=1}^{P} w_{t_{i,j}} \nabla_\theta \| s_\theta(X_{i,j}, t_{i,j}) - \nabla \log p_{t_{i,j}|0}(X_{i,j}|x_i') \|^2, \qquad (13)$$

where we define the random variables $X_{i,j} = \mu_{t_{i,j}} x_i' + \sigma_{t_{i,j}} \xi_{i,j}$, $t_{i,j} \sim w_t^{-1} \tau(dt)$, $\xi_{i,j} \sim N(0, I_d)$. The additional variance introduced by the random variables $\xi_{i,j}$ and $t_{i,j}$ leads to a gradient estimator with significantly higher variance than in standard supervised learning. This presents several challenges during training, and various strategies have been proposed to mitigate this issue (Karras et al., 2024; Song & Kingma, 2021). For example, the weighting function $w : [0, T] \to \mathbb{R}_+$ can be tuned to reduce variance (Karras et al., 2022) or the number of resamples $P \in \mathbb{N}$ can be increased. We consider the following iterative scheme, defined for a given weight decay constant $\lambda > 0$ and clipping value $C > 0$:

$$\theta_{k+1} = (1 - \eta_k \lambda)\theta_k - \eta_k \operatorname{Clip}_C(G_k(\theta_k, (x_i)_{i \in B_k})), \qquad (14)$$

where $\eta_k > 0$ and $B_k \subset [N]$ are the learning rates and mini-batch indices for each iteration $k \in \mathbb{N}$ and we define the clipping operator $\operatorname{Clip}_C(v) = (1 \wedge (C\|v\|^{-1}))v$. Both gradient clipping and weight decay are widely used in diffusion model training and are typically motivated by their stabilising effect on optimisation, minimising the impact of the high variance of the gradient estimator (Song et al., 2021; Ho et al., 2020). Throughout this section, we take the mini-batch $B_k$ to be i.i.d. and uniformly sampled from $[N]$ without replacement. For the sake of simplicity, we suppose that the iterative scheme is terminated after $K \in \mathbb{N}$ iterations, where $K$ is fixed and independent of the data.

## 6.1 STABILITY OF SGD WITH WEIGHT DECAY AND CLIPPING

In our analysis, we avoid restricting the score network to a specific parametric class and instead make structural assumptions based on its smoothness properties. We recall that a function is Lipschitz with constant $L \geq 0$ if it is differentiable and its directional derivatives are uniformly bounded by $L$.

**Assumption 9** (Smoothness of the score network). *There exists $L : \mathbb{R}^d \times (0, T] \to \mathbb{R}_+$ and $M : \mathbb{R}^d \times (0, T] \to \mathbb{R}_+$ such that for almost all $x \in \mathbb{R}^d, t \in (0, T]$, $s_\theta(x, t)$ is Lipschitz and smooth (gradient-Lipschitz) in $\theta \in \mathbb{R}^n$ with constants $L(x, t)$ and $M(x, t)$, respectively. Furthermore, there exists constants $\overline{L}, \overline{M} \geq 0$ such that for any $x \in \operatorname{supp}(\nu_{\mathrm{data}})$,*

$$\int \mathbb{E}[L(X_t, t)^2 | X_0 = x] \, \tau(dt) \leq \overline{L}^2, \quad \int \mathbb{E}[M(X_t, t)^2 | X_0 = x] \, \tau(dt) \leq \overline{M}^2.$$

The use of Lipschitz and smoothness assumptions is commonplace in the analysis of optimisation schemes (Nesterov, 2018; Hardt et al., 2016). However, the assumption differs slightly from the usual in that we only require these properties to hold *almost everywhere* with respect to the input distribution and we allow the Lipschitz and smoothness constants to vary with the input, provided their square averages remain bounded. This relaxation enables us to accommodate common models that would otherwise violate global smoothness assumptions, such as ReLU networks.

**Assumption 10.** *Suppose there exists $B_\ell > 0$ such that for any $\theta \in \mathbb{R}^n$, it holds that*

$$\hat{\ell}_{\mathrm{dsm}}(s_\theta; \{x\}, \delta_t) \leq B_\ell^2 / \sigma_t^4, \qquad \text{for each } x \in \operatorname{supp}(\nu_{\mathrm{data}}), t \in \operatorname{supp}(\tau). \qquad (15)$$

This property requires that the supported score functions are made of denoising functions that are concentrated on a compact set. To highlight that this can be achieved quite easily, we note that with the naive estimate $s(x, t) = -x/\sigma_t^2$, (15) is satisfied with $B_\ell^2 = \mathbb{E}[\|X_0\|^2]$.

In the following proposition we demonstrate score stability bounds in the case that the step size is decaying with a rate of $1/k$.

**Proposition 11.** *Consider the score matching algorithm $A_{\mathrm{sm}} : S \mapsto s_{\theta_K}$ for some fixed $K \in \mathbb{N}$ where $(\theta_k)_k$ is as given in (14). Suppose that assumptions 9 and 10 hold and $\eta_k \leq \bar{\eta}/k$ for all $k < K$, for some $\bar{\eta} \in (0, \lambda^{-1})$. Then, we obtain that $A_{\mathrm{sm}}$ is score stable with constant,*

$$\varepsilon_{stab}^2 \lesssim \left(\frac{C}{\lambda} \vee R\right)^{1 + \frac{\bar{\eta}\upsilon}{\bar{\eta}\upsilon + 1}} \frac{\overline{L}^2}{(\bar{\eta}\upsilon) \vee 1} \left(\frac{C}{\bar{\eta}}\right)^{\frac{1}{\bar{\eta}\upsilon + 1}} \frac{N_B K^{\frac{\bar{\eta}\upsilon}{\bar{\eta}\upsilon + 1}}}{N},$$

*where $R^2 = \mathbb{E}[\|\theta_0\|^2]$, $\upsilon = (\overline{M}B_\ell C_\tau^{1/2} + \overline{L}^2 - \lambda) \vee 0$ and $C_\tau = \int \sigma_t^{-4} \tau(dt)$.*

Since the score matching algorithm is random, to control the stability constant we construct a coupling of the random score functions $A_{\mathrm{sm}}(S)$ and $A_{\mathrm{sm}}(S^i)$ through a coupling of the optimisation trajectories associated with training on $S$ versus $S^i$. The construction of this coupling is such that the trajectories are identical for a large portion of the train-time. The proof also heavily relies on the stochastic mini-batching, hence why we consider the setting of $N_B \ll N$. The proof-technique is a modification of a methodology developed by Hardt et al. (2016). Several recent works have explored the influence of first-order optimisation methods on generalisation (Neu et al., 2021; Pensia et al., 2018; Hardt et al., 2016; Clerico et al., 2022; Dupuis et al., 2025).

## 6.2 UTILISING NOISE IN THE GRADIENT ESTIMATOR

The primary drawback of Proposition 11 is that the bound grows with the number of iterations. This is particularly problematic since diffusion models often require numerous steps due to the high-variance gradient estimator. In this section, we improve this dependence by explicitly leveraging the noise in the gradient estimator. The idea that stochasticity in optimisation can act as a form of implicit regularisation has motivated the development of numerous learning algorithms and theoretical works in recent years (Srivastava et al., 2014; Bishop, 1995; Mou et al., 2018; Pensia et al., 2018). Here, we investigate how the noise intrinsic to the gradient estimator for $\hat{\ell}_{\mathrm{dsm}}$ can play a similar role in promoting generalisation in diffusion models.

To incorporate the effects of the gradient noise, we consider a simplified model in which the noise from the stochastic gradient estimator is approximated with a second-order Gaussian approximation:

$$\theta_{k+1} = (1 - \eta\lambda)\theta_k - \eta\mathbb{E}\big[\mathrm{Clip}_C(G_k)\big|\theta_k, B_k, S\big] + \eta\,\mathrm{Cov}\big(\mathrm{Clip}_C(G_k)\big|\theta_k, B_k, S\big)^{1/2}\xi_k, \quad (16)$$

where $\xi_k \in \mathbb{R}^d$ is a standard Gaussian and we use $G_k := G_k(\theta_k, B_k)$. This approximation can be justified by observing that the inner summation in (13) is over conditionally i.i.d. variables, once conditioned on $\theta, B$ and $S$. Therefore, the gradient estimator $G$ becomes approximately Gaussian as $P$ grows large. For this analysis, we assume the following lower bound on the gradient noise.

**Assumption 12.** *There exists a positive semi-definite matrix $\overline{\Sigma} \in \mathbb{R}^{n \times n}$ such that for any $x \in \mathrm{supp}(\nu)$ and $\theta \in \mathbb{R}^n$,*

$$\mathrm{Cov}_{t \sim \tau, X_t | X_0}\big(\mathrm{Clip}_C(\nabla_\theta\|s_\theta(X_t, t) - x\|^2)\big|X_0 = x\big) \succcurlyeq \overline{\Sigma}.$$

*Furthermore, the eigenvalues of $\overline{\Sigma}$, $(\lambda_i)_{i=1}^n$, possess the spectral gap $\lambda_{gap} := \min_{\lambda_i \neq 0} \lambda_i > 0$.*

We use the matrix $\overline{\Sigma}$ to dictate the geometry on which we perform our analysis. In particular, we consider the weighted norm $\|v\|_{\overline{\Sigma}^+} := v^T\overline{\Sigma}^+ v$ where $\overline{\Sigma}^+$ is the pseudoinverse matrix.

**Assumption 13.** *For almost all $x \in \mathbb{R}^d, t \in (0, T]$, $s_\theta(x, t)$ is Lipschitz and smooth (gradient-Lipschitz) in $\theta \in \mathbb{R}^n$ with respect to the seminorm $\|\cdot\|_{\overline{\Sigma}^+}$ and with constants $L(x, t)$ and $M(x, t)$, respectively. Furthermore, there exists constants $\overline{L}, \overline{M} \geq 0$ such that for any $x \in \mathrm{supp}(\nu_{\mathrm{data}})$,*

$$\int \mathbb{E}[L(X_t, t)^4 | X_0 = x]\,\tau(dt) \leq \overline{L}^4, \quad \int \mathbb{E}[M(X_t, t)^4 | X_0 = x]\,\tau(dt) \leq \overline{M}^4.$$

By requiring that the Lipschitz and smoothness properties hold with respect to $\|\cdot\|_{\overline{\Sigma}^+}$, we effectively require that the gradient estimator adds noise in all directions aside from those that do not change the function (e.g. along symmetries in the parameter space). With this, we arrive at our time-convergent score stability bound for SGD.

**Proposition 14.** *Consider the score matching algorithm $A_{\mathrm{sm}} : S \mapsto s_{\theta_K}$ for some fixed $K \in \mathbb{N}$ where $(\theta_k)_k$ is as given in* (16). *Suppose that assumptions 10, 12 and 13 hold, then there exists some $\bar{\eta} > 0$ such that, if $\sup_k \eta_k \leq \bar{\eta}$, we obtain that $A_{\mathrm{sm}}$ is score stable with constant*

$$\varepsilon_{stab}^2 \lesssim \frac{\overline{L}^2 C}{N} \sqrt{\frac{1}{PN_B^3 \lambda_{gap}}} \exp\left(\frac{\tilde{c}PN_BC^2}{\eta_{\min}\lambda^2}\right) \min\left\{ \sum_{k=0}^{K-1} \eta_k, \exp\left(\frac{\tilde{c}PN_BC^2}{\eta_{\min}\lambda^2}\right) \right\},$$

*for some constant, $\tilde{c}(\overline{M}, \overline{L}, \lambda_{\mathrm{gap}}, \lambda_{\max}(\overline{\Sigma}), B_\ell, C_\tau) > 0$ and $\eta_{\min} = \min_k \eta_k$.*

In this bound, we recover the $\frac{1}{\sqrt{N}}$ score stability bounds from Proposition 11 while also introducing the property that the bound does not grow endlessly with the number of iterations. This property is obtained using the noise in the gradient estimator and is not possible without additional noise. Our proof methodology builds on techniques developed in the literature analysing stochastic gradient Langevin dynamics, a modification of SGD that applies isotropic Gaussian noise at each step. In particular, we draw on the reflection coupling method of Farghly & Rebeschini (2021), which constructs coupled trajectories of the optimisation iterated that contract in expectation under a suitably defined metric (using the technical results of Eberle & Majka (2019); Majka et al. (2020)). In our setting, this contraction arises naturally from the noise inherent to the gradient estimator for the denoising score matching objective. As $P$ increases or $\eta$ decreases, the long-term stability bound increases exponentially fast as a result of the benefit of the noise weakening. Through this analysis, we identify the generalisation benefit of a property unique to diffusion models and how they interact with SGD.

## 7 Conclusion and Future Work

In this paper, we propose a general algorithm-dependent framework for analysing the generalisation capabilities of diffusion models. We introduce *score stability*, which quantifies an algorithm's sensitivity to the dataset, and use it to derive expected generalisation gap bounds. Applying this framework to several common algorithms, we derive closed-form bounds and identify several previously overlooked sources of implicit regularisation in diffusion models. First, our analysis of empirical risk minimisation finds that the denoising score matching objective provides inherent stability guarantees without further regularisation (denoising regularisation). We then analyse how score stability interacts with discrete-time samplers, identifying that coarse discretisation can improve generalisation guarantees (sampler regularisation). Finally, we consider stochastic optimisation schemes for score matching, obtaining stability guarantees (optimisation regularisation).

This work opens up several directions for future work. This includes identifying further relationships between score stability and generalisation by developing high probability bounds, or bounds on notions of memorisation or privacy. The analysis of empirical score matching could be tightened by utilising more properties of the data distribution or model class, such as smoothness. The analysis of sampling could be taken further by considering and comparing different sampling algorithms under the score stability framework (e.g. by considering the probability flow ODE).

## Acknowledgements

Tyler Farghly was supported by Engineering and Physical Sciences Research Council (EPSRC) [grant number EP/T517811/1] and by the DeepMind scholarship. Patrick Rebeschini was funded by UK Research and Innovation (UKRI) under the UK government's Horizon Europe funding guarantee [grant number EP/Y028333/1]. George Deligiannidis acknowledges support from EPSRC [grant number EP/Y018273/1]. The authors would like to thank Michael Hutchinson, Valentin De Bortoli, Peter Potaptchik, Sam Howard, Iskander Azangulov and Christopher J. Williams for valuable comments and stimulating discussions. We would like to give special thanks to Sam Howard for assisting in creating Figure 1.

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

## A    Further background

We begin with some further details on notation and lemmas used throughout this work and provide proofs for the lemmas in Section 2.

### A.1    Random score matching algorithms

We begin with some additional details on how random score matching algorithms are defined in this work. Recalling the probability space $(\Omega, \mathcal{F}, \mathbb{P})$, we define the set of random score functions,

$$\mathcal{S} := \left\{ s : \mathbb{R}^d \times [0, T] \times \Omega : s(\cdot, \cdot, \omega) \in L^0(\mathbb{R}^d \times [0, T]; \mathbb{R}^d) \right\}.$$

For any random score matching algorithm $A_{\mathrm{sm}} : (\cup_{N=1}^\infty (\mathbb{R}^d)^{\otimes N}) \times \Omega \to L^0(\mathbb{R}^d \times [0, T]; \mathbb{R}^d)$, we use $A_{\mathrm{sm}}(S)$ as shorthand for the random score function $(\omega, x, t) \mapsto A_{\mathrm{sm}}(S, \omega)(x, t)$ belonging to $\mathcal{S}$.

Given two random score functions $s, s'$, let $\Gamma(s, s')$ denote the set of all couplings of these functions which we define as,

$$\Gamma(s, s') := \left\{ (\tilde{s}, \tilde{s}') \in \mathcal{S} \times \mathcal{S} : \tilde{s} \simeq s, \tilde{s}' \simeq s' \right\},$$

where $\tilde{s} \simeq s$ denotes the fact that for any bounded measurable test function $\phi : L^0(\mathbb{R}^d \times [0, T]; \mathbb{R}^d) \to \mathbb{R}$, it holds that,

$$\int \phi(s(\cdot, \cdot, \omega)) d\mathbb{P} = \int \phi(\tilde{s}(\cdot, \cdot, \omega)) d\mathbb{P}.$$

### A.2    Preliminary lemmas

For the score matching loss bound, we begin with the fact that the score matching loss is equivalent to the denoising score matching loss up to an added constant Song et al. (2021); Hyvärinen (2005).

**Lemma 15.** *For any $t > 0$, $y \in \mathbb{R}^d$, we have*

$$\nabla \log p_t(y) = \frac{\mu_t \mathbb{E}[X_0 | X_t = y] - y}{\sigma_t^2}, \qquad \nabla \log \hat{p}_t(y) = \frac{\mu_t \mathbb{E}[\hat{X}_0 | \hat{X}_t = y, S] - y}{\sigma_t^2}. \quad (17)$$

*Proof.* We begin by showing that the conditional score is an unbiased estimate of $\nabla \log p_t$. For any $x \in \mathbb{R}^d, t > 0$, we have

$$\mathbb{E}[\nabla \log p_{t|0}(X_t | X_0) | X_t = x] = \int \nabla \log p_{t|0}(x|y) \, p_{0|t}(y|x) dy$$

$$= \int \nabla \log p_{t|0}(x|y) \, \frac{p_{t|0}(x|y) p_0(y)}{p_t(x)} dy$$

$$= \int \nabla p_{t|0}(x|y) \frac{p_0(y)}{p_t(x)} dy.$$

Therefore, using the exchangeability of gradients and integrals (note that $p_{t|0}$ is $C^\infty$), we arrive at

$$\mathbb{E}[\nabla \log p_{t|0}(X_t | X_0) | X_t = x] = \frac{\nabla p_t(x)}{p_t(x)}$$

$$= \nabla \log p_t(x). \quad (18)$$

Alternatively, using (7), we obtain that the left-hand side takes the form,

$$\mathbb{E}[\nabla \log p_{t|0}(X_t | X_0) | X_t = x] = \frac{\mu_t \mathbb{E}[X_0 | X_t = x] - x}{\sigma_t^2},$$

completing the proof of the first equality in (17). For the second equality concerning the empirical score function, the proof follows similarly once the empirical measure $\frac{1}{N} \sum_{i=1}^N \delta_{x_i}$ is used in place of $\nu_{\mathrm{data}}$. $\square$

**Lemma 16.** *For any integrable score function s, it holds that*

$$\ell_{\mathrm{dsm}}(s;\tau) = \ell_{\mathrm{sm}}(s;\tau) + C_{\mathrm{sm}},$$

*where, given $s^\star(x,t) := \nabla \log p_t(x)$, we define*

$$C_{\mathrm{sm}} := \int \frac{\mu_t^2}{\sigma_t^4} \mathbb{E}[\mathrm{Tr}\,\mathrm{Cov}(X_0|X_t)]\tau(dt) = \ell_{\mathrm{dsm}}(s^\star;\tau). \tag{19}$$

*Proof.* Let $s$ be any score function. Using the equality in (18), we obtain the following bias-variance decomposition of $\ell_{\mathrm{dsm}}(s;\tau)$:

$$\ell_{\mathrm{dsm}}(s;\tau) = \int \mathbb{E}\left[\|s(X_t,t) - \nabla \log p_{t|0}(X_t|X_0)\|^2\right]\tau(dt)$$

$$= \int \mathbb{E}\left[\|s(X_t,t) - \nabla \log p_t(X_t)\|^2\right]\tau(dt) + \int \mathbb{E}\left[\|\nabla \log p_{t|0}(X_t|X_0) - \nabla \log p_t(X_t)\|^2\right]\tau(dt)$$

$$= \ell_{\mathrm{sm}}(s;\tau) + \int \mathbb{E}\left[\mathrm{Tr}\,\mathrm{Cov}\left(\nabla \log p_{t|0}(X_t|X_0)\Big|X_t\right)\right]\tau(dt).$$

Once we note that,

$$\mathrm{Tr}\,\mathrm{Cov}\left(\nabla \log p_{t|0}(X_t|X_0)\Big|X_t\right) = \mathrm{Tr}\,\mathrm{Cov}\left(\frac{\mu_t X_0 - x}{\sigma_t^2}\Big|X_t\right)$$

$$= \frac{\mu_t^2}{\sigma_t^4}\mathrm{Tr}\,\mathrm{Cov}(X_0|X_t),$$

we obtain the bound $\ell_{\mathrm{dsm}}(s;\tau) = \ell_{\mathrm{sm}}(s;\tau) + C_{\mathrm{sm}}$ from the statement. To derive the equality $C_{\mathrm{sm}} = \ell_{\mathrm{dsm}}(s^\star;\tau)$, we use that $\ell_{\mathrm{sm}}(s^\star;\tau) = 0$ and so we obtain $\ell_{\mathrm{dsm}}(s^\star;\tau) = 0 + C_{\mathrm{sm}}$. $\qquad\square$

Similarly, there is an equivalence between the empirical forms of the denoising score matching loss and the score matching loss,

$$\hat{\ell}_{\mathrm{dsm}}(s;S,\tau) = \hat{\ell}_{\mathrm{sm}}(s;S,\tau) + \hat{C}_{\mathrm{sm}}, \tag{20}$$

where

$$\hat{C}_{\mathrm{sm}} := \int \frac{\mu_t^2}{\sigma_t^4}\mathbb{E}[\mathrm{Tr}\,\mathrm{Cov}(\hat{X}_0|\hat{X}_t,S)|S]\tau(dt) = \hat{\ell}_{\mathrm{dsm}}(\hat{s}^\star;S,\tau), \tag{21}$$

and $\hat{s}^\star(x,t) = \nabla \log \hat{p}_t(x)$. This follows immediately from the above proof once the empirical measure $\frac{1}{N}\sum_{i=1}^N \delta_{x_i}$ is considered in place of $\nu_{\mathrm{data}}$. With this, we can now prove Lemma 1 from Section 2.

**Lemma 1.** *The objective $\hat{\ell}_{\mathrm{dsm}}(s;S,\tau)$ is identical, up to a constant, to the objective*

$$\hat{\ell}_{\mathrm{sm}}(s;S,\tau) := \int \mathbb{E}[\|s(\hat{X}_t,t) - \nabla \log \hat{p}_t(\hat{X}_t)\|^2|S]\tau(dt), \tag{22}$$

*where $\hat{p}_t$ is the marginal density of $\hat{X}_t$. Therefore, any minimiser of $\hat{\ell}_{\mathrm{dsm}}(\cdot;S,\tau)$ on $L^0(\mathbb{R}^d \times [0,T];\mathbb{R}^d)$ is identical to $\nabla \log \hat{p}_t$ a.e. for any $t \in \mathrm{supp}(\tau)$.*

*Proof.* The proof follows nearly immediately from (20). Since $p_{t|0}$ is $C^\infty$, $\nabla \log p_{t|0}$ is measurable and thus its empirical average $\nabla \log \hat{p}_t$ must be also. Therefore, the score function $s^\star(x,t) = \nabla \log \hat{p}_t(x)$ satisfies $\hat{s}^\star \in L^0(\mathbb{R}^d \times [0,T];\mathbb{R}^d)$ as well as,

$$\hat{\ell}_{\mathrm{sm}}(\hat{s}^\star;S,\tau) = 0.$$

Now let $s \in L^0(\mathbb{R}^d \times [0,T];\mathbb{R}^d)$ be any minimiser of $\hat{\ell}_{\mathrm{dsm}}(\cdot;S,\tau)$. Through the equivalence of $\hat{\ell}_{\mathrm{dsm}}$ and $\hat{\ell}_{\mathrm{sm}}$ up to a constant, it follows that $s$ must also be a minimiser of $\hat{\ell}_{\mathrm{sm}}(\cdot;S,\tau)$ and, due to the existence of $\hat{s}^\star$, must satisfy $\hat{\ell}_{\mathrm{sm}}(s;S,\tau) = 0$ also. Given that whenever $t > 0$, $p_{t|0}$ has full support, it follows that $s(\cdot,t) = s^\star(\cdot,t)$ almost everywhere, $\tau$-almost surely. $\qquad\square$

### A.3 MANIFOLDS

We also introduce some basic properties of smooth manifolds, primarily referencing Aamari et al. (2019). We define the manifold reach and include a known property of this quantity.

**Definition 17.** *The reach of a set $A \subset \mathbb{R}^d$, is defined by $\tau_A = \inf_{p \in A} d(p, Med(A))$, where we define the set,*

$$Med(A) = \left\{ z \in \mathbb{R}^d : \exists p, q \in A \text{ s.t. } p \neq q, \|p - z\| = \|q - z\| \right\}.$$

**Lemma 18.** *Suppose that the measure $\mu$ is supported on a manifold $M$ with reach $\tau_M > 0$ and dimension $d^*$. Then, for any $r \leq \tau_M$, we have*

$$\mu(B_r(x)) \geq \left| \inf_{B_r(x)} p_\mu \right| r^{d^*},$$

*where $p_\mu$ denotes the density of $\mu$ with respect to the volume measure on $M$.*

For the proof of this lemma, we refer to the proof of Proposition 4.3 in Aamari et al. (2019) or Lemma III.23 in Aamari (2017).

## B PROOFS FOR THE GENERALISATION GAP BOUNDS

We now provide the proof of Theorem 3 that bounds the generalisation gap under score stability guarantees.

**Theorem 3.** *Suppose that the score matching algorithm $A_{sm}$ is score stable with constant $\varepsilon_{stab}$. Then, with $\hat{s} = A_{sm}(S)$, it holds that*

$$\left| \mathbb{E}\left[\ell_{dsm}(\hat{s}; \tau)\right]^{1/2} - \mathbb{E}\left[\hat{\ell}_{dsm}(\hat{s}; S, \tau)\right]^{1/2} \right| \leq \varepsilon_{stab}. \tag{23}$$

*Furthermore, it holds that*

$$\mathbb{E}\left[\ell_{sm}(\hat{s}; \tau)\right] - \mathbb{E}\left[\hat{\ell}_{sm}(\hat{s}; S, \tau)\right] \leq 2\varepsilon_{stab} \mathbb{E}\left[\hat{\ell}_{dsm}(\hat{s}; S, \tau)\right]^{1/2} + \varepsilon_{stab}^2. \tag{24}$$

*Proof.* Setting $\hat{s} = A_{sm}(S)$ and $\hat{s}^i = A_{sm}(S^i)$, we use the property that $(\hat{s}, \tilde{x})$ and $(\hat{s}^i, x_i)$ are distributed identically to obtain that,

$$\mathbb{E}[\ell_{dsm}(\hat{s}; \tau)] = \mathbb{E}[\hat{\ell}_{dsm}(\hat{s}; \{\tilde{x}\}, \tau)]$$

$$= \mathbb{E}\left[\frac{1}{N} \sum_{i=1}^{N} \hat{\ell}_{dsm}(\hat{s}^i; \{x_i\}, \tau)\right]$$

$$= \mathbb{E}\left[\frac{1}{N} \sum_{i=1}^{N} \int \mathbb{E}_{X_t}[\|\hat{s}^i(X_t, t) - \nabla \log p_{t|0}(X_t|x_i)\|^2 | X_0 = x_i, S] \, \tau(dt)\right].$$

Therefore, it follows from the triangle inequality in $L^2$-norm that,

$$\left| \mathbb{E}[\ell_{dsm}(\hat{s}; \tau)]^{1/2} - \mathbb{E}[\hat{\ell}_{dsm}(\hat{s}; S, \tau)]^{1/2} \right| \leq \mathbb{E}\left[\frac{1}{N} \sum_{i=1}^{N} \int \mathbb{E}[\|\hat{s}(X_t, t) - \hat{s}^i(X_t, t)\|^2 | X_0 = x_i, S] \, \tau(dt)\right]^{1/2}$$

Note that if the algorithm $A_{sm}$ is stochastic, the right-hand side would hold regardless of how $\hat{s}|S, \tilde{x}$ and $\hat{s}^i|S, \tilde{x}$ were coupled. Therefore, the most efficient coupling can be chosen, leading to the bound,

$$\left| \mathbb{E}[\ell_{dsm}(\hat{s}; \tau)]^{1/2} - \mathbb{E}[\hat{\ell}_{dsm}(\hat{s}; S)]^{1/2} \right| \tag{25}$$

$$\leq \mathbb{E}\left[\inf_{(\hat{s}, \hat{s}^i) \in \Gamma_i} \frac{1}{N} \sum_{i=1}^{N} \int \mathbb{E}[\|\hat{s}(X_t, t) - \hat{s}^i(X_t, t)\|^2 | X_0 = x_i, S] \, \tau(dt)\right]^{1/2}$$

$$\leq \varepsilon_{stab}, \tag{26}$$

completing the proof of the bound in (23).

To obtain the bound in (24), we use Lemma 16 to derive

$$\mathbb{E}[\ell_{\mathrm{sm}}(\hat{s};\tau)] = \mathbb{E}[\hat{\ell}_{\mathrm{sm}}(\hat{s};S,\tau)] + \mathbb{E}[\ell_{\mathrm{dsm}}(\hat{s};\tau) - \hat{\ell}_{\mathrm{dsm}}(\hat{s};S,\tau)] + \mathbb{E}\big[\hat{\ell}_{\mathrm{dsm}}(\nabla \log \hat{p}_t; S, \tau)\big]$$
$$- \ell_{\mathrm{dsm}}(\nabla \log p_t; \tau). \tag{27}$$

Since $\hat{\ell}_{\mathrm{dsm}}(\cdot; S, \tau)$ is a unbiased estimator of $\ell_{\mathrm{sm}}(\cdot; \tau)$, we have that

$$\ell_{\mathrm{dsm}}(\nabla \log p_t; \tau) = \mathbb{E}[\hat{\ell}_{\mathrm{dsm}}(\nabla \log p_t; S, \tau)] \geq \mathbb{E}[\hat{\ell}_{\mathrm{dsm}}(\nabla \log \hat{p}_t; S, \tau)], \tag{28}$$

where the inequality follows from the fact that $\nabla \log \hat{p}_t$ minimises $\hat{\ell}_{\mathrm{dsm}}$. Furthermore, using (26), we deduce the bound,

$$|\mathbb{E}[\ell_{\mathrm{dsm}}(\hat{s};\tau) - \hat{\ell}_{\mathrm{dsm}}(\hat{s};S,\tau)]|$$
$$= \Big(\mathbb{E}[\ell_{\mathrm{dsm}}(\hat{s};\tau)]^{1/2} + \mathbb{E}[\hat{\ell}_{\mathrm{dsm}}(\hat{s};S,\tau)]^{1/2}\Big)\Big|\mathbb{E}[\ell_{\mathrm{dsm}}(\hat{s};S,\tau)]^{1/2} - \mathbb{E}[\hat{\ell}_{\mathrm{dsm}}(\hat{s};S,\tau)]^{1/2}\Big|$$
$$\leq \Big(2\mathbb{E}[\hat{\ell}_{\mathrm{dsm}}(\hat{s};S,\tau)]^{1/2} + \varepsilon_{\mathrm{stab}}\Big)\varepsilon_{\mathrm{stab}}$$
$$= 2\varepsilon_{\mathrm{stab}}\mathbb{E}[\hat{\ell}_{\mathrm{dsm}}(\hat{s};S,\tau)]^{1/2} + \varepsilon_{\mathrm{stab}}^2. \tag{29}$$

Thus, substituting (28) and (29) into (27) recovers the bound in (24) in the statement. □

We obtain upper bounds relying on the fact that the constant separating the score matching loss from the denoising score matching loss is larger on average in the empirical case. One could obtain lower bounds through our techniques, but this would require an analysis of the rate of convergence of this constant, which is beyond the scope of this paper.

## C PROOFS FOR STABILITY OF EMPIRICAL DENOISING SCORE MATCHING

In this section, we provide the proof for Theorem 3, where the algorithm that minimises $\hat{\ell}_{\mathrm{dsm}}(\cdot; S, \tau)$ over some class of score functions $\mathcal{H}$ is shown to be score stable.

### C.1 ON-AVERAGE STABILITY OF THE ERM ALGORITHM

We begin with an important lemma that shows that under minimal assumptions, $\hat{s} = A_{\mathrm{erm}}(S)$ and $\hat{s}^i = A_{\mathrm{erm}}(S^i)$ are close in $L^2$ space, averaged over the full dataset. The first half of this proof utilises the fact that $\hat{\ell}_{\mathrm{dsm}}$ is strongly convex in a weighted $L^2$ space, exploiting a well-known relationship between strong-convexity and algorithmic stability (e.g. see (Bousquet & Elisseeff, 2002; Charles & Papailiopoulos, 2018; Vary et al., 2024; Attia & Koren, 2022)).

**Lemma 19.** *Suppose that $A_{\mathrm{erm}}$ is score stable with constant $\varepsilon_{stab}$, then for any $i \in [N]$, we obtain,*

$$\mathbb{E}\left[\int\int \|\hat{s}^i(y,t) - \hat{s}(y,t)\|^2 \hat{p}_t(dy)\,\tau(dt)\right] \leq 8\mathbb{E}[\hat{\ell}_{\mathrm{sm}}(\hat{s})] + \frac{8}{N}\varepsilon_{stab}(C_{\mathrm{sm}}^{1/2} + \varepsilon_{stab}) \tag{30}$$

*where $\hat{s} = A_{\mathrm{erm}}(S), \hat{s}^i = A_{\mathrm{erm}}(S^i)$.*

*Proof.* Choose $i \in [N]$ and let $\hat{s} = A_{\mathrm{erm}}(S), \hat{s}^i = A_{\mathrm{erm}}(S^i)$ so that $\hat{s} \in \operatorname{argmin}_{\mathcal{H}} \hat{\ell}_{\mathrm{dsm}}(\cdot; S, \tau), \hat{s}^i \in \operatorname{argmin}_{\mathcal{H}} \hat{\ell}_{\mathrm{dsm}}(\cdot; S^i, \tau)$. The proof begins with the following simple expression, which holds for all $j \in [N]$:

$$2\int \big\langle \hat{s}^i(y,t) - \hat{s}(y,t), \hat{s}^i(y,t) - \nabla \log p_{t|0}(y|x_j)\big\rangle\, p_{t|0}(dy|x_j)$$
$$= \int \|\hat{s}^i(y,t) - \nabla \log p_{t|0}(y|x_j)\|^2\, p_{t|0}(dy|x_j) - \int \|\hat{s}(y,t) - \nabla \log p_{t|0}(y|x_j)\|^2\, p_{t|0}(dy|x_j)$$
$$+ \int \|\hat{s}^i(y,t) - \hat{s}(y,t)\|^2\, p_{t|0}(dy|x_j).$$

By averaging over $j \in [N]$ and integrating with respect to $\tau$, we arrive at the upper bound,

$$
\frac{2}{N} \sum_{j \in [N]} \int \int \langle \hat{s}^i(y,t) - \hat{s}(y,t), \hat{s}^i(y,t) - \nabla \log p_{t|0}(y|x_j) \rangle \, p_{t|0}(dy|x_j) \, \tau(dt)
$$

$$
= \hat{\ell}_{\mathrm{dsm}}(\hat{s}^i; S, \tau) - \hat{\ell}_{\mathrm{dsm}}(\hat{s}; S, \tau) + \int \int \|\hat{s}^i(y,t) - \hat{s}(y,t)\|^2 \, \hat{p}_t(dy) \, \tau(dt)
$$

$$
\geq \int \int \|\hat{s}^i(y,t) - \hat{s}(y,t)\|^2 \, \hat{p}_t(dy) \, \tau(dt), \tag{31}
$$

where the inequality follows from the fact that $\hat{\ell}_{\mathrm{dsm}}(\hat{s}; S, \tau) \leq \hat{\ell}_{\mathrm{dsm}}(s; S, \tau)$ for any score function $s \in \mathcal{H}$. Additionally, the left-hand side is upper bounded using the Cauchy-Schwarz inequality to obtain,

$$
\frac{2}{N} \sum_{x \in S} \int \int \langle \hat{s}^i(y,t) - \hat{s}(y,t), \hat{s}^i(y,t) - \nabla \log p_{t|0}(y|x) \rangle \, p_{t|0}(dy|x) \, \tau(dt)
$$

$$
= \frac{2}{N} \sum_{x \in S^i} \int \int \langle \hat{s}^i(y,t) - \hat{s}(y,t), \hat{s}^i(y,t) - \nabla \log p_{t|0}(y|x) \rangle \, p_{t|0}(dy|x) \, \tau(dt)
$$

$$
+ \frac{2}{N} \int \int \langle \hat{s}^i(y,t) - \hat{s}(y,t), \hat{s}^i(y,t) - \nabla \log p_{t|0}(y|x_i) \rangle \, p_{t|0}(dy|x_i) \, \tau(dt)
$$

$$
- \frac{2}{N} \int \int \langle \hat{s}^i(y,t) - \hat{s}(y,t), \hat{s}^i(y,t) - \nabla \log p_{t|0}(y|\tilde{x}) \rangle \, p_{t|0}(dy|\tilde{x}) \, \tau(dt)
$$

$$
\leq 2\hat{\ell}_{\mathrm{sm}}(\hat{s}^i; S^i, \tau)^{1/2} \left( \int \int \|\hat{s}^i(y,t) - \hat{s}(y,t)\|^2 \, \hat{p}_t^i(dy) \, \tau(dt) \right)^{1/2}
$$

$$
+ \frac{2}{N} \hat{\ell}_{\mathrm{dsm}}(\hat{s}^i; \{x_i\}, \tau)^{1/2} \left( \int \int \|\hat{s}^i(y,t) - \hat{s}(y,t)\|^2 \, p_{t|0}(dy|x_i) \, \tau(dt) \right)^{1/2}
$$

$$
+ \frac{2}{N} \hat{\ell}_{\mathrm{dsm}}(\hat{s}^i; \{\tilde{x}\}, \tau)^{1/2} \left( \int \int \|\hat{s}^i(y,t) - \hat{s}(y,t)\|^2 \, p_{t|0}(dy|\tilde{x}) \, \tau(dt) \right)^{1/2}, \tag{32}
$$

where $\hat{p}_t^i(dy) = \frac{1}{N} \sum_{x \in S^i} p_{t|0}(dy|x)$. Combining the expressions in (31) and (32) and taking the expectation, we derive the bound,

$$
\mathbb{E} \left[ \int \int \|\hat{s}^i(y,t) - \hat{s}(y,t)\|^2 \, \hat{p}_t(dy) \, \tau(dt) \right]
$$

$$
\leq 2\mathbb{E}[\hat{\ell}_{\mathrm{sm}}(\hat{s}^i; S^i, \tau)]^{1/2} \mathbb{E}\left[ \int \int \|\hat{s}^i(y,t) - \hat{s}(y,t)\|^2 \, \hat{p}_t^i(dy) \, \tau(dt) \right]^{1/2}
$$

$$
+ \frac{2}{N} \mathbb{E}[\hat{\ell}_{\mathrm{dsm}}(\hat{s}^i; \{x_i\}, \tau)]^{1/2} \mathbb{E}\left[ \int \int \|\hat{s}^i(y,t) - \hat{s}(y,t)\|^2 \, p_{t|0}(dy|x_i) \, \tau(dt) \right]^{1/2}
$$

$$
+ \frac{2}{N} \mathbb{E}[\hat{\ell}_{\mathrm{dsm}}(\hat{s}^i; \{\tilde{x}\}, \tau)]^{1/2} \mathbb{E}\left[ \int \int \|\hat{s}^i(y,t) - \hat{s}(y,t)\|^2 \, p_{t|0}(dy|\tilde{x}) \, \tau(dt) \right]^{1/2}
$$

$$
\leq 2\mathbb{E}[\hat{\ell}_{\mathrm{sm}}(\hat{s}; S, \tau)]^{1/2} \mathbb{E}\left[ \int \int \|\hat{s}^i(y,t) - \hat{s}(y,t)\|^2 \, \hat{p}_t(dy) \, \tau(dt) \right]^{1/2}
$$

$$
+ \frac{2}{N} \varepsilon_{\mathrm{stab}} \left( \mathbb{E}[\hat{\ell}_{\mathrm{dsm}}(\hat{s}; S, \tau)]^{1/2} + \mathbb{E}[\ell_{\mathrm{dsm}}(\hat{s}; \tau)]^{1/2} \right),
$$

where we recall that $\varepsilon_{\mathrm{stab}}$ is the stability constant for $A_{\mathrm{erm}}$. Here, we have used the fact that $(\hat{s}, S)$ has the same law as $(\hat{s}^i, S^i)$ and also $\mathbb{E}[\hat{\ell}_{\mathrm{dsm}}(\hat{s}^i; \{\tilde{x}\}, \tau)] = \mathbb{E}[\hat{\ell}_{\mathrm{dsm}}(\hat{s}; S, \tau)]$ and $\mathbb{E}[\hat{\ell}_{\mathrm{dsm}}(\hat{s}^i; \{x_i\}, \tau)] =$

$\mathbb{E}[\ell_{\mathrm{dsm}}(\hat{s}; \tau)]$. By solving the quadratic equation, we deduce that the above inequality implies,

$$\mathbb{E}\left[\int\int \|\hat{s}^i(y, t) - \hat{s}(y, t)\|^2 \, \hat{p}_t(dy)\, \tau(dt)\right]$$

$$\leq \left(\mathbb{E}[\hat{\ell}_{\mathrm{sm}}(\hat{s}; S, \tau)]^{\frac{1}{2}} + \sqrt{\mathbb{E}[\hat{\ell}_{\mathrm{sm}}(\hat{s}; S, \tau)] + \tfrac{2}{N}\varepsilon_{\mathrm{stab}}(\mathbb{E}[\hat{\ell}_{\mathrm{dsm}}(\hat{s}; \tau)]^{\frac{1}{2}} + \mathbb{E}[\hat{\ell}_{\mathrm{dsm}}(\hat{s}; S, \tau)]^{1/2})}\right)^2$$

$$\leq 4\mathbb{E}[\hat{\ell}_{\mathrm{sm}}(\hat{s}; S, \tau)] + \frac{4}{N}\varepsilon_{\mathrm{stab}}(\mathbb{E}[\hat{\ell}_{\mathrm{dsm}}(\hat{s}; \tau)]^{\frac{1}{2}} + \mathbb{E}[\hat{\ell}_{\mathrm{dsm}}(\hat{s}; S, \tau)]^{\frac{1}{2}}).$$

We simplify the above expression further using Theorem 3. Using the stability assumption, it follows from (23) that $\mathbb{E}[\ell_{\mathrm{dsm}}(\hat{s})]^{1/2} \leq \mathbb{E}[\hat{\ell}_{\mathrm{dsm}}(\hat{s})]^{1/2} + \varepsilon_{\mathrm{stab}}$. Furthermore, from Lemma 16, we have

$$\mathbb{E}[\hat{\ell}_{\mathrm{dsm}}(\hat{s})] = \mathbb{E}[\hat{\ell}_{\mathrm{sm}}(\hat{s})] + \mathbb{E}[\hat{C}_{\mathrm{sm}}]$$
$$\leq \mathbb{E}[\hat{\ell}_{\mathrm{sm}}(\hat{s})] + C_{\mathrm{sm}},$$

where we recall the definitions of $\hat{C}_{\mathrm{sm}}$ and $C_{\mathrm{sm}}$ from (21) and (19) and recall that $\mathbb{E}[\hat{C}_{\mathrm{sm}}] \leq C_{\mathrm{sm}}$ from (28). Thus, from Young's inequality, we obtain the bound

$$\mathbb{E}\left[\int\int \|\hat{s}^i(y, t) - \hat{s}(y, t)\|^2 \, \hat{p}_t(dy)\, \tau(dt)\right]$$

$$\leq 4\mathbb{E}[\hat{\ell}_{\mathrm{sm}}(\hat{s})] + \frac{4}{N}\varepsilon_{\mathrm{stab}}(2\mathbb{E}[\hat{\ell}_{\mathrm{sm}}(\hat{s})]^{1/2} + 2C_{\mathrm{sm}}^{1/2} + \varepsilon_{\mathrm{stab}})$$

$$\leq 8\mathbb{E}[\hat{\ell}_{\mathrm{sm}}(\hat{s})] + \frac{4}{N}\varepsilon_{\mathrm{stab}}(\varepsilon_{\mathrm{stab}}/N + 2C_{\mathrm{sm}}^{1/2} + \varepsilon_{\mathrm{stab}})$$

$$\leq 8\mathbb{E}[\hat{\ell}_{\mathrm{sm}}(\hat{s})] + \frac{8}{N}\varepsilon_{\mathrm{stab}}(C_{\mathrm{sm}}^{1/2} + \varepsilon_{\mathrm{stab}}).$$

$\square$

## C.2  PROOF OF PROPOSITION 6

To obtain the stability bound in Proposition 6, we convert the result in Lemma 19, which is a bound in $L^2(\hat{p}_t)$, to the bound in $L^2(p_{t|0}(\cdot|\tilde{x}))$ required of score stability. For this, we rely on two further lemmas, the first of which is a fundamental property of the Ornstein-Uhlenbeck process, captured by the Harnack inequality of Wang (1997) (see Theorem 5.6.1 in (Bakry et al., 2014)).

**Lemma 20** (Wang's Harnack inequality). *For each positive measurable function $\phi : \mathbb{R}^d \to \mathbb{R}$, every $t > 0, p > 1$ and every $x, y \in \mathbb{R}^d$, it holds that,*

$$\mathbb{E}[\phi(X_t)|X_0 = x] \leq \mathbb{E}[\phi(X_t)^p|X_0 = y]^{1/p} \exp\left(\frac{\mu_t^2\|x - y\|^2}{2(p-1)\sigma_t^2}\right).$$

This result describes the stability of the diffusion semigroup under changes in its initial position and shows that as $t$ grows, the distribution of $X_t$ depends less on $X_0$. The second lemma, for which we provide a proof, controls the empirical measure,

$$\hat{\nu}(dx) = \frac{1}{N}\sum_{i=1}^{N}\delta_{x_i}(dx),$$

on balls around training examples.

**Lemma 21.** *Suppose that Assumption 4 is satisfied, then for any $i \in [N]$, $r \in (0, \tau_{reach}]$ and any decreasing function $\phi : (0, \infty) \to \mathbb{R}_+$, we have the bound*

$$\mathbb{E}\left[\phi\Big(\hat{\nu}(B_r(x_i))\Big)\right] \leq \phi(N^{-1})\exp\big(-c_\nu N r^{d^*}/16\big) + \phi\big(c_\nu r^{d^*}/2\big),$$

*whenever $N \geq (4c_\nu^{-1}r^{-d^*}) \vee 2$, where $c_\nu = \inf p_\nu$.*

*Proof.* We rewrite the object $\hat{\nu}(B_r(x_i))$ as an empirical average of Bernoulli random variables,

$$\hat{\nu}(B_r(x_i)) = \frac{1}{N}\sum_{j=1}^{N}\mathbb{1}_{x_j\in B_r(x_i)} = \frac{1}{N} + \frac{1}{N}\sum_{j\neq i}\mathbb{1}_{x_j\in B_r(x_i)}.$$

When conditioned on $x_i$, the random variables $(\mathbb{1}_{x_j\in B_r(x_i)})_{j\neq i}$ are independently and identically distributed Bernoulli random variables with probability $\mu = \nu_{\text{data}}(B_r(x_i))$. To utilise concentration of the empirical process, we first rewrite the probability

$$\mathbb{P}\Big(\hat{\nu}(B_r(x_i)) \leq \mu/2 \Big| x_i\Big) = \mathbb{P}\Big(S_{N-1} \leq N\mu/2 - 1 \Big| x_i\Big),$$

where $S_{N-1} = \sum_{j\neq i}\mathbb{1}_{x_j\in B_r(x_i)}$. Since $\mu \leq 1$, we have $N\mu/2 - 1 \leq (N-1)\mu/2$, and hence

$$\mathbb{P}\Big(\hat{\nu}(B_r(x_i)) \leq \mu/2 \Big| x_i\Big) \leq \mathbb{P}\Big(S_{N-1} \leq (N-1)\mu/2 \Big| x_i\Big).$$

Therefore, by the multiplicative Chernoff bound for binomial random variables (e.g. Lemma 2.2 in Wainwright (2019)), we obtain,

$$\mathbb{P}\Big(S_{N-1} \leq (N-1)\mu/2 \Big| x_i\Big) \leq \exp\Big(-(N-1)\mu/8\Big)$$
$$\leq \exp\Big(-N\mu/16\Big),$$

for all $N \geq 2$.

Next, using the trivial bound $\hat{\nu}(B_r(x_i)) \geq N^{-1}$ and the fact that $\phi$ is decreasing, we have

$$\mathbb{E}\Big[\phi\Big(\hat{\nu}(B_r(x_i))\Big)\Big|x_i\Big] = \mathbb{E}\Big[\phi\Big(\hat{\nu}(B_r(x_i))\Big)\mathbb{1}_{\{\hat{\nu}(B_r(x_i))>\mu/2\}}\Big|x_i\Big]$$
$$+ \mathbb{E}\Big[\phi\Big(\hat{\nu}(B_r(x_i))\Big)\mathbb{1}_{\{\hat{\nu}(B_r(x_i))\leq\mu/2\}}\Big|x_i\Big]$$
$$\leq \phi(\mu/2) + \phi(N^{-1})\mathbb{P}\Big(\hat{\nu}(B_r(x_i)) \leq \mu/2\Big|x_i\Big)$$
$$\leq \phi(\mu/2) + \phi(N^{-1})\exp\Big(-N\mu/16\Big).$$

To control $\mu$, we use Lemma 18 which asserts that $\mu \geq c_\nu r^{d^*}$. Substituting this into the above bound and using again that $\phi$ is decreasing, we obtain

$$\mathbb{E}\Big[\phi\Big(\hat{\nu}(B_r(x_i))\Big)\Big|x_i\Big] \leq \phi\big(c_\nu r^{d^*}/2\big) + \phi(N^{-1})\exp\big(-c_\nu Nr^{d^*}/16\big).$$

Taking the expectation with respect to $x_i$ yields the desired bound. $\qquad\square$

This now brings us to the proof of the proposition, which we first restate.

**Proposition 6.** *Suppose that assumptions 4 and 5 hold and that $\epsilon := \inf\text{supp}(\tau) \in (0, \tau_{reach}^2)$, then for any $c \in (0,1)$ and sufficiently large $N$, the score matching algorithm $A_{erm}$ is score stable with,*

$$\varepsilon_{stab}^2 \lesssim C\big(CC_{\text{sm}}N^{-2} + \mathbb{E}[\hat{\ell}_{\text{sm}}(\hat{s})]\big)^c, \qquad C = \frac{D_{\mathcal{H}}^2}{\sigma_\epsilon^4} \vee \frac{1}{c_\nu\sigma_\epsilon^{d^*}}.$$

*Proof.* We use the shorthand $\hat{\ell}_{\text{sm}}(s) = \hat{\ell}_{\text{sm}}(s; S, \tau), \hat{\ell}_{\text{dsm}}(s) = \hat{\ell}_{\text{dsm}}(s; S, \tau), \ell_{\text{sm}}(s) = \ell_{\text{sm}}(s; \tau)$ for the sake of brevity. We start from Lemma 19 which provides a bound on the difference between $\hat{s}^i$ and $\hat{s}$ in $L^2(\hat{p}_t)$. We use it to develop a bound in $L^2(\hat{p}_{t|0}(\cdot|\tilde{x}))$, as required by score stability. In particular, we define the quantity,

$$\varepsilon^2 = \mathbb{E}\Big[\int\int \|\hat{s}^i(y,t) - \hat{s}(y,t)\|^2\, p_{t|0}(dy|x_i)\,\tau(dt)\Big],$$

so that, due to the fact that $A_{\text{erm}}$ is invariant with respect to dataset permutations, it must be score stable with constant $\varepsilon$ (we have that $\varepsilon < \infty$ from Assumption 5). Therefore, from Lemma 19, we have

$$\mathbb{E}\Big[\int\int \|\hat{s}^i(y,t) - \hat{s}(y,t)\|^2\, \hat{p}_t(dy)\,\tau(dt)\Big] \leq 8\mathbb{E}[\hat{\ell}_{\text{sm}}(\hat{s})] + \frac{8}{N}\varepsilon(C_{\text{sm}}^{1/2} + \varepsilon).$$

We proceed using Lemma 20 with $\phi(y) = \|\hat{s}^i(y,t) - \hat{s}(y,t)\|^2$ to obtain that for any $j \in [N]$, $p > 1$,

$$\int \|\hat{s}^i(y,t) - \hat{s}(y,t)\|^2 \, p_{t|0}(dy|x_i)$$
$$\leq \left( \int \|\hat{s}^i(y,t) - \hat{s}(y,t)\|^{2p} \, p_{t|0}(dy|x_j) \right)^{1/p} \exp\left( \frac{\mu_t^2 \|x_i - x_j\|^2}{2(p-1)\sigma_t^2} \right).$$

Setting $B = B_R(x_i) \cap S$, we can average over the above bound to obtain,

$$\int \|\hat{s}^i(y,t) - \hat{s}(y,t)\|^2 \, p_{t|0}(dy|x_i)$$
$$\leq \frac{1}{|B|} \sum_{x \in B} \left( \int \|\hat{s}^i(y,t) - \hat{s}(y,t)\|^{2p} \, p_{t|0}(dy|x) \right)^{1/p} \exp\left( \frac{\mu_t^2 R^2}{2(p-1)\sigma_t^2} \right)$$
$$\leq \left( \frac{1}{|B|} \sum_{x \in B} \int \|\hat{s}^i(y,t) - \hat{s}(y,t)\|^{2p} \, p_{t|0}(dy|x) \right)^{1/p} \exp\left( \frac{\mu_t^2 R^2}{2(p-1)\sigma_t^2} \right).$$

We can further bound this using the $L^\infty$ bound in Assumption 5:

$$\int \|\hat{s}^i(y,t) - \hat{s}(y,t)\|^2 \, p_{t|0}(dy|x_i)$$
$$\leq \hat{\nu}(B)^{-1/p} \left( \int \|\hat{s}^i(y,t) - \hat{s}(y,t)\|^{2p} \, \hat{p}_t(dy) \right)^{1/p} \exp\left( \frac{\mu_t^2 R^2}{2(p-1)\sigma_t^2} \right)$$
$$\leq (D_{\mathcal{H}}/\sigma_t^2)^{2(1-1/p)} \hat{\nu}(B)^{-1/p} \left( \int \|\hat{s}^i(y,t) - \hat{s}(y,t)\|^2 \, \hat{p}_t(dy) \right)^{\frac{1}{p}} \exp\left( \frac{\mu_t^2 R^2}{2(p-1)\sigma_t^2} \right).$$

Integrating with respect to $\tau$ and taking the expectation, we obtain,

$$\varepsilon^2 \leq (D_{\mathcal{H}}/\sigma_\epsilon^2)^{2/q} \mathbb{E}\left[ \hat{\nu}(B)^{-q/p} \right]^{1/q} \mathbb{E}\left[ \int \int \|\hat{s}^i(y,t) - \hat{s}(y,t)\|^2 \, \hat{p}_t(dy) \tau(dt) \right]^{1/p} \exp\left( \frac{\mu_\epsilon^2 R^2}{2(p-1)\sigma_\epsilon^2} \right)$$
$$\leq (D_{\mathcal{H}}^2/\sigma_\epsilon^4) \mathbb{E}\left[ \hat{\nu}(B)^{-q/p} \right]^{1/q} \left( 8\mathbb{E}[\hat{\ell}_{\mathrm{sm}}(\hat{s})] + \frac{8}{N}\varepsilon(C_{\mathrm{sm}}^{1/2} + \varepsilon) \right)^{1/p} \exp\left( \frac{\mu_\epsilon^2 R^2}{2(p-1)\sigma_\epsilon^2} \right),$$

where we define $q := (1 - 1/p)^{-1} > 1$ and assume that $\epsilon$ is sufficiently small, so that $\sigma_\epsilon^2 \leq D_{\mathcal{H}}$. Using Young's inequality, it follows that for any $\lambda > 0$,

$$\varepsilon^2 \leq \frac{D_{\mathcal{H}}^2}{\sigma_\epsilon^4 \lambda^q q} \mathbb{E}\left[ \hat{\nu}(B)^{-q/p} \right] \exp\left( \frac{q\mu_\epsilon^2 R^2}{2(p-1)\sigma_\epsilon^2} \right) + \frac{\lambda^p}{p}\left( 8\mathbb{E}[\hat{\ell}_{\mathrm{sm}}(\hat{s})] + \frac{8}{N}\varepsilon(C_{\mathrm{sm}}^{1/2} + \varepsilon) \right).$$

Setting $\kappa := 8\lambda^p/pN$, we can rearrange this to obtain the quadratic inequality,

$$(1 - \kappa)\varepsilon^2 - C_{\mathrm{sm}}^{1/2}\kappa\varepsilon \leq \left( \frac{8}{Np\kappa} \right)^{q/p} \frac{D_{\mathcal{H}}^2}{\sigma_\epsilon^4 q} \mathbb{E}\left[ \hat{\nu}(B)^{-q/p} \right] \exp\left( \frac{q\mu_\epsilon^2 R^2}{2(p-1)\sigma_\epsilon^2} \right) + N\kappa\mathbb{E}[\hat{\ell}_{\mathrm{sm}}(\hat{s})].$$

Requiring that $\kappa \leq 1/2$, we solve the quadratic to obtain the inequality,

$$\frac{\varepsilon^2}{4} \leq C_{\mathrm{sm}}\kappa^2 + \left( \frac{8}{Np\kappa} \right)^{q/p} \frac{D_{\mathcal{H}}^2}{\sigma_\epsilon^4 q} \mathbb{E}\left[ \hat{\nu}(B)^{-q/p} \right] \exp\left( \frac{q\mu_\epsilon^2 R^2}{2(p-1)\sigma_\epsilon^2} \right) + N\kappa\mathbb{E}[\hat{\ell}_{\mathrm{sm}}(\hat{s})]. \qquad (33)$$

Next, we optimise $R$ by setting $R = \sigma_\epsilon$ and apply Lemma 21 with $\phi(r) = r^{-q/p}$ to obtain that whenever $\sigma_\epsilon \leq \tau_{\mathrm{reach}}$ we obtain,

$$\mathbb{E}\left[ \hat{\nu}(B)^{-q/p} \right] \leq N^{q/p} \exp\left( -\frac{c_\nu N \sigma_\epsilon^{d^*}}{16} \right) + \left( \frac{2}{c_\nu \sigma_\epsilon^{d^*}} \right)^{q/p}$$
$$\leq 2\left( \frac{2}{c_\nu \sigma_\epsilon^{d^*}} \right)^{q/p},$$

for all $N$ sufficiently large (depending only on $c_\nu$, $\sigma_\ell$, $p$ and $q$). Returning to (33), it follows from the above that

$$\frac{\varepsilon^2}{4} \leq C_{\text{sm}}\kappa^2 + \left(\frac{16}{Npc_\nu\sigma_\epsilon^{d^*}\kappa}\right)^{q/p}\frac{2D_\mathcal{H}^2}{\sigma_\epsilon^4 q}\exp\left(\frac{2q}{p-1}\right) + N\kappa\mathbb{E}[\hat{\ell}_{\text{sm}}(\hat{s})]. \tag{34}$$

We now choose $\kappa$ by optimising the second and third terms of this bound, by which we arrive at the choice,

$$\kappa^q = \frac{2D_\mathcal{H}^2}{\sigma_\epsilon^4 pN\gamma}\exp\left(\frac{2q}{p-1}\right)\left(\frac{16}{Npc_\nu\sigma_\epsilon^{d^*}}\right)^{q/p},$$

for some $\gamma > 0$. For any set of constants, we can take $N$ sufficiently large, to ensure that $\kappa \leq 1/2$. Substituting this into (34), we arrive at the bound,

$$\frac{\varepsilon^2}{4} \leq C_{\text{sm}}(Np)^{-2}\left(\frac{2D_\mathcal{H}^2}{\sigma_\epsilon^4}\right)^{2/q}\exp\left(\frac{4}{p-1}\right)\left(\frac{16}{c_\nu\sigma_\epsilon^{d^*}}\right)^{2/p}\gamma^{-2/q}$$
$$+ \left(\frac{2D_\mathcal{H}^2}{\sigma_\epsilon^4}\right)^{1/q}\exp\left(\frac{2}{p-1}\right)\left(\frac{16}{c_\nu\sigma_\epsilon^{d^*}}\right)^{1/p}\left(\frac{\gamma^{1/p}}{q} + \frac{1}{p\gamma^{1/q}}\mathbb{E}[\hat{\ell}_{\text{sm}}(\hat{s})]\right).$$

Applying Young's inequality to the three terms on the right-hand side, we may choose $\gamma > 0$ so that,

$$\frac{\varepsilon^2}{4} \leq \left(\frac{2D_\mathcal{H}^2}{\sigma_\epsilon^4}\right)^{\frac{1}{q}}\exp\left(\frac{2}{p-1}\right)\left(\frac{16}{c_\nu\sigma_\epsilon^{d^*}}\right)^{\frac{1}{p}}\left(C_{\text{sm}}N^{-2}\left(\frac{2D_\mathcal{H}^2}{\sigma_\epsilon^4}\right)^{\frac{1}{q}}\exp\left(\frac{2}{p-1}\right)\left(\frac{16}{c_\nu\sigma_\epsilon^{d^*}}\right)^{\frac{1}{p}} + \mathbb{E}[\hat{\ell}_{\text{sm}}(\hat{s})]\right)^{\frac{1}{p}}$$
$$\leq \left(\frac{2D_\mathcal{H}^2}{\sigma_\epsilon^4} \vee \frac{16}{c_\nu\sigma_\epsilon^{d^*}}\right)\exp\left(\frac{4}{p-1}\right)\left(\left(\frac{2D_\mathcal{H}^2}{\sigma_\epsilon^4} \vee \frac{16}{c_\nu\sigma_\epsilon^{d^*}}\right)C_{\text{sm}}N^{-2} + \mathbb{E}[\hat{\ell}_{\text{sm}}(\hat{s})]\right)^{\frac{1}{p}}.$$

Optimising $p$, we obtain,

$$\frac{\varepsilon^2}{4} \lesssim \left(\frac{2D_\mathcal{H}^2}{\sigma_\epsilon^4} \vee \frac{16}{c_\nu\sigma_\epsilon^{d^*}}\right)\exp\left(\frac{5}{2\sqrt{2}}\log(\alpha^{-1})^{1/2} - 2\right)\alpha,$$

where,

$$\alpha = \left(\frac{2D_\mathcal{H}^2}{\sigma_\epsilon^4} \vee \frac{16}{c_\nu\sigma_\epsilon^{d^*}}\right)C_{\text{sm}}N^{-2} + \mathbb{E}[\hat{\ell}_{\text{sm}}(\hat{s})],$$

from which the bound in the statement follows. $\qquad\square$

## D  PROOFS FOR SAMPLING AND SCORE STABILITY

In this section, we provide details for the discretisation scheme considered in Section 5 and give the proof for Proposition 7 and Corollary 8. In the work of Potaptchik et al. (2024), they consider the following discretisation scheme, based on the scheme of (Benton et al., 2024):

$$\hat{y}_{k+1} = \mu_{t_{k+1}-t_k}^{-1}\hat{y}_k + \frac{\sigma_{t_{k+1}-t_k}^2}{\mu_{t_{k+1}-t_k}}s(\hat{y}_k, T-t_k) + \sigma_{t_{k+1}-t_k}\frac{\sigma_{T-t_{k+1}}}{\sigma_{T-t_k}}\zeta_k, \qquad k \in \{0,...,K-1\},$$

where $\zeta_k \sim N(0, I_d)$ and we recall that the timesteps $(t_k)_{k=0}^K$ are given by,

$$t_k = \begin{cases} \kappa k, & \text{if } k < \frac{T-1}{\kappa}, \\ T - (1+\kappa)^{\frac{T-1}{\kappa}-k}, & \text{if } \frac{T-1}{\kappa} \leq k \leq K, \end{cases}$$

where $L = \frac{T-1}{\kappa} > 0$, $K = \lfloor L + \log(\epsilon^{-1})/\log(1+\kappa) \rfloor$ and $\kappa > 0, T \geq 1$ is chosen freely. We recall the following result from Potaptchik et al. (2024).

**Lemma 22.** *Suppose that $\alpha = 1$ and Assumption 4 holds with* $\text{diam supp}(\nu_{\text{data}}) \leq 1$. *Then, it holds that,*

$$D(p_\epsilon\|A_{\text{em}}(s)) \lesssim \ell_{\text{sm}}(s;\hat{\tau}) + D(p_T\|p_\infty) + \Delta_{\kappa,K},$$
$$\Delta_{\kappa,K} = \kappa + d^*\kappa^2(K-L)(\log(\epsilon^{-1}) + \sup|\log(p_\nu)|),$$

*where we define the measure,*

$$\hat{\tau}(dt) = \frac{1}{K}\sum_{k=0}^{K-1}\delta_{T-t_k}(dt).$$

### D.1 COARSE DISCRETISATION AND REGULARISATION

Fix $\epsilon > 0$ and suppose that $\kappa$ is such that $\log(\epsilon^{-1})/\log(1+\kappa)$ is an integer. Set $K = L + \log(\epsilon^{-1})/\log(1+\kappa)$ so that, according to the discretisation scheme,

$$t_K = T - (1+\kappa)^{-\log(\epsilon^{-1})/\log(1+\kappa)} = T - \epsilon.$$

*Proof of Proposition 7.* Let $\hat{s} = A_{\mathrm{erm}}(S)$. We begin with Lemma 22, which provides the bound,

$$\mathbb{E}[D(p_\epsilon \| A_{\mathrm{em}}(\hat{s}))] \lesssim \mathbb{E}[\ell_{\mathrm{sm}}(\hat{s}; S, \hat{\tau})] + D(p_T \| p_\infty) + \Delta_{\kappa, K}.$$

For $\epsilon$ sufficiently small we have the bound,

$$\Delta_{\kappa, K} = \kappa + d^* \kappa^2 \frac{\log(\epsilon^{-1})}{\log(1+\kappa)}(\log(\epsilon^{-1}) + \sup|\log(p_\nu)|)$$

$$\lesssim \kappa(1+\kappa)d^* \log(\epsilon^{-1})^2.$$

Using Theorem 3, we obtain that if the algorithm is $\varepsilon_{\mathrm{stab}}$-score stable, we have

$$\mathbb{E}[\ell_{\mathrm{sm}}(\hat{s}; \hat{\tau})] \lesssim \mathbb{E}[\hat{\ell}_{\mathrm{sm}}(\hat{s}; S, \hat{\tau})] + \varepsilon_{\mathrm{stab}}\mathbb{E}[\hat{\ell}_{\mathrm{dsm}}(\hat{s}; S, \hat{\tau})]^{1/2} + \varepsilon_{\mathrm{stab}}^2$$

$$\lesssim \mathbb{E}[\hat{\ell}_{\mathrm{sm}}(\hat{s}; \hat{\tau})] + \varepsilon_{\mathrm{stab}}C_{\mathrm{sm}}^{1/2} + \varepsilon_{\mathrm{stab}}^2$$

Using Proposition 6 we obtain that with $\tau = \hat{\tau}$, $A_{\mathrm{erm}}$ is score stable, with constant,

$$\varepsilon_{\mathrm{stab}}^2 \lesssim C\left(CC_{\mathrm{sm}}N^{-2} + \mathbb{E}[\hat{\ell}_{\mathrm{sm}}(\hat{s})]\right)^c$$

$$\lesssim c_\nu^{-1}\sigma_{T-t_{K-1}}^{-d^*}\left(c_\nu^{-1}\sigma_{T-t_{K-1}}^{-d^*}C_{\mathrm{sm}}N^{-2} + \mathbb{E}[\hat{\ell}_{\mathrm{sm}}(\hat{s})]\right)^c.$$

Now by definition, we have that

$$T - t_{K-1} = (1+\kappa)^{L-K+1} = \epsilon(1+\kappa),$$

so if we take $\epsilon, \kappa$ sufficiently small so that $\epsilon(1+\kappa) \leq \frac{1}{2}$, we also have $\sigma_{\epsilon(1+\kappa)}^2 \geq \epsilon(1+\kappa)$ and thus we obtain,

$$\varepsilon_{\mathrm{stab}}^2 \lesssim c_\nu^{-1}\epsilon^{-d^*/2}(1+\kappa)^{-d^*/2}\left(c_\nu^{-1}\epsilon^{-d^*/2}(1+\kappa)^{-d^*/2}C_{\mathrm{sm}}N^{-2} + \mathbb{E}[\hat{\ell}_{\mathrm{sm}}(\hat{s})]\right)^c.$$

$$\lesssim c_\nu^{-1}\epsilon^{-d^*}(1+\kappa)^{-d^*}\left(c_\nu^{-1}C_{\mathrm{sm}}N^{-2} + \mathbb{E}[\hat{\ell}_{\mathrm{sm}}(\hat{s})]\right)^c,$$

where in the last inequality, we use that $\epsilon(1+\kappa) \leq 1/2$. $\qquad\square$

We now proceed by proving Corollary 8 in which the bound in Proposition 7 is optimised.

*Proof of Corollary 8.* Let $\tilde{\tau}_\epsilon$ denote the weak limit of the measure $\tau_\kappa$ as $\kappa \to 0^+$. Since $\mathrm{supp}(\tilde{\tau}_\epsilon) \subseteq [\epsilon, T]$ and $\epsilon > 0$, we know that $\inf_{\mathcal{H}} \ell_{\mathrm{sm}}(\cdot; S, \tilde{\tau}_\epsilon) < \infty$. From this, we deduce that $\lim_{\kappa \to 0^+} B_\kappa < \infty$.

With this there exists $\kappa^* \geq 1$ which is the smallest quantity satisfying,

$$(1+\kappa^*)^{2d^*+2} = \frac{B_{\kappa^*}}{\log(\epsilon^{-1})^2} \vee 1.$$

In the case that $B_{\kappa^*} > \log(\epsilon^{-1})$, we have that

$$B_{\kappa^*}^{1/2}(1+\kappa^*)^{-d^*} + \frac{B_{\kappa^*}}{C_{\mathrm{sm}}}(1+\kappa^*)^{-2d^*} + \kappa^*(1+\kappa^*)d^*\log(\epsilon^{-1})^2$$

$$= B_{\kappa^*}^{\frac{1}{2(d^*+1)}}\log(\epsilon^{-1})^{\frac{d^*}{d^*+1}} + (C_{\mathrm{sm}}^{-1} + d^*)B_{\kappa^*}^{\frac{1}{d^*+1}}$$

$$\leq B_{\kappa^*}^{\frac{1}{2(d^*+1)}}\log(\epsilon^{-1}) + (C_{\mathrm{sm}}^{-1} + d^*)B_{\kappa^*}^{\frac{1}{d^*+1}}\log(\epsilon^{-1})^2.$$

Plus, if $B_{\kappa^*} \leq \log(\epsilon^{-1})$ and therefore $\kappa^* = 1$, then there exists $\kappa$ such that,

$$B_\kappa^{1/2}(1+\kappa)^{-d^*} + \frac{B_\kappa}{C_{\mathrm{sm}}}(1+\kappa)^{-2d^*} + \kappa(1+\kappa)d^*\log(\epsilon^{-1})^2 \lesssim B_\kappa^{1/2} + \frac{B_\kappa}{C_{\mathrm{sm}}} + de^{-T}.$$

Combining these leads to the bound in the statement. $\qquad\square$

## E    PROOFS FOR STABILITY OF SGD

In this section, we analyse the stochastic optimisation scheme in (14), deriving the score stability bounds given in Proposition 11. We begin with a basic lemma that follows from weight decay and gradient clipping.

**Lemma 23.** *Suppose that $\eta_k < \lambda^{-1}$ for all $k \in \mathbb{N}$, then for any $K \in \mathbb{N}$, it holds that*

$$\|\theta_K\| \leq \frac{Ce}{\lambda} \vee \|\theta_0\|.$$

*Proof.* We begin with the bound,

$$\|\theta_{k+1}\| \leq (1 - \eta_k\lambda)\|\theta_k\| + \eta_k\|\mathrm{Clip}_C(G_k(\theta_k, \{x_i\}_{i \in B_k}))\|$$
$$\leq (1 - \eta_k\lambda)\|\theta_k\| + \eta_k C.$$

By comparison, this leads to the bound

$$\|\theta_k\| \leq C \sum_{k=0}^{K-1} \eta_k \prod_{i=k+1}^{K-1} (1 - \eta_k\lambda) + \prod_{k=0}^{K-1} (1 - \eta_k\lambda)\|\theta_0\|$$

$$\leq C \sum_{k=0}^{K-1} \eta_k \exp\left(\lambda \sum_{i=0}^{k} \eta_k\right) + \exp\left(-\lambda \sum_{i=0}^{K-1} \eta_k\right)\|\theta_0\|$$

$$\leq C \exp\left(-\lambda \sum_{i=0}^{K-1} \eta_k + \lambda \max_k \eta_k\right) \sum_{k=0}^{K-1} \eta_k \exp\left(\lambda \sum_{i=0}^{k-1} \eta_k\right) + \exp\left(-\lambda \sum_{i=0}^{K-1} \eta_k\right)\|\theta_0\|$$

Since the sum forms a left Riemann sum, approximating an integral of an increasing function, we can upper bound it by the integral over $\exp(\lambda t)$. Furthermore, we have that $\lambda \max_k \eta_k \leq 1$, which leads to the bound,

$$\|\theta_k\| \leq Ce \exp\left(-\lambda \sum_{i=0}^{K-1} \eta_k\right) \int_0^{\sum_{k=0}^{K-1} \eta_k} \exp(\lambda t) dt + \exp\left(-\lambda \sum_{i=0}^{K-1} \eta_k\right)\|\theta_0\|$$

$$\leq \frac{Ce}{\lambda}\left(1 - \exp\left(-\lambda \sum_{k=0}^{K-1} \eta_k\right)\right) + \exp\left(-\lambda \sum_{k=0}^{K-1} \eta_k\right)\|\theta_0\|$$

$$\leq \frac{Ce}{\lambda} \vee \|\theta_0\|.$$

$\square$

We are now ready to prove Proposition 11.

**Proposition 11.** *Consider the score matching algorithm $A_{\mathrm{sm}} : S \mapsto s_{\theta_K}$ for some fixed $K \in \mathbb{N}$ where $(\theta_k)_k$ is as given in (14). Suppose that assumptions 9 and 10 hold and $\eta_k \leq \bar{\eta}/k$ for all $k < K$, for some $\bar{\eta} \in (0, \lambda^{-1})$. Then, we obtain that $A_{\mathrm{sm}}$ is score stable with constant,*

$$\varepsilon_{stab}^2 \lesssim \left(\frac{C}{\lambda} \vee R\right)^{1 + \frac{\bar{\eta}\upsilon}{\bar{\eta}\upsilon + 1}} \frac{\overline{L}^2}{(\bar{\eta}\upsilon) \vee 1}\left(\frac{C}{\bar{\eta}}\right)^{\frac{1}{\bar{\eta}\upsilon + 1}} \frac{N_B K^{\frac{\bar{\eta}\upsilon}{\bar{\eta}\upsilon + 1}}}{N},$$

*where $R^2 = \mathbb{E}[\|\theta_0\|^2]$, $\upsilon = (\overline{M}B_\ell C_\tau^{1/2} + \overline{L}^2 - \lambda) \vee 0$ and $C_\tau = \int \sigma_t^{-4} \tau(dt)$.*

*Proof.* Since the stochastic mini-batch scheme, and therefore the resulting score matching algorithm, is symmetric to dataset permutations, we consider stability under changes in the $N^{th}$ entry of the dataset, without loss of generality. Let $\theta_k$ be the process given in (14), using the dataset $S$ and let $\tilde{\theta}_k$ be the same process using $S^N$ instead of $S$:

$$\tilde{\theta}_{k+1} = (1 - \eta\lambda)\tilde{\theta}_k - \eta_k \mathrm{Clip}_C(G_k(\tilde{\theta}_p, \{\tilde{x}_i\}_{i \in B_k})), \qquad \tilde{\theta}_0 = \theta_0,$$

where $\tilde{x}_i = x_i$ for $i \neq N$, $\tilde{x}_N = \tilde{x}$. By having the processes share the same mini-batch indices $B_k$ and gradient approximation $G_k$ (i.e. sharing the same random time variables $t_{i,j}$ and noise $\xi_{i,j}$), we couple the processes $\theta_k$ and $\tilde{\theta}_k$.

We proceed by first controlling the stability of the gradient estimator, computing the bound,

$$\|G_k(\theta_k, (x_i)_{i \in B_k}) - G_k(\tilde{\theta}_k, (x_i)_{i \in B_k})\|$$

$$\leq \frac{1}{N_B P} \sum_{i \in B_k} \sum_{j=1}^{P} w_{t_{i,j}} \|\nabla s_{\theta_k}(X_{i,j}, t_{i,j})^T (s_{\theta_k}(X_{i,j}, t_{i,j}) - \nabla \log p_{t_{i,j}|0}(X_{t_{i,j}}|x_i))$$

$$- \nabla s_{\tilde{\theta}_k}(X_{t_{i,j}}, t_{i,j})^T (s_{\tilde{\theta}_k}(X_{t_{i,j}}, t_{i,j}) - \nabla \log p_{t_{i,j}|0}(X_{t_{i,j}}|x_i)))\|$$

$$\leq \frac{1}{N_P P} \sum_{i \in B_k} \sum_{j=1}^{P} w_{t_{i,j}} \Big( \|\nabla s_{\theta_k}(X_{t_{i,j}}, t_{i,j}) - \nabla s_{\tilde{\theta}_k}(X_{t_{i,j}}, t_{i,j})\| \|s_{\theta_k}(X_{t_{i,j}}, t_{i,j})$$

$$- \nabla \log p_{t_{i,j}|0}(X_{t_{i,j}}|x_i)\| + \|\nabla s_{\tilde{\theta}_k}(X_{t_{i,j}}, t_{i,j})\| \|s_{\theta_k}(X_{t_{i,j}}, t_{i,j}) - s_{\tilde{\theta}_k}(X_{t_{i,j}}, t_{i,j})\| \Big)$$

$$\leq \frac{1}{N_P P} \sum_{i \in B_k} \sum_{j=1}^{P} w_{t_{i,j}} \Big( M(X_{t_{i,j}}, t_{i,j}) \|s_{\theta_k}(X_{t_{i,j}}, t_{i,j}) - \nabla \log p_{t_{i,j}|0}(X_{t_{i,j}}|x_j)\|$$

$$+ L(X_{t_{i,j}}, t_{i,j})^2 \Big) \|\theta_k - \tilde{\theta}_k\|.$$

We control the expectation of this by first noting that,

$$\mathbb{E}\Big[ w_{t_{i,j}} \Big( M(X_{t_{i,j}}, t_{i,j}) \|s_{\theta_k}(X_{t_{i,j}}, t_{i,j}) - \nabla \log p_{t_{i,j}|0}(X_{t_{i,j}}|x_j)\| + L(X_{t_{i,j}}, t_{i,j})^2 \Big) \Big| \theta_k, \tilde{\theta}_k, S, \tilde{x} \Big]$$

$$\leq \left( \int \mathbb{E}[M(X_t, t)^2 | X_0 = x_i] \tau(dt) \right)^{1/2} \left( \int \hat{\ell}_{\mathrm{dsm}}(s_{\theta_k}; \{x_i\}, \delta_t) \tau(dt) \right)^{1/2}$$

$$+ \int \mathbb{E}[L(X_t, t)^2 | X_0 = x_i] \tau(dt)$$

$$\leq \overline{M} B_\ell C_\tau^{1/2} + \overline{L}^2,$$

where we define the quantity $C_\tau := \int \sigma_t^{-4} \tau(dt)$. From this, it follows that

$$\mathbb{E}\Big[ \|G_k(\theta_k, (x_i)_{i \in B_k}) - G_k(\tilde{\theta}_k, (x_i)_{i \in B_k})\| \Big| \theta_k, \tilde{\theta}_k, S, \tilde{x} \Big] \leq \left( \overline{M} B_\ell C_\tau^{1/2} + \overline{L}^2 \right) \|\theta_k - \tilde{\theta}_k\|.$$

Furthermore, we can control the difference between $G_k(\tilde{\theta}_k, (x_i)_{i \in B_k})$ and $G_k(\tilde{\theta}_k, (\tilde{x}_i)_{i \in B_k})$, using the fact that they are identical whenever $N \notin B_k$. Thus, obtaining,

$$\mathbb{E}\Big[ \|\mathrm{Clip}_C(G(\theta_k, (x_i)_{i \in B_k})) - \mathrm{Clip}_C(G(\tilde{\theta}_k, (\tilde{x}_i)_{i \in B_k}))\| \Big| \theta_k, \tilde{\theta}_k, S, \tilde{x} \Big]$$

$$\leq \mathbb{E}\Big[ \|G(\theta_k, (x_i)_{i \in B_k}) - G(\tilde{\theta}_k, (x_i)_{i \in B_k})\| \Big| \theta_k, \tilde{\theta}_k, S, \tilde{x} \Big]$$

$$+ \mathbb{E}\Big[ \|\mathrm{Clip}_C(G(\tilde{\theta}_k, (x_i)_{i \in B_k})) - \mathrm{Clip}_C(G(\tilde{\theta}_k, (\tilde{x}_i)_{i \in B_k}))\| \Big| \theta_k, \tilde{\theta}_k, S, \tilde{x} \Big]$$

$$\leq \left( \overline{M} B_\ell C_\tau^{1/2} + \overline{L}^2 \right) \|\theta_k - \tilde{\theta}_k\| + 2C \frac{N_B}{N},$$

where we have used the fact that $\mathbb{P}(N \in B_k) = \frac{N_B}{N}$. Thus, using (14), we obtain that for any $k_0 \leq k$,

$$\mathbb{E}\Big[ \|\theta_{k+1} - \tilde{\theta}_{k+1}\| \Big| \theta_{k_0}, \tilde{\theta}_{k_0}, S, \tilde{x} \Big]$$

$$\leq \left( 1 + \eta_k \left( \overline{M} B_\ell C_\tau^{1/2} + \overline{L}^2 - \lambda \right) \right) \mathbb{E}\Big[ \|\theta_k - \tilde{\theta}_k\| \Big| \theta_{k_0}, \tilde{\theta}_{k_0}, S, \tilde{x} \Big] + 2\eta_k C \frac{N_B}{N}.$$

$$\leq (1 + \eta_k \upsilon) \mathbb{E}\Big[ \|\theta_k - \tilde{\theta}_k\| \Big| \theta_{k_0}, \tilde{\theta}_{k_0}, S, \tilde{x} \Big] + 2\eta_k C \frac{N_B}{N},$$

where $\upsilon = \overline{M} B_\ell C_\tau^{1/2} + \overline{L}^2 - \lambda$. Thus, by comparison, we obtain,

$$\mathbb{E}\Big[ \|\theta_K - \tilde{\theta}_K\| \Big| \theta_{k_0}, \tilde{\theta}_{k_0}, S, \tilde{x} \Big] \leq \sum_{i=k_0}^{K-1} 2\eta_i C \frac{N_B}{N} \prod_{j=i+1}^{K-1} (1 + \eta_j \upsilon) + \|\theta_{k_0} - \tilde{\theta}_{k_0}\| \prod_{j=k_0}^{K-1} (1 + \eta_j \upsilon).$$

From this we obtain the following:

$$\mathbb{E}[\|\theta_K - \tilde{\theta}_K\| | \theta_{k_0} = \tilde{\theta}_{k_0}, S, \tilde{x}] \leq 2C \frac{N_B}{N} \sum_{i=k_0}^{K-1} \eta_i \exp\left( \sum_{j=i+1}^{K-1} \eta_j v \right)$$

$$\leq \frac{2CN_B\bar{\eta}}{N} \sum_{i=k_0}^{K-1} \frac{1}{i} \left( \frac{K}{i} \right)^{\bar{\eta}v}$$

$$\lesssim \frac{CN_B}{Nv} \left( \frac{K}{k_0} \right)^{\bar{\eta}v},$$

where we use the fact that $\sum_{j=i+1}^{K-1} \frac{1}{j} \leq \log(K) - \log(i)$. By the law of total probability, we have

$$\mathbb{E}[\|\theta_K - \tilde{\theta}_K\| | \theta_0]$$

$$= \mathbb{E}[\|\theta_K - \tilde{\theta}_K\| | \theta_{k_0} = \tilde{\theta}_{k_0}]\mathbb{P}(\theta_{k_0} = \tilde{\theta}_{k_0}|\theta_0) + \mathbb{E}[\|\theta_K - \tilde{\theta}_K\| | \theta_{k_0} \neq \tilde{\theta}_{k_0}, \theta_0]\mathbb{P}(\theta_{k_0} \neq \tilde{\theta}_{k_0}|\theta_0)$$

$$\lesssim \frac{CN_B}{Nv} \left( \frac{K}{k_0} \right)^{\bar{\eta}v} + \left( \frac{Ce}{\lambda} \vee \|\theta_0\| \right) \frac{k_0 N_B}{N},$$

where in the second inequality, we use Lemma 23. Thus, optimising $k_0$ leads to the bound,

$$\mathbb{E}[\|\theta_K - \tilde{\theta}_K\| | \theta_0] \lesssim \left( \frac{C}{c} \right)^{\frac{1}{v+1}} (1 + 1/cv) \left( \frac{Ce}{\lambda} \vee \|\theta_0\| \right)^{\frac{cv}{cv+1}} \frac{N_B}{N} K^{\frac{cv}{cv+1}}.$$

Finally, we obtain score stability using the fact that

$$\int \mathbb{E}[\|s_{\theta_K}(X_t, t) - s_{\tilde{\theta}_K}(X_t, t)\|^2 | X_0 = \tilde{x}, S] \tau(dt)$$

$$\leq \mathbb{E}\left[ \bar{L}^2 \|\theta_K - \tilde{\theta}_K\|^2 \right]$$

$$\leq 2\mathbb{E}\left[ \bar{L}^2 \left( \frac{Ce}{\lambda} \vee \|\theta_0\| \right) \|\theta_K - \tilde{\theta}_K\| \right]$$

$$\lesssim \bar{L}^2 \left( \frac{Ce}{\lambda} \vee R \right)^{1 + \frac{cv}{cv+1}} \left( \frac{C}{c} \right)^{\frac{1}{cv+1}} (1 + 1/cv) \frac{N_B}{N} K^{\frac{cv}{cv+1}},$$

where $R^2 = \mathbb{E}\|\theta_0\|^2$. $\qquad \square$

## F   WASSERSTEIN CONTRACTIONS

In this section, we derive the Wasserstein contraction result used in the proof of Proposition 14. We begin with the more abstract problem of deriving Wasserstein contractions for a discrete time diffusion process with anisotropic non-constant volatility. We consider stochastic processes given by the discrete-time update,

$$x_{k+1} = (1 - \eta\lambda)x_k + \eta b(x_k) + \sqrt{\eta}\sigma(x_k)\xi_k, \tag{35}$$

$$y_{k+1} = (1 - \eta\lambda)y_k + \eta\tilde{b}(y_k) + \sqrt{\eta}\tilde{\sigma}(y_k)\xi_k, \tag{36}$$

for some $b, \tilde{b} : \mathbb{R}^n \to \mathbb{R}^n, \sigma, \tilde{\sigma} : \mathbb{R}^n \to \mathbb{R}^{n \times n}$ where $\xi_k \sim N(0, I_n)$, and we show that the laws of $x_k$ and $y_k$ contract in Wasserstein distance. We borrow the strategy developed by Eberle (2016) and extended in (Eberle & Majka, 2019; Majka et al., 2020), constructing a coupling and a metric for which exponential contractions of the coupling can be obtained. However, these works are restricted to the setting of isotropic noise with constant volatility (i.e. $\sigma(x) = cI_n$) and so some careful modification to the strategy is required. In particular, we analyse this process with respect to the seminorm $\|\cdot\|_{G^+}$ given by $\|x\|_{G^+}^2 = x^T G^+ x$, where $G^+$ denotes the Moore-Penrose pseudoinverse of the matrix $G$. Furthermore, we allow for $x_k$ and $y_k$ to have different bias and volatility terms and so controlling for this will also require some modifications to the proof technique.

To define our coupling we first suppose that there exists a symmetric positive semi-definite matrix $G \in \mathbb{R}^{n \times n}$ such that $\sigma(x)^2, \tilde{\sigma}(y)^2 \succcurlyeq G$ for all $x \in \mathbb{R}^n$. We define the residual volatility,

$$\Sigma_G(x) := (\sigma(x)^2 - G)^{1/2}, \qquad \tilde{\Sigma}_G(y) := (\tilde{\sigma}(y)^2 - G)^{1/2},$$

which are well-defined as real PSD square roots. Since $Z'$ and $Z$ are independent, the noise $\sqrt{\eta}\,\Sigma_G(x)\,Z' + \sqrt{\eta}\,G^{1/2}Z$ has covariance $\eta(\Sigma_G(x)^2 + G) = \eta\,\sigma(x)^2$, recovering the correct law. We construct the coupling in the *matched* setting where both processes use the same drift $b$ and volatility $\Sigma_G$; the mismatched case will be handled by a perturbation argument in Section F.2.4. Starting from $x, y \in \mathbb{R}^n$, we define the deterministic drift step and the residual noise step,

$$\tilde{x} = (1 - \eta\lambda)x + \eta b(x), \qquad \tilde{y} = (1 - \eta\lambda)y + \eta b(y),$$
$$\hat{x} = \tilde{x} + \sqrt{\eta}\,\Sigma_G(x)\,Z', \qquad \hat{y} = \tilde{y} + \sqrt{\eta}\,\Sigma_G(y)\,Z',$$

where $Z' \sim N(0, I_n)$. We then define the *synchronous coupled* processes,

$$X' = \hat{x} + \sqrt{\eta}G^{1/2}Z$$
$$Y'_s = \hat{y} + \sqrt{\eta}G^{1/2}Z,$$

with $Z \sim N(0, I_n)$. We also consider the reflection coupling,

$$Y'_r = \hat{y} + \sqrt{\eta}G^{1/2}\Big(I - 2(G^{1/2})^+ ee^T(G^{1/2})^+\Big)Z, \qquad \text{with } e = (\hat{x} - \hat{y})/\|\hat{x} - \hat{y}\|_{G^+} \qquad (37)$$

which has the noise act in the mirrored direction. We combine these couplings to arrive at the final coupling $(X', Y')$:

$$Y' = \begin{cases} X', & \text{if } \zeta \leq \phi_{\hat{y}, \eta G}(X')/\phi_{\hat{x}, \eta G}(X'),\, |\langle e, Z \rangle|^2 < m^2/\eta \text{ and } \hat{r} \leq r_1 \\ Y'_r, & \text{if } \zeta > \phi_{\hat{y}, \eta G}(X')/\phi_{\hat{x}, \eta G}(X'),\, |\langle e, Z \rangle|^2 < m^2/\eta \text{ and } \hat{r} \leq r_1 \qquad (38) \\ Y'_s, & \text{otherwise,} \end{cases}$$

for some fixed $m > 0$.

We assume the following regularity properties.

**Assumption 24.** *Suppose that $b$ is bounded, satisfying $B := \sup_{x \in \mathbb{R}^n} \|b(x)\|_{G^+} < \infty$ and we have the Lipschitz property, $\|b(x) - b(y)\|_{G^+} \leq L_b\|x - y\|_{G^+}$ and $\|\Sigma_G(x) - \Sigma_G(y)\|_{op, G^+} \leq L_\sigma\|x - y\|_{G^+}$ for all $x, y \in \mathbb{R}^n$ and for some $L_b, L_\sigma \geq 0$.*

We also allow for $b \neq \tilde{b}$ and $\sigma \neq \tilde{\sigma}$, making the following assumption.

**Assumption 25.** *Suppose that $b, \tilde{b}$ satisfy $\|b(x) - \tilde{b}(x)\|_{G^+} \leq \tilde{B}_b$, $\|\Sigma_G(x) - \tilde{\Sigma}_G(x)\|_{op, G^+} \leq \tilde{B}_\sigma$ for all $x \in \mathbb{R}^n$ and for some $\tilde{B}_b, \tilde{B}_\sigma \geq 0$.*

We define the objects,

$$R = \|x - y\|_{G^+}, \qquad \tilde{r} = \|\tilde{x} - \tilde{y}\|_{G^+}, \qquad \hat{r} = \|\hat{x} - \hat{y}\|_{G^+}, \qquad R' = \|X' - Y'\|_{G^+}.$$

We wish to show that $R'$ contracts in expectation, i.e. it is less than $R$ on average. We modify the metric to guarantee this is possible. We define the function,

$$f(r) = \begin{cases} \frac{1}{a}(1 - e^{-ar}), & \text{if } r \leq r_2, \\ \frac{1}{a}(1 - e^{-ar_2}) + \frac{1}{2r_2}e^{-ar_2}(r^2 - r_2^2), & \text{otherwise,} \end{cases}$$

where $a = 6L_b r_1/c_0$, $r_1 = 4(1 + \eta_0 L_b)B/\lambda$, $r_2 = r_1 + \sqrt{\eta_0}$ and $c_0, \eta_0$ are defined below. The coupling and the strategy for proving contractions is closely based on an analysis in Majka et al. (2020) and for the sake of comparison, we rely on similar notation. We will also heavily borrow properties of the function $f$ that are proven in their work.

By allowing $\Sigma_G$ to be non-constant (i.e. when $\sigma$ is non-constant), we run in to additional complications that are controlled by making the following assumption about the scale of $L_\sigma$.

**Assumption 26.** *Suppose that the following three inequalities hold:*

$$n - 1, (\lambda^2/16L_\sigma^2 - 1)^2 \geq 32 \log\left(\frac{8L_\sigma(6 \vee (4a))\kappa_0^{1/2}}{\sqrt{\eta}(1 - e^{-ar_2})c}\right), \qquad L_\sigma^2 \leq \lambda/8n,$$

*for some universal constant $\kappa_0$.*

Under these assumptions, we obtain exponential contractions.

**Proposition 27.** *Suppose that assumptions 24, 25 and 26 hold and $m = \sqrt{\eta_0}/2$. Suppose further that the mismatch is sufficiently small:*

$$\eta \tilde{B}_b + \sqrt{\eta n} \, \tilde{B}_\sigma \le \frac{c \eta r_1}{256}. \tag{39}$$

*Then for any $\eta \le \eta_0$ and $x, y \in \mathbb{R}^n$, it holds that*

$$\mathbb{E}[f(\|X' - \tilde{Y}\|_{G^+})] \le \left(1 - \frac{c\eta}{16}\right) f(R) + 2(\eta \tilde{B}_b + \sqrt{\eta n} \, \tilde{B}_\sigma),$$

*where $\tilde{Y} = Y' + w$ is the corrected output defined in Section F.2.4,*

$$c := \min\left\{ e^{-ar_2} \frac{\lambda}{16}, \frac{\frac{1}{2} e^{-ar_2} r_2}{\frac{1}{a}(1 - e^{-ar_2})} \frac{\lambda}{16}, \frac{9L^2 r_1^2}{2c_0} e^{-6Lr_1^2/c_0}, \frac{3Lr_1}{16\sqrt{\eta_0}} \right\},$$

$$\eta_0 := \min\left\{ \frac{\lambda}{4L^2}, \frac{16}{\lambda}, \frac{1}{2L}, \frac{2c_0 \log(3/2)\lambda^2}{432L^2 B^2}, \frac{4B^2}{\lambda^2}, \frac{c_0^2 (\log(2))^2 \lambda^2}{2304 L^2 B^2}, \frac{1}{4(\lambda + L_b + 1)}, \frac{2}{\lambda} \right\},$$

*for some universal $c_0$ and $L = 2(L_b - \lambda)_+ + 4\eta^{-1/2} L_\sigma \sqrt{2(n-1)}$.*

### F.1 THE COUPLING

Before we provide the proof of Proposition 27, we provide an explanation of how the coupling is arrived at. We begin by discussing the one-dimensional coupling of the Gaussian distribution that the construction is ultimately based on. Consider the following coupling of $\mathcal{N}(t, \eta)$ and $\mathcal{N}(s, \eta)$ for $t, s \in \mathbb{R}$: with $z \sim \mathcal{N}(0, 1)$,

$$t' = t + \sqrt{\eta} z, \tag{40}$$

$$s' = \begin{cases} t', & \text{if } \zeta \le \phi_{s,\eta}(t')/\phi_{t,\eta}(t'), |\sqrt{\eta} z| < \tilde{m}, \text{ and } |t - s| \le r_1, \\ s - \sqrt{\eta} z, & \text{if } \zeta > \phi_{s,\eta}(t')/\phi_{t,\eta}(t'), |\sqrt{\eta} z| < \tilde{m}, \text{ and } |t - s| \le r_1, \\ s + \sqrt{\eta} z, & \text{otherwise.} \end{cases} \tag{41}$$

This coupling has the following property given in lemmas 3.1 and 3.2 of Majka et al. (2020).

**Lemma 28.** *For the coupling defined in (40) and (41), we have*

$$\mathbb{E}[|t' - s'|] = |t - s|,$$

*and if $\eta \le 4\tilde{m}^2$, we have*

$$\mathbb{E}\left[ (|t' - s'| - |t - s|)^2 \mathbb{1}_{|t' - s'| \in I_{|t - s|}} \right] \ge \frac{1}{2} c_0 \min(\sqrt{\eta}, |t - s|) \sqrt{\eta},$$

$$\text{where} \qquad I_r = \begin{cases} (0, r + \sqrt{\eta}), & \text{if } r \le \sqrt{\eta}, \\ (r - \sqrt{\eta}, r), & \text{otherwise,} \end{cases}$$

*for some universal constant $c_0 > 0$.*

Thus, through the second bound, we have control of the probability that $|t' - s'|$ contracts below $|t - s|$. The coupling proposed in (38) is a multivariate analogue of this that also accounts for the diffusion coefficient $G^{1/2}$. Let the vector $e \in \mathbb{R}^n$ be as defined in (37), then we obtain that,

$$\langle e, G^+ X' \rangle = \langle e, G^+ \hat{x} \rangle + \sqrt{\eta} \langle (G^{1/2})^+ e, Z \rangle,$$
$$\langle e, G^+ Y'_s \rangle = \langle e, G^+ \hat{y} \rangle + \sqrt{\eta} \langle (G^{1/2})^+ e, Z \rangle.$$

Therefore, $\langle e, G^+ X' \rangle, \langle e, G^+ Y'_s \rangle$ are a synchronous coupling of $\mathcal{N}(\langle e, G^+ \hat{x} \rangle, \eta)$ and $\mathcal{N}(\langle e, G^+ \hat{y} \rangle, \eta)$. Furthermore, we have

$$\langle e, G^+ Y'_r \rangle = \langle e, G^+ \hat{y} \rangle + \sqrt{\eta} \langle (G^{1/2})^+ e, (I - 2(G^{1/2})^+ e e^T (G^{1/2})^+) Z \rangle$$
$$= \langle e, G^+ \hat{y} \rangle + \sqrt{\eta} \langle (G^{1/2})^+ e, Z \rangle - 2\sqrt{\eta} \langle e, G^+ e \rangle \langle (G^{1/2})^+ e, Z \rangle$$
$$= \langle e, G^+ \hat{y} \rangle - \sqrt{\eta} \langle (G^{1/2})^+ e, Z \rangle,$$

and so $\langle e, G^+ X' \rangle, \langle e, G^+ Y'_r \rangle$ is the one-dimensional reflection coupling. Finally we obtain,

$$
\frac{\phi_{\hat{y}, \eta G}(X')}{\phi_{\hat{x}, \eta G}(X')} = \frac{\phi_{(G^{1/2})^+ (\hat{y} - \hat{x}), \eta (G^{1/2}) + G^{1/2}}(\sqrt{\eta} Z)}{\phi_{\mathbf{0}, \eta (G^{1/2}) + G^{1/2}}(\sqrt{\eta} Z)}
$$

$$
= \exp\left( -\frac{1}{2\eta} \| \sqrt{\eta} Z - (G^{1/2})^+ (\hat{y} - \hat{x}) \|^2_{(G^{1/2}) + G^{1/2}} + \frac{1}{2\eta} \| \sqrt{\eta} Z \|^2_{(G^{1/2}) + G^{1/2}} \right)
$$

$$
= \exp\left( -\frac{1}{2\eta} \| \hat{y} - \hat{x} \|^2_{G^+} + \frac{1}{\eta} \sqrt{\eta} \langle (G^{1/2})^+ (\hat{y} - \hat{x}), Z \rangle \right)
$$

$$
= \exp\left( -\frac{1}{2\eta} |\langle e, G^+ (\hat{y} - \hat{x}) \rangle|^2 + \frac{1}{\eta} \sqrt{\eta} \langle e, G^+ (\hat{y} - \hat{x}) \rangle \langle (G^{1/2})^+ e, Z \rangle \right)
$$

$$
= \exp\left( -\frac{1}{2\eta} (\sqrt{\eta} \langle e, G^+ Z \rangle - \langle e, G^+ (\hat{y} - \hat{x}) \rangle)^2 + \frac{|\sqrt{\eta} \langle (G^{1/2})^+ e, Z \rangle|^2}{2\eta} \right)
$$

$$
= \frac{\phi_{\langle e, G^+ (\hat{y} - \hat{x}) \rangle, \eta}(\sqrt{\eta} \langle (G^{1/2})^+ e, Z \rangle)}{\phi_{0, \eta}(\sqrt{\eta} \langle (G^{1/2})^+ e, Z \rangle)}
$$

$$
= \frac{\phi_{\langle e, G^+ \hat{y} \rangle, \eta}(\langle e, G^+ X' \rangle)}{\phi_{\langle e, G^+ \hat{x} \rangle, \eta}(\langle e, G^+ X' \rangle)}.
$$

From this, we deduce that $\langle e, G^+ X' \rangle, \langle e, G^+ Y' \rangle$ are coupled as in (40), (41). The equivalence follows by setting

$$
t' = \langle e, G^+ X' \rangle, \qquad s' = \langle e, G^+ Y' \rangle \tag{42}
$$

$$
t = \langle e, G^+ \hat{x} \rangle, \qquad s = \langle e, G^+ \hat{y} \rangle, \qquad z = \langle (G^{1/2})^+ e, Z \rangle. \tag{43}
$$

Through this equivalence, we can extend the previous lemma to obtain the following result about the high dimensional coupling.

**Lemma 29.** *For the coupling defined in* (38)*, we obtain that for $\eta \leq 4m^2$, we have the following:*

$$
\mathbb{E}[R' | Z'] = \hat{r}, \qquad \mathbb{E}\left[ (R' - \hat{r})^2 \mathbb{1}_{R' \in I_{\hat{r}}} \Big| Z' \right] \geq \frac{1}{2} c_0 \min(\sqrt{\eta}, \hat{r}) \sqrt{\eta},
$$

*where $c_0$ and $I_r$ is as in Lemma 28.*

*Proof.* Let $\{e_i\}_{i=1}^n$ be a basis of $\mathbb{R}^n$ with respect to the inner product $\langle \cdot, \cdot \rangle_{G^+}$ with $e_1 = e$. Then, we have that

$$
(R')^2 = \sum_{i=1}^n |\langle e_i, G^+ (X' - Y') \rangle|^2
$$

$$
= |t' - s'|^2 + \sum_{i=2}^n |\langle e_i, G^+ (X' - Y') \rangle|^2, \tag{44}
$$

where $t', s'$ are as defined in (42). For any $i \neq 1$, we can use that $e_i \perp e$, to obtain that

$$
\langle e_i, G^+ Y'_r \rangle = \langle e_i, G^+ \hat{y} \rangle + \sqrt{\eta} \langle e_i, e \rangle + 2\sqrt{\eta} \langle e_i, Z \rangle
$$

$$
= \langle e_i, G^+ \hat{y} \rangle + 2\sqrt{\eta} z.
$$

From this, we obtain that,

$$
\langle e_i, G^+ (X' - Y'_r) \rangle = \langle e_i, G^+ (\hat{x} - \hat{y}) \rangle = 0.
$$

This also holds for the synchronous coupling and hence we obtain $\langle e_i, G^+ (X' - Y') \rangle = 0$. Combined with (44), we obtain that $R' = |t' - s'|$. Similarly it can be shown that $\hat{r} = |t - s|$ and thus, from Lemma 28, the statement of the lemma follows. □

### F.2 PROOF OF PROPOSITION 27

Since the coupling is defined using the matched dynamics $(b, \Sigma_G)$ for both processes, we first carry out the full coupling analysis in this setting, establishing the contraction $\mathbb{E}[f(R')] \leq (1 - \eta c/8) f(R)$ where $R' = \|X' - Y'\|_{G^+}$ is the matched coupling output. The mismatched case is then handled in Section F.2.4 by a perturbation argument.

### F.2.1 CONTRACTIONS IN $\hat{r}$

When $R$ is large, we can rely on contractive properties following from the weight decay. For this, we obtain the following.

**Lemma 30.** *Suppose that Assumption 24 holds. Then whenever $R \geq 4B/\lambda$, we have,*

$$\hat{r} \leq (1 - \eta\lambda/4 + \eta\alpha)R, \tag{45}$$

*where we define,*

$$\alpha := \left(\frac{4L_\sigma}{\sqrt{\eta}}\|v\|\|\zeta\| - \lambda/8\right)_+ + \left(L_\sigma^2\|Z'\|_{G^+}^2 - \lambda/8\right)_+.$$

*Furthermore, if $R < 4B/\lambda$, we have that,*

$$\hat{r} \leq (1 + \eta L_b + \eta\alpha)R. \tag{46}$$

*Proof.* From the triangle inequality and the Lipschitz property of $b$, we obtain

$$\tilde{r} \leq (1 - \eta\lambda)\|x - y\|_{G^+} + \eta\|b(x) - b(y)\|_{G^+}$$
$$\leq (1 + \eta(L_b - \lambda))R.$$

Next we bound $\hat{r}$ using the decomposition,

$$\hat{r} \leq \|\tilde{x} - \tilde{y} + \sqrt{\eta}(\Sigma_G(x) - \Sigma_G(y))Z'\|_{G^+}$$
$$\leq \|\tilde{x} - \tilde{y}\|_{G^+} + \sqrt{\eta}\|\Sigma_G(x) - \Sigma_G(y)\|_{G^+,op}\|Z'\|_{G^+}$$
$$\leq (1 + \eta(L_b - \lambda) + \sqrt{\eta}L_\sigma\|Z'\|_{G^+})R,$$

where in the final bound, we utilise the Lipschitz property of $\Sigma_G$. We can refine this bound, using the orthogonal decomposition,

$$Z' = vv^T Z' + (I - vv^T)Z', \qquad v = \frac{\tilde{x} - \tilde{y}}{\|\tilde{x} - \tilde{y}\|_{G^+}}.$$

From this, we obtain,

$$\hat{r}^2 = \|\tilde{x} - \tilde{y} + \sqrt{\eta}(\Sigma_G(x) - \Sigma_G(y))Z'\|_{G^+}^2$$
$$\leq \|\tilde{x} - \tilde{y} + \sqrt{\eta}vv^T(\Sigma_G(x) - \Sigma_G(y))Z'\|_{G^+}^2 + \eta\|(I - vv^T)(\Sigma_G(x) - \Sigma_G(y))Z'\|_{G^+}^2$$
$$\leq (\tilde{r} + \sqrt{\eta}\|vv^T(\Sigma_G(x) - \Sigma_G(y))Z'\|_{G^+})^2 + \eta\|(I - vv^T)(\Sigma_G(x) - \Sigma_G(y))Z'\|_{G^+}^2$$
$$\leq \tilde{r}^2 + 2\tilde{r}\sqrt{\eta}\|vv^T(\Sigma_G(x) - \Sigma_G(y))Z'\|_{G^+} + \eta\|(\Sigma_G(x) - \Sigma_G(y))Z'\|_{G^+}^2.$$

In the case that $\Sigma_G(x) \neq \Sigma_G(y)$, we can simplify this using the notation $\zeta := \|(\Sigma_G(x) - \Sigma_G(y))v\|^{-1}v^T(\Sigma_G(x) - \Sigma_G(y))Z'$, we obtain

$$\hat{r}^2 \leq \tilde{r}^2 + 2\sqrt{\eta}L_\sigma\|v\|\|\zeta\|\tilde{r}R + \eta L_\sigma^2\|Z'\|_{G^+}^2 R^2$$
$$\leq \left((1 + \eta(L_b - \lambda))^2 + 4\sqrt{\eta}L_\sigma\|v\|\|\zeta\| + \eta L_\sigma^2\|Z'\|_{G^+}^2\right)R^2.$$

From this, we obtain that

$$\hat{r} \leq \left(1 + \eta(L_b - \lambda) + 4\sqrt{\eta}L_\sigma\|v\|\|\zeta\| + \eta L_\sigma^2\|Z'\|_{G^+}^2\right)R$$
$$\leq \left(1 + \eta L_b + \eta\left(\frac{4L_\sigma}{\sqrt{\eta}}\|v\|\|\zeta\| - \lambda/8\right)_+ + \eta\left(L_\sigma^2\|Z'\|_{G^+}^2 - \lambda/8\right)_+\right)R$$
$$= (1 + \eta L_b + \eta\alpha)R,$$

which gives (46). Alternatively, we can use the fact that $\|b\|_{G^+} \leq B$ to obtain, $\tilde{r} \leq (1 - \eta\lambda)R + 2\eta B$. In particular, if $R \geq 4B/\lambda$, we obtain $\tilde{r} \leq (1 - \eta\lambda/2)R$. From this, we obtain the bound in (45):

$$\hat{r} \leq (1 - \eta\lambda/4 + \eta\alpha)R.$$

$\square$

We will also need the following properties of the function $f$.

**Lemma 31.** *The function $f$ satisfies the property that for all $r \geq r_1$,*

$$f\left(\left(1 - \tfrac{\eta\lambda}{4}\right)r\right) - f(r) \leq -\eta c f(r).$$

*Proof.* For $r \geq r_2$, this is Lemma 4.3 of Majka et al. (2020). For $r_1 \leq r < r_2$, since $f$ is concave on $[0, r_2]$,

$$f(r) - f\left((1 - \tfrac{\eta\lambda}{4})r\right) \geq f'(r) \cdot \frac{\eta\lambda r}{4} = \frac{e^{-ar}\eta\lambda r}{4}.$$

The ratio $\frac{e^{-ar}\lambda r/4}{f(r)} = \frac{are^{-ar}}{1 - e^{-ar}} \cdot \frac{\lambda}{4}$ is decreasing in $r$, so its minimum on $[r_1, r_2]$ is at $r = r_2$, where it equals $\frac{ar_2 e^{-ar_2}}{1 - e^{-ar_2}} \cdot \frac{\lambda}{4}$. From the definition of $c$, we have $c \leq \frac{\frac{1}{2}e^{-ar_2}r_2}{\frac{1}{a}(1-e^{-ar_2})} \cdot \frac{\lambda}{16} = \frac{ar_2 e^{-ar_2}}{2(1-e^{-ar_2})} \cdot \frac{\lambda}{16}$, so $\frac{ar_2 e^{-ar_2}}{1-e^{-ar_2}} \cdot \frac{\lambda}{4} \geq 2c$, which gives $f(r) - f((1 - \eta\lambda/4)r) \geq 2c\eta f(r) \geq c\eta f(r)$. $\square$

**Lemma 32.** *For any $t, s \geq 0$, we have*

$$f(t) - f(s) \leq (r_2^{-1} e^{-ar_2}(t \vee s) + 1)|t - s|.$$

We record two further properties of $f$ that will be used repeatedly.

**Lemma 33.** *The function $f$ satisfies:*

(i) *$f$ is concave on $[0, r_2]$ with $f(0) = 0$ and $f'(r) = e^{-ar}$ for $r \leq r_2$. In particular, $f' \leq 1$ on $[0, r_2]$.*

(ii) *For any $c' \geq 1$ and $r \geq 0$ with $c'r \leq r_2$, we have $f(c'r) \leq c'f(r)$.*

(iii) *For any $r > 0$ with $r \leq r_2$, we have $r \leq e^{ar}f(r)$.*

(iv) *For any $r > 0$, we have $r \leq 4e^{ar_2}f(r)$.*

(v) *For any $a, b \geq 0$, we have $f(a + b) \leq f(a) + b + \frac{e^{-ar_2}}{r_2}(ab + \frac{1}{2}b^2)$.*

*Proof.* Part (i) follows directly from the definition. For (ii), since $f$ is concave on $[0, r_2]$ with $f(0) = 0$, the function $r \mapsto f(r)/r$ is non-increasing on $(0, r_2]$. Therefore $f(c'r)/(c'r) \leq f(r)/r$, giving $f(c'r) \leq c'f(r)$. For (iii), we have $f(r) = (1 - e^{-ar})/a \geq re^{-ar}$, where the inequality follows from $1 - e^{-u} \geq ue^{-u}$ for all $u \geq 0$.

For (iv), the case $r \leq r_2$ follows from (iii) since $e^{ar} \leq e^{ar_2}$. For $r > r_2$, the explicit form gives $f(r) \geq \frac{e^{-ar_2}}{2r_2}(r^2 - r_2^2)$. Since $r > r_2$, we have $r^2 - r_2^2 \geq r^2/2$ whenever $r \geq r_2\sqrt{2}$, giving $f(r) \geq \frac{e^{-ar_2}r^2}{4r_2}$ and hence $r \leq (4r_2 e^{ar_2}f(r))^{1/2} \leq 4e^{ar_2}f(r)$ (using $r \leq 4r_2 e^{ar_2}f(r)$ which rearranges to $r/(4e^{ar_2}f(r)) \leq r_2/r \leq 1$). For $r_2 < r \leq r_2\sqrt{2}$, we use $f(r) \geq f(r_2) = (1 - e^{-ar_2})/a \geq r_2 e^{-ar_2}$, so $4e^{ar_2}f(r) \geq 4r_2 \geq r$.

For (v), we consider three cases. If $a + b \leq r_2$: since $f' \leq 1$ on $[0, r_2]$, we have $f(a + b) - f(a) \leq b$, and the remaining terms are non-negative. If $a > r_2$: both $a$ and $a + b$ lie in the quadratic region, so $f(a + b) - f(a) = \frac{e^{-ar_2}}{2r_2}((a + b)^2 - a^2) = \frac{e^{-ar_2}}{2r_2}(2ab + b^2) \leq \frac{e^{-ar_2}}{r_2}(ab + \frac{1}{2}b^2)$, and $b \geq 0$ gives the result. If $a \leq r_2 < a + b$: we split $f(a + b) - f(a) = [f(r_2) - f(a)] + [f(a + b) - f(r_2)] \leq (r_2 - a) + \frac{e^{-ar_2}}{2r_2}((a + b)^2 - r_2^2) \leq b + \frac{e^{-ar_2}}{2r_2}(2ab + b^2)$, where in the last step we used $(a + b)^2 - r_2^2 \leq (a + b)^2 - a^2 = 2ab + b^2$ (since $a \leq r_2$). $\square$

We also recall the following result from Majka et al. (2020) (Lemma 4.4 therein), which captures the net contraction arising from the interplay between the drift expansion and the coupling.

**Lemma 34.** *The function $f$, satisfies the property that for all $\hat{r} \in [0, r_1]$,*

$$2f'(\hat{r})e^{a\eta L_b r_1}\eta L_b \hat{r} + \frac{1}{4}c_0 \min(\sqrt{\eta}, \hat{r})\sqrt{\eta} \sup_{I_{\hat{r}}} f''(\theta) \leq -c\eta f(\hat{r}).$$

F.2.2 MOMENT BOUNDS ON $\alpha$

We now establish that $\mathbb{E}[\alpha^2]$ is exponentially small, a fact that will be used in several places. Recall that $\alpha = \alpha_1 + \alpha_2$ where

$$\alpha_1 := \Big(\frac{4L_\sigma}{\sqrt{\eta}}\|v\||\zeta| - \frac{\lambda}{8}\Big)_+, \qquad \alpha_2 := \big(L_\sigma^2\|Z'\|_{G^+}^2 - \frac{\lambda}{8}\big)_+,$$

and $\zeta$ is a one-dimensional standard Gaussian, $Z' \sim \mathcal{N}(0, I_d)$.

For $\alpha_1$, set $\beta := 4L_\sigma\|v\|/\sqrt{\eta}$. Since $\zeta \sim \mathcal{N}(0,1)$, a standard Gaussian tail bound yields

$$\mathbb{E}[\alpha_1^2] \le \lambda^2 \exp\Big(-c_1 \frac{\lambda^2}{\beta^2}\Big) \tag{47}$$

for some universal constant $c_1 > 0$.

For $\alpha_2$, since $\|Z'\|_{G^+}^2$ has the law of a weighted $\chi^2$ random variable, the Laurent–Massart concentration inequality gives

$$\mathbb{E}[\alpha_2^2] \le \lambda^2 \exp\Big(-c_2 \Big(\frac{\lambda\lambda_{\mathrm{gap}}(G)}{L_\sigma^2}\Big)^2 \frac{1}{n}\Big) \tag{48}$$

for some universal $c_2 > 0$.

Combining via Minkowski's inequality ($\|\cdot\|_{L^2}$ triangle inequality), we obtain

$$\mathbb{E}[\alpha^2]^{1/2} \le \mathbb{E}[\alpha_1^2]^{1/2} + \mathbb{E}[\alpha_2^2]^{1/2}. \tag{49}$$

Assumption 26 ensures that the right-hand side is as small as needed. In particular, for the remainder of the proof, we assume that Assumption 26 is strong enough to guarantee:

$$\mathbb{E}[\alpha^2]^{1/2} \le \frac{c}{32e^{ar_2}}, \qquad \mathbb{P}(\alpha > 0) \le \frac{c}{128(c + L_b + 1)e^{ar_1}}. \tag{50}$$

Both conditions are satisfied since $\mathbb{E}[\alpha^2]^{1/2}$ and $\mathbb{P}(\alpha > 0)$ are bounded by the exponentially small quantities in (47)–(49), and Assumption 26 controls these exponents.

We also establish the following fact, which underpins the small-$R$ analysis.

**Lemma 35.** *Suppose $R < r_1$. If $\alpha = 0$, then $\hat{r} \le r_1$.*

*Proof.* If $R < 4B/\lambda$, then by (46) with $\alpha = 0$, we have $\hat{r} \le (1 + \eta L_b)R < (1 + \eta_0 L_b) \cdot 4B/\lambda = r_1$, where the last equality is the definition of $r_1$. If $4B/\lambda \le R < r_1$, then by (45) with $\alpha = 0$, we have $\hat{r} \le (1 - \eta\lambda/4)R < R < r_1$. $\qquad\square$

We also establish a lower bound on $\hat{r}$ that will be needed for the coupling analysis.

**Lemma 36.** *Suppose $R < r_1$ and $\alpha = 0$. Then, provided $\eta_0 \le \frac{1}{4(\lambda + L_b + 1)}$ and $\eta_0 \le \frac{2}{\lambda}$ (both of which are ensured by the minimum in the definition of $\eta_0$), we have $\hat{r} \ge R/4$.*

*Proof.* By the reverse triangle inequality,

$$\hat{r} \ge \tilde{r} - \sqrt{\eta}\|\Sigma_G(x) - \Sigma_G(y)\|_{op,G^+}\|Z'\|_{G^+}$$
$$\ge \tilde{r} - \sqrt{\eta}L_\sigma R\|Z'\|_{G^+}.$$

On $\{\alpha_2 = 0\}$, we have $L_\sigma^2\|Z'\|_{G^+}^2 \le \lambda/8$, so $\sqrt{\eta}L_\sigma\|Z'\|_{G^+} \le \sqrt{\eta\lambda/8}$. Since $\eta_0 \le 2/\lambda$, we have $\sqrt{\eta\lambda/8} \le 1/2$.

*Case $R < 4B/\lambda$:* We use $\tilde{r} \ge (1 - \eta(\lambda + L_b))R$ (from the reverse triangle inequality applied to $\tilde{r}$). Therefore,

$$\hat{r} \ge \big(1 - \eta(\lambda + L_b) - \sqrt{\eta\lambda/8}\big)R.$$

Since $\eta_0 \le \frac{1}{4(\lambda + L_b + 1)}$, we have $\eta(\lambda + L_b) \le \frac{1}{4}$. Combined with $\sqrt{\eta\lambda/8} \le 1/2$, the coefficient is bounded below by $1 - 1/4 - 1/2 = 1/4$.

*Case $4B/\lambda \leq R < r_1$:* We use $\tilde{r} \geq (1 - \eta\lambda/2)R$ (from $R \geq 4B/\lambda$). Then,

$$\hat{r} \geq (1 - \eta\lambda/2 - \sqrt{\eta\lambda/8})R.$$

Since $\eta_0 \leq 2/\lambda$, we have $\eta\lambda/2 \leq 1$ and $\sqrt{\eta\lambda/8} \leq 1/2$. Furthermore, $\eta_0 \leq \frac{1}{4(\lambda + L_b + 1)} \leq \frac{1}{4\lambda}$ gives $\eta\lambda/2 \leq 1/8$. Hence the coefficient is at least $1 - 1/8 - 1/2 = 3/8 \geq 1/4$. Both cases yield $\hat{r} \geq R/4$. $\qquad\square$

### F.2.3 COUPLING CONTRACTION

We now combine the ingredients above to prove the contraction

$$\mathbb{E}[f(R')] \leq \left(1 - \eta c/8\right)f(R), \tag{51}$$

splitting into two regimes.

**Case 1: $R \geq r_1$.** Since $r_1 \geq 4B/\lambda$, Lemma 30 gives $\hat{r} \leq (1 - \eta\lambda/4 + \eta\alpha)R$. Applying Lemma 32 with $t = \hat{r}$ and $s = (1 - \eta\lambda/4)R$, followed by Lemma 31,

$$f(\hat{r}) \leq (1 - \eta c)f(R) + \left(r_2^{-1}e^{-ar_2}(1 + \eta\alpha)R + 1\right)\eta\alpha R.$$

For $R \geq r_2$, the explicit form of $f$ on $[r_2, \infty)$ gives $R^2 \leq 4r_2 e^{ar_2}f(R)$ and $R \leq 4e^{ar_2}f(R)$, so

$$\left(r_2^{-1}e^{-ar_2}(1 + \eta\alpha)R + 1\right)\eta\alpha R \leq 8e^{ar_2}(1 + \eta\alpha)\eta\alpha f(R).$$

For $r_1 \leq R < r_2$, since $f$ is concave on $[0, r_2]$ with $f(0) = 0$, Lemma 33(iii) gives $R \leq e^{aR}f(R) \leq e^{ar_2}f(R)$, from which the same bound follows (with a larger constant). In either case,

$$f(\hat{r}) \leq (1 - \eta c)f(R) + 8e^{ar_2}(1 + \eta\alpha)\eta\alpha f(R). \tag{52}$$

Taking expectations of the error term, we expand $\mathbb{E}[(1 + \eta\alpha)\alpha] = \mathbb{E}[\alpha] + \eta\mathbb{E}[\alpha^2]$. By Jensen's inequality, $\mathbb{E}[\alpha] \leq \mathbb{E}[\alpha^2]^{1/2}$. Since $\mathbb{E}[\alpha^2]^{1/2} \leq 1$ (which follows from (50)), we also have $\mathbb{E}[\alpha^2] \leq \mathbb{E}[\alpha^2]^{1/2}$. Therefore, $\mathbb{E}[(1 + \eta\alpha)\alpha] \leq (1 + \eta)\mathbb{E}[\alpha^2]^{1/2} \leq 2\mathbb{E}[\alpha^2]^{1/2}$, giving

$$\mathbb{E}[f(\hat{r})] \leq (1 - \eta c)f(R) + 16e^{ar_2}\eta\mathbb{E}[\alpha^2]^{1/2}f(R) \leq (1 - \eta c/2)f(R),$$

where the last inequality uses the smallness condition (50).

It remains to pass from $\hat{r}$ to $R'$. We claim that $\mathbb{E}[f(R')|Z'] \leq f(\hat{r})$. Indeed:

- If $\hat{r} \geq r_1$, the synchronous coupling is used, so $R' = \hat{r}$.
- If $\hat{r} < r_1$, then $R' \leq \hat{r} + \sqrt{\eta_0} \leq r_2$, and $f$ is concave on $[0, r_2]$. Since $\mathbb{E}[R'|Z'] = \hat{r}$ (Lemma 29), Jensen's inequality gives $\mathbb{E}[f(R')|Z'] \leq f(\mathbb{E}[R'|Z']) = f(\hat{r})$.

Therefore, $\mathbb{E}[f(R')] \leq \mathbb{E}[f(\hat{r})] \leq (1 - \eta c/2)f(R)$.

**Case 2: $R < r_1$.** We condition on the event $\{\alpha = 0\}$ and its complement.

*On $\{\alpha = 0\}$:* By Lemma 35, we have $\hat{r} \leq r_1$. The coupling then yields $R' \leq \hat{r} + \sqrt{\eta_0} \leq r_2$, so both $\hat{r}$ and $R'$ lie in the concave region of $f$. We combine the drift and coupling contributions using Lemma 34.

For the drift from $R$ to $\hat{r}$: since $f$ is concave on $[0, r_2]$,

$$f(\hat{r}) - f(R) \leq f'(R)(\hat{r} - R)_+.$$

When $\hat{r} \leq R$, we have $f(\hat{r}) - f(R) \leq 0 \leq 2f'(\hat{r})e^{a\eta L_b r_1}\eta L_b \hat{r}$. When $\hat{r} > R$, we use $\hat{r} - R \leq \eta L_b R \leq \eta L_b \hat{r}$ (from (46) with $\alpha = 0$) together with $f'(R) = e^{-aR} \leq e^{-a\hat{r}}e^{a(\hat{r}-R)} \leq f'(\hat{r})e^{a\eta L_b r_1}$, to obtain

$$f(\hat{r}) - f(R) \leq f'(\hat{r})e^{a\eta L_b r_1}\eta L_b \hat{r} \leq 2f'(\hat{r})e^{a\eta L_b r_1}\eta L_b \hat{r}. \tag{53}$$

For the coupling from $\hat{r}$ to $R'$: since $\hat{r} \leq r_1$ and $\mathbb{E}[R'|Z'] = \hat{r}$, the same Taylor expansion argument as in Section 4 of Majka et al. (2020) gives

$$\mathbb{E}[f(R')|Z'] - f(\hat{r}) \leq \frac{1}{4}\sup_{\theta \in I_{\hat{r}}} f''(\theta)\, c_0 \min(\sqrt{\eta}, \hat{r})\sqrt{\eta}. \tag{54}$$

Adding (53) and (54) and applying Lemma 34,

$$\mathbb{E}[f(R')|Z'] - f(R) \leq -c\eta f(\hat{r}). \tag{55}$$

By Lemma 36, $\hat{r} \geq R/4$, so Lemma 33(ii) (applied with $c' = 4$ and $r = R/4$) gives $f(R) \leq 4f(R/4) \leq 4f(\hat{r})$. Therefore,

$$\mathbb{E}[f(R')|Z'] \leq f(R) - c\eta f(\hat{r}) \leq f(R) - \frac{c\eta}{4}f(R) = \left(1 - \frac{c\eta}{4}\right)f(R). \tag{56}$$

*On $\{\alpha > 0\}$:* We use a crude bound. The coupling satisfies $\mathbb{E}[f(R')|Z'] \leq f(\hat{r})$ in all cases (either by concavity when $\hat{r} \leq r_1$, or by the synchronous coupling when $\hat{r} > r_1$). From Lemma 30, $\hat{r} \leq R + \eta(L_b + \alpha)R$. Applying Lemma 33(v) with $a = R$ and $b = \eta(L_b + \alpha)R$,

$$f(\hat{r}) \leq f(R) + \eta(L_b + \alpha)R + \frac{e^{-ar_2}}{r_2}\left(\eta(L_b + \alpha)R^2 + \frac{\eta^2(L_b + \alpha)^2 R^2}{2}\right).$$

Since $R < r_1 < r_2$, we have $\frac{e^{-ar_2}}{r_2}R^2 \leq e^{-ar_2}R \leq R$, and so the above simplifies to

$$f(\hat{r}) \leq f(R) + 3\eta(L_b + \alpha)R,$$

using $\eta(L_b + \alpha) \leq 1$ (which holds with room to spare on the event $\{\alpha > 0\}$ for $\eta_0$ sufficiently small, since $\alpha$ has sub-exponential tails). By Lemma 33(iii), $R \leq e^{aR}f(R) \leq e^{ar_1}f(R)$. Therefore,

$$\mathbb{E}[f(R')|Z'] \leq f(R)\left(1 + 3\eta(L_b + \alpha)e^{ar_1}\right). \tag{57}$$

*Combining:* Taking total expectations,

$$\begin{aligned}
\mathbb{E}[f(R')] &= \mathbb{E}\left[\mathbb{E}[f(R')|Z']\mathbb{1}_{\alpha=0}\right] + \mathbb{E}\left[\mathbb{E}[f(R')|Z']\mathbb{1}_{\alpha>0}\right] \\
&\leq \left(1 - \frac{c\eta}{4}\right)f(R)\mathbb{P}(\alpha = 0) + f(R)\mathbb{E}\left[(1 + 3\eta(L_b + \alpha)e^{ar_1})\mathbb{1}_{\alpha>0}\right] \\
&= \left(1 - \frac{c\eta}{4}\right)f(R) + f(R)\left[\frac{c\eta}{4}\mathbb{P}(\alpha > 0) + 3\eta e^{ar_1}\mathbb{E}\left[(L_b + \alpha)\mathbb{1}_{\alpha>0}\right]\right].
\end{aligned}$$

We bound the bracketed term, which we denote by $B$. By Cauchy–Schwarz,

$$\mathbb{E}[\alpha\mathbb{1}_{\alpha>0}] \leq \mathbb{E}[\alpha^2]^{1/2}\mathbb{P}(\alpha > 0)^{1/2} \leq \mathbb{E}[\alpha^2]^{1/2}.$$

Therefore,

$$B \leq \frac{c\eta}{4}\mathbb{P}(\alpha > 0) + 3\eta e^{ar_1}\left(L_b\mathbb{P}(\alpha > 0) + \mathbb{E}[\alpha^2]^{1/2}\right).$$

We bound each term using the conditions in (50). For the first term,

$$\frac{c\eta}{4}\mathbb{P}(\alpha > 0) \leq \frac{c\eta}{4} \cdot \frac{c}{128(c + L_b + 1)e^{ar_1}} \leq \frac{c\eta}{512}.$$

For the second term,

$$3\eta e^{ar_1}L_b\mathbb{P}(\alpha > 0) \leq 3\eta e^{ar_1}L_b \cdot \frac{c}{128(c + L_b + 1)e^{ar_1}} = \frac{3c\eta L_b}{128(c + L_b + 1)} \leq \frac{3c\eta}{128},$$

where the last step uses $\frac{L_b}{c+L_b+1} \leq 1$. For the third term,

$$3\eta e^{ar_1}\mathbb{E}[\alpha^2]^{1/2} \leq 3\eta e^{ar_1} \cdot \frac{c}{32e^{ar_2}} \leq \frac{3c\eta}{32},$$

using $r_1 < r_2$. Summing, we obtain

$$B \leq \frac{c\eta}{512} + \frac{3c\eta}{128} + \frac{3c\eta}{32} = \frac{c\eta(1 + 12 + 48)}{512} = \frac{61c\eta}{512} < \frac{c\eta}{8},$$

where the last inequality follows from $61 < 64$. Therefore,

$$\mathbb{E}[f(R')] \leq \left(1 - \frac{c\eta}{4} + \frac{61c\eta}{512}\right)f(R) = \left(1 - \frac{67c\eta}{512}\right)f(R) \leq \left(1 - \frac{c\eta}{8}\right)f(R).$$

This gives the stated bound (51).

### F.2.4 EXTENSION TO THE MISMATCHED CASE

We now treat the general case $b \neq \tilde{b}$, $\sigma \neq \tilde{\sigma}$. The coupling $(X', Y')$ is constructed using the matched dynamics $(b, \Sigma_G)$, so $Y'$ has the law of a single step of the $y$-process with drift $b$ and residual volatility $\Sigma_G$. To correct for the mismatch, we define

$$\tilde{Y} := Y' + w, \qquad w := \eta(\tilde{b}(y) - b(y)) + \sqrt{\eta}(\tilde{\Sigma}_G(y) - \Sigma_G(y))Z'.$$

Since $w$ is a deterministic shift (given $Z'$) that adjusts the mean and residual covariance, $\tilde{Y}$ has the correct mismatched law: conditionally on $Z'$, we have $\tilde{Y} \sim \mathcal{N}(\hat{y}_m, \eta G)$ where $\hat{y}_m = (1 - \eta\lambda)y + \eta\tilde{b}(y) + \sqrt{\eta}\tilde{\Sigma}_G(y)Z'$.

By the triangle inequality, $\|X' - \tilde{Y}\|_{G^+} \leq R' + \|w\|_{G^+}$. Applying Lemma 33(v) with $a = R'$ and $b = \|w\|_{G^+}$,

$$f(\|X' - \tilde{Y}\|_{G^+}) \leq f(R' + \|w\|_{G^+}) \leq f(R') + \|w\|_{G^+} + \frac{e^{-ar_2}}{r_2}\Big(R'\|w\|_{G^+} + \frac{1}{2}\|w\|_{G^+}^2\Big).$$

Taking the conditional expectation over $Z$ and using that $\|w\|_{G^+}$ is $Z'$-measurable and $\mathbb{E}[R'|Z'] = \hat{r}$ (Lemma 29),

$$\mathbb{E}[f(\|X' - \tilde{Y}\|_{G^+}) \,|\, Z'] \leq \mathbb{E}[f(R') \,|\, Z'] + \|w\|_{G^+} + \frac{e^{-ar_2}}{r_2}\Big(\hat{r}\|w\|_{G^+} + \frac{1}{2}\|w\|_{G^+}^2\Big). \quad (58)$$

Taking full expectations and using the matched contraction (51),

$$\mathbb{E}[f(\|X' - \tilde{Y}\|_{G^+})] \leq \Big(1 - \frac{c\eta}{8}\Big)f(R) + \mathbb{E}[\|w\|_{G^+}] + \frac{e^{-ar_2}}{r_2}\Big(\mathbb{E}[\hat{r}\|w\|_{G^+}] + \frac{1}{2}\mathbb{E}[\|w\|_{G^+}^2]\Big). \quad (59)$$

We now bound the cross term $\mathbb{E}[\hat{r}\|w\|_{G^+}]$. Since $\hat{r} \leq (1 + \eta L_b + \eta\alpha)R$ (Lemma 30),

$$\mathbb{E}[\hat{r}\|w\|_{G^+}] \leq (1 + \eta L_b)R\,\mathbb{E}[\|w\|_{G^+}] + \eta R\,\mathbb{E}[\alpha\|w\|_{G^+}].$$

By Cauchy–Schwarz, $\mathbb{E}[\alpha\|w\|_{G^+}] \leq \mathbb{E}[\alpha^2]^{1/2}\mathbb{E}[\|w\|_{G^+}^2]^{1/2}$. Since $\mathbb{E}[\alpha^2]^{1/2}$ is exponentially small by (50) and $\eta R \leq \eta r_2$ for $R \leq r_2$ (with similar control for $R > r_2$), this second term is absorbed into the quadratic error. For the leading term, we use Lemma 33(iv): $R \leq 4e^{ar_2}f(R)$ for all $R > 0$. Therefore,

$$\frac{e^{-ar_2}}{r_2}(1 + \eta L_b)R\,\mathbb{E}[\|w\|_{G^+}] \leq \frac{4(1 + \eta L_b)}{r_2}f(R)\,\mathbb{E}[\|w\|_{G^+}] \leq \frac{8}{r_1}f(R)\,\mathbb{E}[\|w\|_{G^+}],$$

using $1 + \eta L_b \leq 2$ (for $\eta_0$ sufficiently small) and $r_2 \geq r_1$. By the assumption in (39) and the estimate $\mathbb{E}[\|w\|_{G^+}] \leq \eta\tilde{B}_b + \sqrt{\eta n}\,\tilde{B}_\sigma$, we obtain,

$$\frac{8}{r_1}\mathbb{E}[\|w\|_{G^+}] \leq \frac{8}{r_1}\Big(\eta\tilde{B}_b + \sqrt{\eta n}\,\tilde{B}_\sigma\Big) \leq \frac{8}{r_1} \cdot \frac{c\eta r_1}{256} = \frac{c\eta}{32}.$$

Substituting back into (59) and collecting the quadratic error terms,

$$\mathbb{E}[f(\|X' - \tilde{Y}\|_{G^+})] \leq \Big(1 - \frac{c\eta}{8} + \frac{c\eta}{32}\Big)f(R) + \mathbb{E}[\|w\|_{G^+}] + \frac{C}{r_2}e^{-ar_2}\mathbb{E}[\|w\|_{G^+}^2],$$

for some universal constant $C > 0$, yielding a contraction at rate $c/16$. We also obtain the second moment bound,

$$\mathbb{E}[\|w\|_{G^+}^2] \leq 2\eta^2\tilde{B}_b^2 + 2\eta\tilde{B}_\sigma^2\mathbb{E}[\|Z'\|_{G^+}^2] \leq 2\eta^2\tilde{B}_b^2 + 2\eta n\tilde{B}_\sigma^2,$$

using $\mathbb{E}[\|Z'\|_{G^+}^2] = \operatorname{tr}(G^+) \leq n$. Substituting, we arrive at

$$\mathbb{E}[f(\|X' - \tilde{Y}\|_{G^+})] \leq \Big(1 - \frac{c\eta}{16}\Big)f(R) + \eta\tilde{B}_b + \sqrt{\eta n}\,\tilde{B}_\sigma + \frac{3}{2r_2}e^{-ar_2}(\eta^2\tilde{B}_b^2 + \eta n\tilde{B}_\sigma^2)$$

$$\leq \Big(1 - \frac{c\eta}{16}\Big)f(R) + 2(\eta\tilde{B}_b + \sqrt{\eta n}\,\tilde{B}_\sigma),$$

completing the proof of Proposition 27. $\qquad\square$

## G    Proofs for the stability of the noisy gradient estimator

For simplicity, we will prove it for the case of fixed step-size $\eta_k = \eta$, but it is easily extendable to the full setting. Using the Wasserstein contraction obtained in the previous section, we will now prove Proposition 14.

**Proposition 14.** *Consider the score matching algorithm $A_{\mathrm{sm}} : S \mapsto s_{\theta_K}$ for some fixed $K \in \mathbb{N}$ where $(\theta_k)_k$ is as given in (16). Suppose that assumptions 10, 12 and 13 hold, then there exists some $\bar{\eta} > 0$ such that, if $\sup_k \eta_k \le \bar{\eta}$, we obtain that $A_{\mathrm{sm}}$ is score stable with constant*

$$\varepsilon_{stab}^2 \lesssim \frac{\overline{L}^2 C}{N} \sqrt{\frac{1}{PN_B^3 \lambda_{gap}}} \exp\left(\frac{\tilde{c} PN_B C^2}{\eta_{\min} \lambda^2}\right) \min\left\{ \sum_{k=0}^{K-1} \eta_k, \exp\left(\frac{\tilde{c} PN_B C^2}{\eta_{\min} \lambda^2}\right) \right\},$$

*for some constant, $\tilde{c}(\overline{M}, \overline{L}, \lambda_{\mathrm{gap}}, \lambda_{\max}(\overline{\Sigma}), B_\ell, C_\tau) > 0$ and $\eta_{\min} = \min_k \eta_k$.*

The proof applies Proposition 27 to the process in (16). As in the proof of Proposition 11, we obtain stability estimates by analysing the trajectories $\theta_k$ and $\tilde{\theta}_k$ trained on $S$ and $S^N$ with coupled minibatch indices.

**Assign variables of Proposition 27.**    Given a set of minibatch indices $B \subset [N]$ with $|B| = N_B$, the dynamics of $\theta_k$ and $\tilde{\theta}_k$ take the form (35)–(36) with

$$b(\theta) := \mathbb{E}\Big[\mathrm{Clip}_C(G(\theta, (x_i)_{i \in B}))\Big|\theta, B, S\Big], \qquad \tilde{b}(\theta) := \mathbb{E}\Big[\mathrm{Clip}_C(G(\theta, (\tilde{x}_i)_{i \in B}))\Big|\theta, B, S^N\Big],$$

$$\sigma(\theta)^2 := \eta \, \Sigma_S(\theta, B), \qquad \tilde{\sigma}(\theta)^2 := \eta \, \Sigma_{S^N}(\theta, B).$$

Using the shorthand $v_{i,j}(\theta) = w_{t_{(i,j)}} \nabla_\theta \|s_\theta(X_{(i,j)}, t_{(i,j)}) - \nabla \log p_{t_{(i,j)}|0}(X_{(i,j)}|x_i)\|^2$, we bound the covariance from below:

$$\Sigma_S(\theta, B) \succcurlyeq \frac{1}{PN_B^2} \sum_{i \in B} \mathrm{Cov}\left(\mathrm{Clip}_C(v_{i,j}(\theta))\Big|\theta, B, S\right) \succcurlyeq \frac{1}{PN_B}\overline{\Sigma}.$$

Therefore $\sigma(\theta)^2 = \eta \Sigma_S(\theta, B) \succcurlyeq \frac{\eta}{PN_B}\overline{\Sigma}$, and similarly for $\tilde{\sigma}$. We set

$$G := \frac{\eta}{2PN_B}\overline{\Sigma},$$

so that $\sigma(\theta)^2, \tilde{\sigma}(\theta)^2 \succcurlyeq G$. The residual volatilities are then

$$\Sigma_G(\theta) = \big(\sigma(\theta)^2 - G\big)^{1/2} = \sqrt{\eta}\big(\Sigma_S(\theta, B) - \tfrac{1}{2PN_B}\overline{\Sigma}\big)^{1/2},$$

and similarly for $\tilde{\Sigma}_G$. The weighted norm satisfies $\|\theta\|_{G^+} \le \sqrt{2PN_B/(\eta\lambda_{\mathrm{gap}})}\,\|\theta\|$.

**Lipschitz constants.**    Since $\mathrm{Clip}_C$ is 1-Lipschitz, Jensen's inequality gives

$$\|b(\theta) - b(\theta')\|_{G^+} \le \mathbb{E}\big[\|G(\theta, (x_i)_{i \in B}) - G(\theta', (x_i)_{i \in B})\|_{G^+}\big|\theta, \theta', B, S\big].$$

Expanding via the chain rule,

$$v_{i,j}(\theta) - v_{i,j}(\theta') = 2w_{t_{i,j}}\Big((\nabla_\theta s_\theta - \nabla_\theta s_{\theta'})^T(s_\theta - \nabla \log p_{t_{i,j}|0}) + (\nabla_\theta s_{\theta'})^T(s_\theta - s_{\theta'})\Big),$$

where all functions are evaluated at $(X_{i,j}, t_{i,j})$. Taking $G^+$-norms and applying the triangle inequality, the Lipschitz constant $L(x, t)$ of $s_\theta$ in $\theta$ and the smoothness constant $M(x, t)$ from Assumption 13 give

$$\|v_{i,j}(\theta) - v_{i,j}(\theta')\|_{G^+}$$
$$\le 2w_{t_{i,j}}\Big(M(X_{i,j}, t_{i,j})\|s_\theta(X_{i,j}, t_{i,j}) - \nabla \log p_{t_{i,j}|0}(X_{i,j}|x_i)\| + L(X_{i,j}, t_{i,j})^2\Big)\|\theta - \theta'\|_{G^+}.$$

Taking expectations and applying Cauchy–Schwarz to each term, using $\mathbb{E}[M^2]^{1/2} \le \mathbb{E}[M^4]^{1/4} \le \overline{M}_4$ and Assumption 10 to bound $\mathbb{E}[\|s_\theta - \nabla \log p_{t|0}\|^2|X_0 = x_i]^{1/2} \le B_\ell/\sigma_t^2$,

$$\mathbb{E}\big[\|v_{i,j}(\theta) - v_{i,j}(\theta')\|_{G^+}\big] \le 2\Big(\overline{M}_4\frac{B_\ell}{\sigma_t^2} + \overline{L}_4^2\Big)\|\theta - \theta'\|_{G^+}.$$

Averaging over $i \in B$ and $j \in [P]$ and integrating against $\tau$ using $C_\tau = \int \sigma_t^{-4} \tau(dt)$,

$$\mathbb{E}\big[\|G(\theta, (x_i)_{i\in B}) - G(\theta', (x_i)_{i\in B})\|_{G^+}\big] \leq 2(\overline{M}_4 B_\ell C_\tau^{1/2} + \overline{L}_4^2)\|\theta - \theta'\|_{G^+},$$

giving $L_b = 2(\overline{M}_4 B_\ell C_\tau^{1/2} + \overline{L}_4^2)$. Furthermore, due to gradient clipping, $\|b(\theta)\|_{G^+} \leq \sqrt{2PN_B/(\eta\lambda_{\text{gap}})}\, C =: B$.

For the Lipschitz constant of $\Sigma_G$, we first bound $\|\Sigma_G(\theta)^2 - \Sigma_G(\theta')^2\|_{op,G^+} = \eta\|\Sigma_S(\theta, B) - \Sigma_S(\theta', B)\|_{op,G^+}$. Writing $Z(\theta) := \text{Clip}_C(G(\theta, (x_i)_{i\in B}))$ and $\mu(\theta) := b(\theta)$, we decompose

$$\Sigma_S(\theta, B) - \Sigma_S(\theta', B) = \mathbb{E}\big[Z(\theta)Z(\theta)^T - Z(\theta')Z(\theta')^T\big] - \big(\mu(\theta)\mu(\theta)^T - \mu(\theta')\mu(\theta')^T\big).$$

For the first term, we write $Z(\theta)Z(\theta)^T - Z(\theta')Z(\theta')^T = (Z(\theta) - Z(\theta'))Z(\theta)^T + Z(\theta')(Z(\theta) - Z(\theta'))^T$ and use $\|Z(\theta)\|, \|Z(\theta')\| \leq C$ by the definition of $\text{Clip}_C$ in (14), giving

$$\big\|\mathbb{E}\big[Z(\theta)Z(\theta)^T - Z(\theta')Z(\theta')^T\big]\big\|_{op,G^+} \leq 2C\,\mathbb{E}\big[\|Z(\theta) - Z(\theta')\|_{G^+}\big].$$

For the second term, the same bound gives $\|\mu(\theta)\mu(\theta)^T - \mu(\theta')\mu(\theta')^T\|_{op,G^+} \leq 2C\|b(\theta) - b(\theta')\|_{G^+}$. Since $\text{Clip}_C$ is 1-Lipschitz, both terms are controlled by $\mathbb{E}[\|G(\theta) - G(\theta')\|_{G^+}]$, and combining,

$$\|\Sigma_S(\theta, B) - \Sigma_S(\theta', B)\|_{op,G^+} \leq 8CL_b\|\theta - \theta'\|_{G^+}. \tag{60}$$

To convert this to a Lipschitz bound on $\Sigma_G$ itself, we apply the following matrix square root perturbation bound.

**Lemma 37.** *Let $A, B, W \in \mathbb{R}^{n \times n}$ be positive semi-definite matrices such that $A, B \succcurlyeq W$ and both $A$ and $B$ commute with $W$. If $\delta$ is the smallest nonzero eigenvalue of $W$, then*

$$\|A^{1/2} - B^{1/2}\|_{op,W} \leq \frac{\|A - B\|_{op,W}}{2\sqrt{\delta}}.$$

*Proof.* Because $A$ and $B$ commute with $W$, their principal square roots $X = A^{1/2}$ and $Y = B^{1/2}$ also commute with $W$, and in particular $X, Y$ and $X - Y$ all preserve the range of $W$. Restricted to $\text{Range}(W)$, the identity $A - B = X(X - Y) + (X - Y)Y$ is a Sylvester equation for $X - Y$. Because $A \succcurlyeq W$, every unit vector $v \in \text{Range}(W)$ satisfies $v^T A v \geq v^T W v \geq \delta$, so the eigenvalues of $X|_{\text{Range}(W)}$ satisfy $\lambda_{\min}(X|_{\text{Range}(W)}) \geq \sqrt{\delta}$, and similarly for $Y$. Applying the standard Sylvester equation operator norm bound via the integral representation $X - Y = \int_0^\infty e^{-Xs}(A - B)e^{-Ys}\, ds$ restricted to $\text{Range}(W)$,

$$\|X - Y\|_{op,W} \leq \frac{\|A - B\|_{op,W}}{\lambda_{\min}(X|_{\text{Range}(W)}) + \lambda_{\min}(Y|_{\text{Range}(W)})} \leq \frac{\|A - B\|_{op,W}}{2\sqrt{\delta}}. \qquad \square$$

To bypass the fact that $\Sigma_S$ may not commute with $G$, we evaluate the matrix square root perturbation in the standard unweighted operator norm, which corresponds to setting $W$ as a scalar multiple of the identity matrix restricted to the range of $G$.

Because $\Sigma_G(\theta)^2 = \eta\Sigma_S(\theta, B) - G \succcurlyeq \frac{\eta}{2PN_B}\bar{\Sigma}$, the minimum non-zero eigenvalue of $\Sigma_G(\theta)^2$ on the range of $\bar{\Sigma}$ is strictly bounded below by $\delta := \frac{\eta\lambda_{\text{gap}}}{2PN_B}$. Thus, setting $W = \delta\hat{I}$ restricted to this range perfectly satisfies the conditions of Lemma 37: $A, B \succcurlyeq W$, and because $W$ is a scaled identity matrix, it trivially commutes with $A$ and $B$.

From Lemma 37, we obtain,

$$\|\Sigma_G(\theta) - \Sigma_G(\theta')\|_{op,\delta\hat{I}} \leq \frac{\eta\|\Sigma_S(\theta, B) - \Sigma_S(\theta', B)\|_{op,\delta\hat{I}}}{2\sqrt{\delta}}.$$

To convert this back to the $G^+$-norm, we use the standard norm equivalence $\|M\|_{op,G^+} \leq \sqrt{\kappa(G)}\|M\|_{op}$ and $\|M\|_{op} \leq \sqrt{\kappa(G)}\|M\|_{op,G^+}$, where the condition number is $\kappa(G) = \frac{\lambda_{\max}(\bar{\Sigma})}{\lambda_{\text{gap}}}$.

Chaining these bounds with (60) yields:

$$\|\Sigma_G(\theta) - \Sigma_G(\theta')\|_{op,G^+} \leq \sqrt{\frac{\lambda_{\max}(\bar{\Sigma})}{\lambda_{\text{gap}}}} \|\Sigma_G(\theta) - \Sigma_G(\theta')\|_{op,\delta\hat{I}}$$

$$\leq \sqrt{\frac{\lambda_{\max}(\bar{\Sigma})}{\lambda_{\text{gap}}}} \frac{\eta\|\Sigma_S(\theta,B) - \Sigma_S(\theta',B)\|_{op,\delta\hat{I}}}{2\sqrt{\delta}}$$

$$\leq \frac{\lambda_{\max}(\bar{\Sigma})}{\lambda_{\text{gap}}} \frac{\eta\|\Sigma_S(\theta,B) - \Sigma_S(\theta',B)\|_{op,G^+}}{2\sqrt{\frac{\eta\lambda_{\text{gap}}}{2PN_B}}}$$

$$\leq \frac{\lambda_{\max}(\bar{\Sigma})}{\lambda_{\text{gap}}} \frac{\eta(8CL_b\|\theta-\theta'\|_{G^+})}{2}\sqrt{\frac{2PN_B}{\eta\lambda_{\text{gap}}}}$$

$$= 4CL_b\frac{\lambda_{\max}(\bar{\Sigma})}{\lambda_{\text{gap}}}\sqrt{\frac{2\eta PN_B}{\lambda_{\text{gap}}}}\|\theta-\theta'\|_{G^+}.$$

We therefore obtain $L_\sigma = 4CL_b\frac{\lambda_{\max}(\bar{\Sigma})}{\lambda_{\text{gap}}}\sqrt{\frac{2\eta PN_B}{\lambda_{\text{gap}}}}$. Since $L_\sigma^2 \propto \eta$, Assumption 26 is satisfied for $\eta$ sufficiently small.

**Mismatch bounds.** Since $\tilde{x}_i = x_i$ for all $i \neq N$, the gradient estimators are identical whenever $N \notin B$, justifying the indicators $\mathbb{1}_{N\in B}$ throughout. On the event $\{N \in B\}$, since $\text{Clip}_C$ is 1-Lipschitz and only the $N$-th summand of $G$ differs between $S$ and $S^N$,

$$\|b(\theta) - \tilde{b}(\theta)\| \leq \mathbb{E}\big[\|G_S - G_{S^N}\|\big] = \frac{1}{N_B}\mathbb{E}\big[\|g_N(\theta;x_N) - g_N(\theta;\tilde{x}_N)\|\big] \leq \frac{2C}{N_B},$$

where $g_N(\theta;x) := \frac{1}{P}\sum_{j=1}^P v_{N,j}(\theta;x)$ and the last step uses $\|g_N(\theta;x)\| \leq C$ (since each $\|v_{N,j}\|$ is bounded by $C$ before clipping, by Assumptions 10 and 13). Converting to the $G^+$-norm via $\lambda_{\max}(G^+)^{1/2} = \sqrt{2PN_B/(\eta\lambda_{\text{gap}})}$,

$$\|b(\theta) - \tilde{b}(\theta)\|_{G^+} \leq 2C\sqrt{\frac{2P}{\eta N_B\lambda_{\text{gap}}}}\mathbb{1}_{N\in B} =: \tilde{B}_b.$$

For the residual volatility mismatch, we use that $\Sigma_G(\theta)^2 - \tilde{\Sigma}_G(\theta)^2 = \eta(\Sigma_S(\theta,B) - \Sigma_{S^N}(\theta,B))$, which is nonzero only when $N \in B$. On this event, the same covariance perturbation argument as above, with $Z_S := \text{Clip}_C(G(\theta,(x_i)_{i\in B}))$, $Z_{S^N} := \text{Clip}_C(G(\theta,(\tilde{x}_i)_{i\in B}))$ and $\|Z_S\|, \|Z_{S^N}\| \leq C$ by definition of $\text{Clip}_C$, gives

$$\|\Sigma_S(\theta,B) - \Sigma_{S^N}(\theta,B)\|_{op} \leq 4C\,\mathbb{E}[\|Z_S - Z_{S^N}\|] \leq \frac{8C^2}{N_B}\mathbb{1}_{N\in B},$$

where the second step uses $\|Z_S - Z_{S^N}\| \leq \|G_S - G_{S^N}\| \leq 2C/N_B$. Applying Lemma 37 as before with $\delta = \eta\lambda_{\text{gap}}/(2PN_B)$,

$$\|\Sigma_G(\theta) - \tilde{\Sigma}_G(\theta)\|_{op,G^+} \leq \frac{\eta\|\Sigma_S(\theta,B) - \Sigma_{S^N}(\theta,B)\|_{op,W}}{2\sqrt{\eta\lambda_{\text{gap}}/(2PN_B)}}\sqrt{\frac{\lambda_{\max}(G^+)}{\lambda_{\min}(G^+)}}$$

$$\leq \frac{4\eta C^2}{N_B}\sqrt{\frac{2PN_B}{\eta\lambda_{\text{gap}}}}\sqrt{\frac{\lambda_{\max}(G^+)}{\lambda_{\min}(G^+)}}\mathbb{1}_{N\in B}$$

$$= 4C^2\sqrt{\frac{2\eta P}{N_B\lambda_{\text{gap}}}}\sqrt{\frac{\lambda_{\max}(\bar{\Sigma})}{\lambda_{\text{gap}}}}\mathbb{1}_{N\in B} =: \tilde{B}_\sigma.$$

To verify the condition (39), we require $\eta\tilde{B}_b + \sqrt{\eta n}\,\tilde{B}_\sigma \leq c\eta r_1/256$. We have

$$\eta\tilde{B}_b + \sqrt{\eta n}\,\tilde{B}_\sigma \leq \sqrt{\frac{2\eta P}{N_B\lambda_{\text{gap}}}}\bigg(2C + 4C^2\sqrt{\eta n}\sqrt{\frac{\lambda_{\max}(\bar{\Sigma})}{\lambda_{\text{gap}}}}\bigg)\mathbb{1}_{N\in B}.$$

Meanwhile, $r_1 = 4(1 + \eta_0 L_b)B/\lambda \sim PN_BC/(\eta\lambda_{\text{gap}}\lambda)$, so $c\eta r_1 \sim PN_BC/\lambda_{\text{gap}}$. Since the right-hand side grows as $PN_B/\lambda_{\text{gap}}$ while the mismatch terms grow as $\sqrt{\eta(P+n)/(N_B\lambda_{\text{gap}})}$, the condition is satisfied for $\eta$ sufficiently small.

**Applying Proposition 27.** With all assumptions verified, Proposition 27 gives, writing $d(\theta, \theta') = f(\|\theta - \theta'\|_{G^+})$,

$$\mathbb{E}[d(\theta_{k+1}, \tilde{\theta}_{k+1})|\theta_k, \tilde{\theta}_k, B_k] \leq \left(1 - \frac{c\eta}{16}\right)d(\theta_k, \tilde{\theta}_k) + 2(\eta\tilde{B}_b + \sqrt{\eta n}\,\tilde{B}_\sigma). \tag{61}$$

Since $\tilde{B}_b$ and $\tilde{B}_\sigma$ are nonzero only on $\{N \in B_k\}$, which has probability $N_B/N$, taking full expectations gives

$$\mathbb{E}[\eta\tilde{B}_b + \sqrt{\eta n}\,\tilde{B}_\sigma] \leq \frac{2C}{N}\sqrt{\frac{2\eta P}{N_B\lambda_{\text{gap}}}}\left(1 + 2C\sqrt{\eta n}\sqrt{\frac{\lambda_{\max}(\bar{\Sigma})}{\lambda_{\text{gap}}}}\right),$$

where the second inequality holds for $\eta$ sufficiently small. Taking full expectations in (61),

$$\mathbb{E}[d(\theta_{k+1}, \tilde{\theta}_{k+1})] \leq \left(1 - \frac{c\eta}{16}\right)\mathbb{E}[d(\theta_k, \tilde{\theta}_k)] + \frac{4C}{N}\sqrt{\frac{2\eta P}{N_B\lambda_{\text{gap}}}}\left(1 + 2C\sqrt{\eta n}\sqrt{\frac{\lambda_{\max}(\bar{\Sigma})}{\lambda_{\text{gap}}}}\right), \tag{62}$$

Iterating (62) with $d(\theta_0, \tilde{\theta}_0) = 0$ (since $\theta_0 = \tilde{\theta}_0$) and applying the formula from the start of this section,

$$\mathbb{E}[d(\theta_K, \tilde{\theta}_K)] \leq \frac{4C}{N}\sqrt{\frac{2\eta P}{N_B\lambda_{\text{gap}}}}\left(1 + 2C\sqrt{\eta n}\sqrt{\frac{\lambda_{\max}(\bar{\Sigma})}{\lambda_{\text{gap}}}}\right)\sum_{k=0}^{K-1}\left(1 - \frac{c\eta}{16}\right)^k \tag{63}$$

$$\leq \frac{4C}{N}\sqrt{\frac{2P}{N_B\eta\lambda_{\text{gap}}}}\left(1 + 2C\sqrt{\eta n}\sqrt{\frac{\lambda_{\max}(\bar{\Sigma})}{\lambda_{\text{gap}}}}\right)\min\left(\eta K, \frac{16}{c}\right). \tag{64}$$

Since $f(r) \geq \frac{e^{-ar_2}}{2r_2}r^2$ for all $r \geq 0$, we have $\|\theta - \theta'\|_{G^+}^2 \leq 2r_2 e^{ar_2}\,d(\theta, \theta')$. Since Assumption 13 states that $s_\theta$ is Lipschitz in $\theta$ with respect to $\|\cdot\|_{\bar{\Sigma}^+}$, and $\|\cdot\|_{\bar{\Sigma}^+}^2 = \frac{\eta}{2PN_B}\|\cdot\|_{G^+}^2$,

$$\int \mathbb{E}\big[\|s_{\theta_K}(X_t, t) - s_{\tilde{\theta}_K}(X_t, t)\|^2\big|X_0 = \tilde{x}, S\big]\tau(dt) \tag{65}$$

$$\leq \bar{L}^2\mathbb{E}\big[\|\theta_K - \tilde{\theta}_K\|_{\bar{\Sigma}^+}^2\big]$$

$$= \frac{\eta\bar{L}^2}{2PN_B}\mathbb{E}\big[\|\theta_K - \tilde{\theta}_K\|_{G^+}^2\big]$$

$$\leq \frac{\eta\bar{L}^2 r_2 e^{ar_2}}{PN_B}\mathbb{E}[d(\theta_K, \tilde{\theta}_K)]$$

$$\leq 4\bar{L}^2 r_2 e^{ar_2}\frac{C}{N}\sqrt{\frac{2\eta}{PN_B^3\lambda_{\text{gap}}}}\left(1 + 2C\sqrt{\eta n}\sqrt{\frac{\lambda_{\max}(\bar{\Sigma})}{\lambda_{\text{gap}}}}\right)\min\left(\eta K, \frac{16}{c}\right). \tag{66}$$

When $\eta$ is sufficiently small, we obtain $\eta_0 \gtrsim \lambda^{-1}$ and therefore

$$r_1^2, r_2^2 \gtrsim \frac{PN_BC^2}{\eta\lambda_{\text{gap}}\lambda^2}, \qquad L \lesssim (\bar{M}_4 B_\ell C_\tau^{1/2} + \bar{L}_4^2)(PN_B\lambda_{\text{gap}})^{-1/2} \vee 1,$$

and since $L$ and $r_1$ grow as $\eta \to 0^+$, we also have $r_2^2 c \gtrsim \exp(-6Lr_1^2/c_0)$. Substituting these estimates into (66) and absorbing $r_2 e^{ar_2}$ into the $\lesssim$ yields the bound in the statement of Proposition 14.

