# OpenReview forum: "Implicit Regularisation in Diffusion Models: An Algorithm-Dependent Generalisation Analysis"
_ICLR.cc/2026/Conference — ICLR 2026 Poster_

### Official Review · Reviewer_dqKw · 2025-10-31

**Soundness:** 3
**Presentation:** 3
**Contribution:** 3
**Rating:** 6
**Confidence:** 4

**Summary:**

This work studies the implicit regularization of denoising score matching, brought by the early stopping, timestep discretization and optimization algorithm. The analysis is based on a novel score stability framework, which converts the generalization bound into the sum of empirical score matching error plus a stability term.

**Strengths:**

1. The proposed score stability framework is sound and novel.

2. The analysis is comprehensive, covering a broad range of aspects, including discretization, early stopping and optimization algorithm.

**Weaknesses:**

1. No empirical verification of the theorem is provided.

2. it is unclear to me how the results explain how diffusion model generalizes despite the empirical optimum can only memorize.

3. Empirically, early stopping and discretization are no the key to generalization.

**Questions:**

1. Can you provide some simulation to validate the theory? For example, just considering learning the score of a simple Gaussian distribution? Can you show by early stopping or using a coarser discretization, the generalization (KL divergence) is improved? Currently, the bound is not  intuitive for interpretation without solid example.

2. How can your theory explain diffusion model can generalize despite the empirical optimum can only memorize?

3. Although the theoretical result is rigorous, in practice, early stopping and discretization are not the key factors of generalization. What do you think about this? What are other factors that enable generalization?

---

> ### Author Response · Authors · 2025-11-24
>
> We thank the reviewer for their thorough comments and suggestions. We address their listed weaknesses and questions below.
>
> ## "How can your theory explain diffusion model can generalize despite the empirical optimum can only memorize? [...] early stopping and discretization are not the key factors..."
>
> In the introduction, we build on Pidstrigach (2022), who shows that under idealised conditions (near perfect optimisation and sampling) the empirical denoising optimum memorises the training data. Our work studies two deviations from that regime:
> (i) imperfect opt. / alg. bias (Section 6), and
> (ii) imperfect sampling via finite-time and finite-step samplers (Sections 4–5).
>
> In the imperfect sampling setting we assume the empirical denoising minimiser, but we show that with early stopping and/or coarse time discretisation—both standard in practice—one can obtain non-trivial population KL guarantees. We do not claim that early stopping or discretisation are the only or  dominant sources of generalisation in diffusion models. Our claims are more modest here: that these techniques activate a source of implicit regularisation in the empirical objective that can produce generalisation without further regularisation. Since these are common techniques which are usually motivated by efficiency or numerical stability, we believe that this theoretical connection to generalisation could have a practical impact on how people approach diffusion model research moving forward.
>
> These mechanisms complement more familiar sources of inductive bias, such as architectural bias and optimisation bias. Section 6 focuses on the latter: we isolate two sources of implicit reg. that are often overlooked whilst acknowledging that diffusion models are complex and that a complete understanding of their generalisation cannot be obtained in a single research paper—it will requires a lot of further work. We believe the score stability framework and the examples we provide are a useful building block for this larger ongoing direction of research.
>
> ## “Can you provide some simulation to validate the theory?... a simple Gaussian distribution?”
>
> We thank the reviewer for this concrete suggestion. Following it, we considered the 1D Gaussian toy setting, considering the empirical score on a dataset of 40 samples. See the figures in the document in the supplementary material.
>
> We plot KL against discretisation steps observing a U-shaped behaviour: increasing the number of steps improves KL but beyond a certain point, further refinement worsens KL. This matches the trade-off in Proposition 7 between discretisation error (improving with more steps) and the implicit regularisation effect (decreasing with more steps). A second figure shows how this relationship is affected by changing the early stopping time. A third figure shows histograms of samples: with few steps, the distribution is visibly smoothed between data points at the expense of further bias.
>
> Proposition 8 explains how, when this trade-off is optimally balanced, one can obtain near-linear decay of population KL in $N$, once large enough relative to $\epsilon^{-d^\ast}$. We agree that this behaviour is not intuitive a priori, which is why we believe a theoretical treatment is valuable.
>
> ## “No empirical verification of the theorem is provided.”
>
> Scaling this to realistic high-dimensional diffusion models is challenging for the following reasons:
>
> 1. High-dimensional KL estimation requires a very large number of samples. Even in 1D we needed thousands of samples for a low-variance estimate, and diffusion-model distributions can be poorly conditioned for density estimation (e.g., mass near low-dimensional manifolds).
> 2. Our theory, like much of generalisation theory, focuses on how population error changes with $N$. Estimating rates would require multiple runs for many large values of $N$. Since training practical diffusion models can take days per run, this quickly becomes prohibitively expensive.
> 3. Sections 4–5 assume access to the exact empirical denoising minimiser. Approximating this would amount to deliberately overfitting a large score network, and it is unclear whether standard optimisers would converge to this solution rather than to an inductively biased one.
>
> Precisely because these regimes are hard to probe experimentally, we see rigorous analysis as particularly valuable. Experimental and theoretical studies of generalisation in diffusion models often require very different setups, which is why several influential works have focused on purely theoretical questions [1, 2, 3, 4] and others focus only on experiments. We view detailed experimental verification as important follow-up work, but not essential for the current theoretical contribution.
>
> [1] "Diffusion Models are...", Oko et al. ICLR workshop/ICML 2023
> [2] "Score Approximation...", Chen et al. ICML 2023
> [3] "Theoretical guarantees...", Silveri et al. NeurIPS 2024
> [4] "On the Generalization...", Li et al. NeurIPS 2024

---

> > ### Comment · Reviewer_dqKw · 2025-11-27
> >
> > Thanks for your rebuttal. I believe the paper has improved by adding the extra empirical experiments. Hence I increase my score.

---

> > > ### Author Response · Authors · 2025-12-02
> > >
> > > We greatly appreciate the reviewer considering increasing their score, and we agree that the synthetic experiment has made the paper easier to follow. For the camera-ready version, we will use the additional page to integrate this experiment with the main body.
> > >
> > > The link for the document has expired, but it is now enclosed within the supplementary material for the reviewer's interest.

---

### Official Review · Reviewer_HHPm · 2025-11-01

**Soundness:** 3
**Presentation:** 3
**Contribution:** 2
**Rating:** 4
**Confidence:** 2

**Summary:**

Diffusion models generalize their training data, and how this happens must depend on some combination of (i) the training objective, (ii) optimization (e.g., using SGD), and (iii) the details of sampling. Prior work has attempted to bound generalization error given various assumptions (e.g., by assuming the 'manifold hypothesis'), but these bounds might neglect various details important in practice. The authors attempt to rectify this by introducing the notion of "score stability", and show that this notion allows them to derive interesting bounds on generalization error.

**Strengths:**

The paper is relatively clearly written and the math claims appear to be rigorous (although I did not check them in detail). The notion of score stability is interesting and appears to be a useful conceptual/technical contribution. In the main text, the authors strike a good balance between presenting their results and presenting the intuition for them, without overwhelming the reader with technical detail.

**Weaknesses:**

I understand that the point of the paper is to prove bounds in the tradition of recent formal (mathematical) work on diffusion models, but I am still left wondering about the extent to which these bounds are tight and/or interesting in practice. There are no experiments in the paper; would it be possible to conduct any experiments to show that the bounds are interesting, or that they are at least qualitatively consistent with what one sees in experiments? Relatedly, figures (either showing schematics, or experiments) would be helpful in getting the reader to follow the arguments.

Also relatedly, are there any interesting (even toy) settings where the reader can analytically see that these bounds are interesting, because various things can be explicitly computed?

There is some interesting work on how mislearning in diffusion models supports generalization. Three examples I can think of are the Kadkhodaie et al. 2024 geometry-adaptive harmonic representations paper (https://arxiv.org/abs/2310.02557), the Kamb and Ganguli 2025 combinatorial creativity paper (https://arxiv.org/abs/2412.20292), and the Vastola 2025 generalization through variance paper (https://arxiv.org/abs/2504.12532). The authors don't have to cite these if they don't find them interesting/appropriate; I just mean to say that there's other work (not in the same formal mathematical tradition the authors follow) on the basic question they're interested in. This work may be useful in framing their contribution.

**Questions:**

1. Can the authors show via some numerical experiments that their bounds are tight or at least qualitatively interesting?

2. Can the authors come up with a toy setting where things can be explicitly computed, which may help show that their derived bounds are interesting?

---

> ### Author Response · Authors · 2025-11-24
>
> We thank the reviewer for their thorough review and address concerns below. We that hope they consider increasing the rating from a weak reject.
>
> ## Related non-theoretic papers
> We thank the reviewer for highlighting these recent works, in particular Vastola, which reaches a conclusion closely aligned with our Section 4. Our current related work section focuses on theoretical papers; in the revision we will broaden it to discuss these position-style works.
>
> ## “[...] wondering about the extent to which these bounds are tight and/or interesting in practice”
> Our work has several interesting qualitative and theoretical contributions. We will emphasise these takeaways more clearly. In brief:
>
> **Qualitative insights**
> * Early stopping/coarse discretisation can enable generalisation despite overfitting. Beyond efficiency or numerical stability, we show they activate an implicit regularisation effect of denoising score matching (“denoising regularisation”) that yields population guarantees even when the empirical loss is minimised, echoing and extending Vastola’s intuition.
> * Coarse discretisation induces a bias–generalisation trade-off. More steps reduce discretisation bias but weaken denoising regularisation; suitably tuned discretisation can still generalise in the overfitted regime.
> * Diffusion vs classical overparameterised learning. In our setting, generalisation is possible in an overparameterised, overfitted regime without explicit regularisation, in contrast to standard overparameterised supervised-learning intuitions.
> * Minibatch noise can be beneficial. The large optimisation noise typical of diffusion training can promote score stability and hence generalisation, rather than being purely a nuisance.
>
> **Theoretical insights**
> * First alg.-dep. framework for gen. in diffusion models.
> * Function-space view that links DSM to convex ERM stability.
> * Use of Harnack inequalities to connect OU-process smoothing to gen.
> * Denoising effectively smooths the opt. landscape, requiring only on-average smoothness for SGD
> * Noise in diffusion model SGD can produce Wasserstein contractions.
>
> We do not claim our upper bounds are tight. Establishing tightness would require matching lower bounds under comparable assumptions, which we view as natural future work rather than something numerical experiments could resolve.
>
> ## Tightness and approximation
>
> Our approach follows the long tradition of using inequalities and non-asymptotic bounds to study learning algorithms in realistic regimes. For comparison, Vastola instead considers a highly simplified limit (first-order probability flow approximation, linear scores, no sampler discretisation, $N \to \infty$), which allows explicit formulas but only in an idealised setting.
>
> We analyse an exact practical sampler (no continuous limit), a general class of scores, and finite $N$, at the cost of bounds rather than closed forms. Our bounds provide non-asymptotic control of the generalisation gap, whereas Vastola, and papers like it only inuit a relationship with generalisation. These approaches are complementary: neither our bounds nor such approximations are absolutely “tight”, but together they build a more complete conceptual picture of why diffusion models generalise.
>
> ## “Can the authors come up with a toy setting…?” / “There are no experiments…” / Diagrams
>
> Motivated by this comment (and dqKw’s), we added a simple **1D Gaussian toy example**. Please observe "Can you provide some simulation to validate the theory?" in our rebuttal for dqKw. We will include these figures and a schematic of the Section 4 proof sketch in the revised version.
>
> Extending this analysis to realistic, high-dimensional models is challenging because:
> 1. KL estimation in high dimensions requires very large sample sizes; even in 1D we needed thousands of samples for a stable estimate.
> 2. Our theory focuses on how population error scales with $N$; probing this would require many large-$N$ runs, each expensive to train.
> 3. Sections 4–5 assume the exact empirical denoising minimiser; approximating this would require deliberate overfitting, and standard optimisers may produce inductive bias.
>
> As its hard to probe generalisation rates empirically, we see rigorous analysis as particularly valuable. Several influential works on diffusion models have taken a purely theoretical route [1–4], while others are purely empirical since these perspectives require different setups. We view detailed large-scale experiments as important follow-up work, but not as a prerequisite for a theoretical contribution. Even if the reviewer is not interested in this kind of analysis, we hope that they consider the interest to the wider ML community, as indicated by previous highly cited publications.
>
> [1] “Diffusion Models are…”, Oko et al., ICLR Workshop / ICML 2023
>
> [2] “Score Approximation…”, Chen et al., ICML 2023
>
> [3] “Theoretical Guarantees…”, Silveri et al., NeurIPS 2024
>
> [4] “On the Generalization…”, Li et al., NeurIPS 2024

---

> > ### Author Response · Authors · 2025-12-02
> >
> > We were not aware that we could add an additional comment, so we will now share the brief discussion on the Gaussian toy example that was shared with reviewer dqKw. The experiments can be found in the document "Gaussian numerical example for ICLR reviewers.pdf" in the supplementary material.
> >
> > We plot KL against discretisation steps observing a U-shaped behaviour: increasing the number of steps improves KL but beyond a certain point, further refinement worsens KL. This matches the trade-off in Proposition 7 between discretisation error (improving with more steps) and the implicit regularisation effect (decreasing with more steps). A second figure shows how this relationship is affected by changing the early stopping time. A third figure shows histograms of samples: with few steps, the distribution is visibly smoothed between data points at the expense of further bias.
> >
> > Proposition 8 explains how, when this trade-off is optimally balanced, one can obtain near-linear decay of population KL in $N$, once large enough relative to $\epsilon^{-d^\ast}$. We agree that this behaviour is not intuitive a priori, which is why we believe a theoretical treatment is valuable.
> >
> > We would like to stress to the reviewer that even in this simple scenario where we had access to the empirical risk minimiser, a great deal of computation was required. Extending the experiments to large-scale realistic settings would be highly computationally difficult, and far exceeds the scope of this work. We believe that, because this subject (the scaling of the expected generalisation gap as a function of $N$, taking an algorithmic-viewpoint) is difficult to analyse experimentally, theoretical frameworks like the one that we propose in this work can be of use. While we understand the reviewer's scepticism towards this type of theoretical work, we hope that they take into account that this is a direction of research that is of interest to a large part of the the ICLR and broader diffusion model research community.

---

### Official Review · Reviewer_bmqA · 2025-11-03

**Soundness:** 3
**Presentation:** 2
**Contribution:** 2
**Rating:** 4
**Confidence:** 3

**Summary:**

This paper studies generalisation in denoising diffusion models from an algorithm-dependent perspective. It introduces score stability, a measure of how sensitive score-matching algorithms are to changes in the training data, and uses it to derive generalisation gap bounds. The authors identify three sources of implicit regularisation: (i) early stopping in the denoising process, (ii) coarse discretisation in the sampling algorithm, and (iii) optimisation noise from SGD with gradient clipping and weight decay. They show that each of these sources improves generalisation in different ways, such as limiting memorisation, controlling discretisation error, or inducing contractive training dynamics. By grounding the analysis in algorithmic properties rather than model structure, the paper provides a unifying theoretical framework for understanding why diffusion models generalise despite high dimensionality. Overall, it links practical training heuristics to formal generalisation guarantees, revealing overlooked mechanisms that naturally regularise diffusion models.

**Strengths:**

**1. Introduction of an algorithm-dependent generalization framework:** The paper introduces score stability, a novel, algorithm-dependent framework for analyzing generalization in diffusion models. Unlike previous works that rely on algorithm-independent uniform convergence bounds, this approach explicitly quantifies how training and sampling algorithms affect generalization. By linking the sensitivity of the learned score function to single-sample perturbations with the expected generalization gap, the framework provides theoretical insight into why diffusion models generalize despite high dimensionality.

**2. Identification of multiple sources of implicit regularization:** The paper systematically analyses three distinct sources of implicit regularization in diffusion model training: denoising regularisation via early stopping, sampler regularisation through coarse discretization, and optimization regularization induced by stochastic gradient dynamics. By grounding these effects in the algorithmic framework of score stability, the work reveals mechanisms that were previously overlooked in the literature.

**Weaknesses:**

**1. Difficulty in computing score stability:** While the paper introduces the notion of score stability and shows its theoretical connection to generalization, it does not provide a clear or practical method for computing or estimating $\epsilon_{stab}$ for a given algorithm and dataset. In practice, evaluating score stability requires measuring the sensitivity of the learned score function to changes in individual data points, which may involve retraining or coupling multiple stochastic outputs—a process that can be computationally expensive, especially for large datasets and high-dimensional diffusion models.

**2. The loss function for diffusion models has no integration over time:** The analysis in the paper treats the diffusion model loss as time-agnostic, meaning it does not explicitly integrate or weigh contributions across different noise levels or time steps. In practice, the difficulty of learning the score function varies with time: early steps with high noise are often easier to model, whereas later steps with low noise are more sensitive and crucial for high-quality generation. Ignoring this temporal structure may oversimplify the analysis and obscure how generalization and stability depend on the progression of the diffusion process.

**Questions:**

**Q1.** How can score stability be practically estimated for large-scale diffusion models?

**Q2.** Can the framework be extended to conditional or latent diffusion models?

**Q3.** How sensitive are the generalisation bounds to different optimisation schemes?

**Details Of Ethics Concerns:**

N.A.

---

> ### Author Response · Authors · 2025-11-23
>
> We thank the reviewer for their thoughtful feedback and questions, which we address in the discussion below. We also thank the reviewer for recognising the contribution of our work in proposing a bespoke algorithm-dependent analysis for diffusion models as well as identifying several sources of generalisation that are often overlooked in the diffusion model literature.
>
> We hope the reviewer agrees that we have addressed most of the issues raised in their review, and we hope that if the reviewer agrees, they will consider increasing their score from a marginal reject.
>
> **R1: “The loss function for diffusion models has no integration over time”**
> The reviewer appears to be mistaken here. Both the score matching and denoising score matching objectives in our work explicitly integrate over time with respect to a time-weighting measure $\tau$ (see Eqs. (3) and (6)). Also, both optimisation schemes we consider use randomly sampled values of $t \sim \tau$ at each update. We agree that omitting the integration over time would be a serious mistake, and we will clarify this more prominently in a revised version.
>
> **R2: “Can the framework be extended to conditional or latent diffusion models?”**
>
> Yes. The framework can naturally be applied to conditional diffusion models. One simply updates Definition 2 to work with the conditional score $s(x, t \mid x_{\text{obs}})$ and the corresponding conditional score with the $i$-th example exchanged, $s^{(i)}(x, t \mid x_{\text{obs}})$, and defines score stability in terms of the discrepancy between these two.
>
> For latent diffusion models, let $\phi$ be the decoder from latent to data space. If $p_\epsilon$ and $q_K$ are distributions in latent space, then by the data processing inequality for KL divergence
> $$
> D(\phi_* p_{\epsilon} | \phi_* q_K) \leq D(p_{\epsilon} | q_K),
> $$
> so we can carry our analysis over to data space with no worse KL error.
>
> We thank the reviewer for pointing this out and will include a brief discussion of these extensions. We also view this as a promising future direction that highlights the generality of the score stability framework.
>
> **R3: “How sensitive are the generalisation bounds to different optimisation schemes?”**
>
> This is an important question. In our analysis, the dependence on the optimisation scheme is captured entirely through the score stability constant, which then controls the generalisation gap. In Section 6.1 we relate score stability to the standard notion of uniform argument stability from the learning theory literature, and in Section 6.2 we show that, under additional contractivity assumptions, the resulting bounds do not grow with the number of optimisation steps.
>
> A more systematic comparison of different optimisation algorithms through their induced score stability constants would be very interesting, and we believe our framework provides a natural way to do this. However, such a study would form a separate paper and is beyond the scope of the present work.
>
> **R4: “Difficulty in computing score stability [...] How can score stability be practically estimated for large-scale diffusion models?”**
>
> We emphasise that score stability is introduced here as a *theoretical* concept, analogous to VC dimension or Rademacher complexity; our results do not require practitioners to compute it in order to train or use diffusion models.
>
> The reviewer may be suggesting that score stability could be turned into an algorithmic tool, for example to prevent memorisation or to design principled regularisers. We agree that this is an appealing direction, and one could imagine approximate estimators based on influence-function or Monte Carlo estimation. However, we see this as interesting follow-up work that lies well outside the scope of the current paper, which is purely a theoretical investigation.

---

> > ### Comment · Reviewer_bmqA · 2025-11-27
> >
> > Thank the reviewer for addressing my concerns. I have raised my score.

---

### Meta-Review · Area_Chair_W2tx · 2026-01-13

**Summary:**

The paper develops an algorithm-dependent generalisation framework for diffusion models based on "score stability" and applies it to identify implicit sources of regularisation in several learning settings. The reviews found the paper clearly written, the analytic framework sound and novel, and the analysis comprehensive. The reviewers raised concerns about the lack of empirical validation and practical applicability, the difficulty in estimating score stability, and the lack of integration over time in the loss.

**Reviewer Concerns:**

The rebuttal addressed the lack of empirical validation by adding a toy experiment and clarified concerns about integration over time and the difficulty of estimating score stability. Concerns about the usefulness of the bounds and the absence of large-scale empirical validation remain unresolved.

**Reviewer Scores:**

Reviewers bmqA and dqKw would likely increase their scores based on the responses, and have also stated so. Reviewer HHPm would likely not be satisfied with the scale of the added experiments, and the reviewer's score would likely remain unchanged. If both Reviewers bmqA and dqKw increased their scores, the overall evaluation would shift sufficiently toward acceptance, and I therefore recommend accepting the paper.

---

### Decision · Program_Chairs · 2026-01-26

Accept (Poster)